# Beyond Masks: Efficient, Flexible Diffusion Language Models via Deletion-Insertion Processes

**Fangyu Ding**[1,2,*]   **Ding Ding**[1,2,*]   **Sijin Chen**[3,*]   **Kaibo Wang**[1,2]
**Peng Xu**[2]   **Zijin Feng**[2]   **Haoli Bai**[2]   **Kai Han**[2]   **Youliang Yan**[2]
**Binhang Yuan**[1,†]   **Jiacheng Sun**[2,†]
[1]HKUST   [2]Huawei Foundation Model Dept   [3]CUHK

## Abstract

While Masked Diffusion Language Models (MDLMs) relying on token masking and unmasking have shown promise in language modeling, their computational efficiency and generation flexibility remain constrained by the masking paradigm. In this paper, we propose Deletion-Insertion Diffusion language models (DID) that rigorously formulate token deletion and insertion as discrete diffusion processes, replacing the masking and unmasking processes in current MDLMs. DID improves training and inference efficiency by eliminating two major sources of computational overhead in MDLMs: the computations on non-informative 1) `<MASK>` tokens inherent to the paradigm, and 2) `<PAD>` tokens introduced in variable-length settings. Furthermore, DID offers greater flexibility by: 1) natively supporting variable-length sequences without requiring fixed-length padding, and 2) an intrinsic self-correction mechanism during generation due to insertion that dynamically adjusts token positions. To train DID, we design a score-based approach that assigns scores to token insertion operations and derive appropriate training objectives. The objectives involve subsequence counting problems, which we efficiently solve via a parallelized dynamic programming algorithm. Our experiments across fixed and variable-length settings demonstrate the advantage of DID over baselines of MDLMs and existing insertion-based LMs, in terms of modeling performance, sampling quality, and training/inference speed, without any hyperparameter tuning.

https://github.com/FMD-NEXT/DID

## 1 Introduction

Diffusion language models (DLMs) (Austin et al., 2021; Campbell et al., 2022; Lou et al., 2024) have rapidly emerged as a powerful paradigm for language modeling, offering a compelling alternative to the dominant autoregressive (AR) approach. They offer distinct advantages, including bidirectional context modeling and the potential for parallel decoding. Within this domain, a body of work on Masked Diffusion Language Models (MDLMs) (Nie et al., 2024; 2025; Ou et al., 2025; Sahoo et al., 2024; Shi et al., 2024) is the most widely studied. These models operate through a forward process that progressively corrupts each token into an absorbing state `<MASK>` and a backward process that reconstructs the original sequence by iteratively unmasking tokens from a fully masked sequence, with a fixed sequence length during the diffusion process.

Despite their success, MDLMs are fundamentally limited by their fixed sequence length. The first issue lies in their restricted generation flexibility, which leads to challenges in modeling variable lengths and performing self-correction: once a token is unmasked, its content and position become fixed, thereby risking error accumulation in a similar sense to autoregressive models. The second issue is their substantial computational inefficiency, as the model must repeatedly process full-length sequences. Under the typical log-linear noise schedule, about half of the FLOPs are allocated to the non-informative `<MASK>` tokens during both training and inference. Further, if MDLMs are applied

---

*Equal contribution.
†Correspondence to {biyuan@ust.hk, sunjiacheng1@huawei.com}.

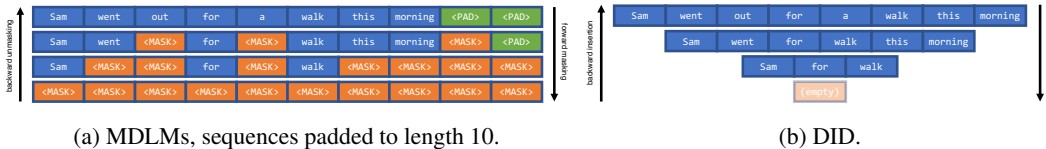

(a) MDLMs, sequences padded to length 10.        (b) DID.

Figure 1: Conceptual diagrams of MDLMs and Deletion-Insertion Diffusion language models (DID).

to variable-length sequences, their fixed-length nature demands padding to the same length (Nie et al., 2024; 2025; Wu et al., 2025b; Gong et al., 2024) (Fig. 1a), allocating extra FLOPs to the non-informative `<PAD>` tokens. This means generating a shorter sequence is not faster.

To address these issues, we propose Deletion-Insertion Diffusion language models (DID), a novel discrete diffusion paradigm that fundamentally differs from MDLMs. DID replaces the masking-unmasking processes in MDLMs with deletion-insertion processes (Fig. 1b). Specifically, tokens are progressively deleted in the forward process until the sequence is empty; in the backward process, generation starts from an empty sequence and iteratively inserts tokens until a complete sequence is reconstructed. DID eliminates the `<MASK>` and `<PAD>` tokens used in MDLMs, saving FLOPs and improving computational efficiency. Regarding generation flexibility, DID natively supports variable-length data, and as an insertion-based language model, features an intrinsic self-correction mechanism that dynamically adjusts token positions during generation.

We implement DID by addressing several non-trivial design and training challenges. First, we rigorously formulate the deletion and insertion processes within the discrete diffusion framework, and develop a score-based approach built upon the Denoising Score Entropy (DSE) (Lou et al., 2024) objective to train DID. Concretely, we define an insertion score that models the probability of inserting any token at any position of a sequence at a given time interval, and derive a corresponding Denoising Insertion Score Entropy (DISE) training objective. The DISE objective is based on a ratio of subsequence counts in the clean data after and before an insertion, which serves as the training target for the insertion score. To efficiently compute the ratio, we develop a parallelized dynamic programming algorithm that exploits data parallelism. Moreover, we demonstrate that under the fixed-length setting of MDLMs, the DISE objective can be further simplified to a form resembling cross-entropy, further improving parameterization and learning of the insertion score.

Comprehensive experiments demonstrate the effectiveness of DID in enhancing efficiency and flexibility. In fixed-length language modeling benchmarks, DID achieves superior performance compared to strong MDLM baselines (e.g., RADD (Ou et al., 2025)) when aligned by computational budget (FLOPs) (Tab. 1). Compared to MDLM baselines, DID could accelerate training by up to $1.99\times$ and $3.42\times$ (Tab. 3, 5) and inference by up to $1.58\times$ and $3.79\times$ (Tab. 2, 4), for models trained on fixed-length and variable-length datasets, respectively. Moreover, DID shows strong performance in variable-length settings, outperforming MDLMs and existing insertion-based LMs in sampling quality and consistency with data length distribution (Tab. 4, Fig. 2).

In summary, our main contributions are as follows:

- We propose DID, a novel diffusion LM based on deletion-insertion processes that eliminates the use of `<MASK>` and `<PAD>` tokens, improving the computational efficiency and generation flexibility of DLMs.
- We develop DISE, a score-based training objective, and an efficient parallel dynamic programming implementation that enables effective learning of DID's insertion process for language generation.
- Our experiments demonstrate the superior efficiency and flexibility of DID over baselines of MDLMs and other insertion-based LMs on language modeling, generation quality, generation length modeling, and training/inference speed.

## 2 PRELIMINARIES AND RELATED WORKS

### 2.1 CONTINUOUS-TIME DISCRETE DIFFUSION

A continuous-time discrete diffusion model consists of a forward noising process and a backward denoising process, both continuous-time Markov chains. In the forward process, the training samples

are progressively corrupted into pure noise. The model aims to learn its backward process that inverts this corruption, and generate new samples by sampling through the backward process from noise.

**Continuous-time Markov chain.** We consider a discrete state space $\mathcal{X}$. A continuous-time Markov chain (CTMC) is a process $\boldsymbol{x}_t$ on $\mathcal{X}$ with $t \in [0, 1]$, starting from an initial data distribution $p_0(\boldsymbol{x}_0)$. A CTMC is characterized by a time-dependent transition rate matrix $Q_t \in \mathbb{R}^{|\mathcal{X}| \times |\mathcal{X}|}$. For distinct states $\boldsymbol{x}_t, \boldsymbol{y} \in \mathcal{X}$, $Q_t(\boldsymbol{x}_t, \boldsymbol{y}) \geq 0$ defines the instantaneous transition rate from $\boldsymbol{x}_t$ to $\boldsymbol{y}$. This means, at an infinitesimal time interval $[t, t + \Delta t]$, the transition probability is given by:

$$p_{t+\Delta t|t}(\boldsymbol{y}|\boldsymbol{x}_t) = \delta(\boldsymbol{x}_t, \boldsymbol{y}) + Q_t(\boldsymbol{x}_t, \boldsymbol{y})\Delta t, \tag{1}$$

where $\delta$ is the Kronecker delta. In other words, the evolution of the marginal distribution $p_t \in \Delta_{|\mathcal{X}|}$ follows the Kolmogorov forward equation $\frac{dp_t}{dt} = p_t Q_t$. Note that the diagonal entries of $Q_t$ should satisfy $Q_t(\boldsymbol{x}_t, \boldsymbol{x}_t) = -\sum_{\boldsymbol{x}_t \neq \boldsymbol{y}} Q_t(\boldsymbol{x}_t, \boldsymbol{y})$ to ensure the weights of $p_t$ add up to 1.

**Determining the forward process.** In the forward process, a common parameterization (Campbell et al., 2022) of the transition rate matrix is $Q_t = \sigma(t)Q$, where $\sigma(t)$ is a scalar noise schedule and $Q$ is a constant rate matrix determined by the model design. Taking this to the Kolmogorov forward equation, one can analytically solve the marginal distributions by $p_t = p_s P_{t|s}$, where $P_{t|s} = \exp((\bar{\sigma}(t) - \bar{\sigma}(s))Q)$ is the transition probability matrix from time $s$ to time $t$, and $\bar{\sigma}(t) = \int_0^t \sigma(\tau)d\tau$.

**Learning the backward process.** It is known that the time reversal of this process is also a CTMC, with its infinitesimal transition probability similar to Eq.1:

$$p_{t-\Delta t|t}(\boldsymbol{y}|\boldsymbol{x}_t) = \delta(\boldsymbol{x}_t, \boldsymbol{y}) + \tilde{Q}_t(\boldsymbol{x}_t, \boldsymbol{y})\Delta t. \tag{2}$$

The reverse transition rate matrix $\tilde{Q}_t$ is associated to its forward counterpart $Q_t$ by the identity $\tilde{Q}_t(\boldsymbol{x}_t, \boldsymbol{y}) = Q_t(\boldsymbol{y}, \boldsymbol{x}_t)s(\boldsymbol{x}_t, t)_{\boldsymbol{y}}$ for $\boldsymbol{x}_t \neq \boldsymbol{y}$, and $\tilde{Q}_t(\boldsymbol{x}_t, \boldsymbol{x}_t) = -\sum_{\boldsymbol{x}_t \neq \boldsymbol{y}} \tilde{Q}_t(\boldsymbol{x}_t, \boldsymbol{y})$ (Lou et al., 2024). Here, $s(\boldsymbol{x}_t, t)_{\boldsymbol{y}} = p_t(\boldsymbol{y})/p_t(\boldsymbol{x}_t)$ is the **concrete score**, which is generally intractable and commonly approximated by a parameterized network $s_\theta(\boldsymbol{x}_t, t)_{\boldsymbol{y}}$ trained with the Denoising Score Entropy (DSE) objective (Lou et al., 2024), a negative evidence lower bound (ELBO) for discrete diffusion models (details in Appendix D.1):

$$\mathcal{L}_\theta^{\text{DSE}}(\boldsymbol{x}_0) = \mathbb{E}_{t,\boldsymbol{x}_t} \sum_{\boldsymbol{y} \neq \boldsymbol{x}_t} Q_t(\boldsymbol{y}, \boldsymbol{x}_t) \left[ s_\theta(\boldsymbol{x}_t, t)_{\boldsymbol{y}} - \frac{p_{t|0}(\boldsymbol{y}|\boldsymbol{x}_0)}{p_{t|0}(\boldsymbol{x}_t|\boldsymbol{x}_0)} \log s_\theta(\boldsymbol{x}_t, t)_{\boldsymbol{y}} + K\left( \frac{p_{t|0}(\boldsymbol{y}|\boldsymbol{x}_0)}{p_{t|0}(\boldsymbol{x}_t|\boldsymbol{x}_0)} \right) \right], \tag{3}$$

where the expectation is taken over $t \sim \text{Unif}([0, 1])$ and $\boldsymbol{x}_t \sim p_{t|0}(\boldsymbol{x}_t|\boldsymbol{x}_0)$, and $K(a) = a(\log a - 1)$.

## 2.2 Insertion-Based Language Models

Classical insertion-based models, such as the Insertion Transformer (Stern et al., 2019), Levenshtein Transformer (Gu et al., 2019), and InsNet (Lu et al., 2022), pioneered sequence generation via iterative insertion or edit operations. These models demonstrate the potential of flexible decoding orders and parallel decoding. However, their training objectives are defined at the level of local edit policies, rather than arising from a probabilistic modeling of the global data distribution.

Recently, insertion operations have been integrated into discrete diffusion and flow matching. Flexible Masked Diffusion Models (FlexMDMs) (Kim et al., 2025) augment MDMs with a learned insertion expectation to insert additional <MASK> tokens during generation; this enables variable-length generation but still relies on the masking-unmasking paradigm. Edit Flows (Havasi et al., 2025) instead define a CTMC directly over variable-length sequences via insertion, deletion, and substitution edits. To make sequence-level flow matching tractable, Edit Flows augment the state space with auxiliary edit-path variables and estimate the training objective by Monte Carlo sampling of these paths. While this preserves theoretical correctness, it adds implementation complexity and an additional source of stochasticity beyond the randomness of the CTMC forward process. Insertion Language Models (ILMs) (Patel et al., 2025) draw inspiration from diffusion approaches and formulate a CTMC forward process that deletes tokens to learn a backward insertion process. However, ILMs are not diffusion models and cannot learn the true backward insertion process since their training objective is more heuristic rather than likelihood-bounded. They also suffer from several practical limitations; for instance, they can only insert one token per step and require another network to determine when to stop the generation.

DID bridges the gap between flexible insertion-based generation and rigorous discrete diffusion. Unlike classical insertion models and ILMs, DID is well-grounded in a continuous-time diffusion framework: deletion and insertion form the forward and backward CTMCs over the full variable-length sequence space, and the DISE objective is inherited from the DSE objective to enable likelihood-bounded training. Unlike Edit Flows, however, DID does not introduce auxiliary edit-path variables. Thanks to the independence structure of the deletion process, the forward transition probabilities admit a closed-form expression in terms of subsequence counts, and the subsequence count ratios that appear in DISE can be computed **exactly** via parallel dynamic programming. Therefore, DID avoids the additional variance and engineering overhead associated with sampling edit paths, while maintaining a principled, likelihood-bounded training objective and completely eliminating the use of computationally wasteful `<MASK>` and `<PAD>` tokens.

## 3 DID: DELETION-INSERTION DIFFUSION LANGUAGE MODELS

We propose DID to improve the efficiency and flexibility of diffusion language models. Instead of masking and unmasking in MDLMs, DID reconstructs the diffusion processes with deletion and insertion. In this section, we rigorously formulate the forward deletion process in Sec. 3.1, backward insertion process and sampling algorithm in Sec. 3.2, develop a score-based approach to train DID in Sec. 3.3, discuss efficient implementation supporting parallelism for DID training objectives in Sec. 3.4, and analyze the additional optimization for the fixed-length data setting considered by MDLMs in Sec. 3.5.

### 3.1 FORWARD PROCESS: DELETION

The forward process of DID is a CTMC on the state space $\cup_{d=0}^{\infty} \mathcal{V}^d$ that gradually shortens the sequence length by deleting tokens, thus equipping the model with variable-length ability. Similar to MDLMs, we define this forward process through independent token-level deletions with rate $\sigma(t)$. Specifically, at the token level, a token $v \in \mathcal{V}$ can be deleted (denoted by transition to an empty state $\varnothing$) with an infinitesimal rate $\sigma(t)$, or remain unchanged otherwise. Thus, the transition probability within infinitesimal time $\Delta t$ is:

$$p_{t+\Delta t|t}(v'|v) = \begin{cases} \sigma(t)\Delta t, & v' = \varnothing, \\ 1 - \sigma(t)\Delta t, & v' = v. \end{cases} \tag{4}$$

Based on this independent token-level process, the sequence-level transition probability between timesteps $s$ and $t$ with $0 < s < t < 1$ can be derived as:

$$p_{t|s}(\boldsymbol{x}_t|\boldsymbol{x}_s) = (1 - e^{-(\bar{\sigma}(t)-\bar{\sigma}(s))})^{|\boldsymbol{x}_s|-|\boldsymbol{x}_t|} e^{-(\bar{\sigma}(t)-\bar{\sigma}(s))|\boldsymbol{x}_t|} N(\boldsymbol{x}_t, \boldsymbol{x}_s). \tag{5}$$

Here, $|\boldsymbol{x}|$ denotes the length of the sequence $\boldsymbol{x}$, $\bar{\sigma}(t) = \int_0^t \sigma(\tau)d\tau$ is the integrated noise rate, and $N(\boldsymbol{x}_t, \boldsymbol{x}_s)$ is the number of occurrences of $\boldsymbol{x}_t$ as distinct subsequences in $\boldsymbol{x}_s$. [1] This number accounts for the multiplicity of all the possible independent deletion paths from $\boldsymbol{x}_s$ to $\boldsymbol{x}_t$; see Appendix D.2 for the proof of Eq. 5.

The infinitesimal transition of the forward process is captured by the sequence-level transition rate matrix $Q_t$. Due to token-wise independence, at most one deletion can occur within an infinitesimal time interval with non-negligible ($\Omega(\Delta t)$) probability. Thus, the rate $Q_t(\boldsymbol{y}, \boldsymbol{x}_t)$ is non-zero only when $\boldsymbol{y} = \boldsymbol{x}_t$ or $\boldsymbol{y} \succ_1 \boldsymbol{x}_t$, i.e. $\boldsymbol{x}_t$ is the result of deleting exactly one token from $\boldsymbol{y}$, and the rate for $\boldsymbol{y} \succ_1 \boldsymbol{x}_t$ is (details in Appendix D.3):

$$Q_t(\boldsymbol{y}, \boldsymbol{x}_t) = \lim_{\Delta t \to 0} \frac{p_{t+\Delta t|t}(\boldsymbol{x}_t|\boldsymbol{y})}{\Delta t} = \sigma(t)N(\boldsymbol{x}_t, \boldsymbol{y}). \tag{6}$$

Note that, in our implementation, we prepend an undeletable special token `<BOS>` at the beginning of each sequence, so the above derivations should exclude `<BOS>`. Therefore, the fully noised sequence is a single `<BOS>`, which also serves as the initial input token at the first generation step to represent an empty sequence.

---

[1] For example, $N(\text{<BOS> b a g}, \text{<BOS> b a b g b a g}) = 5$. The distinct subsequences are (highlighted in bold): **<BOS> b a** b **g** b a g, **<BOS> b a** b g b a **g**, **<BOS> b** a b g b **a g**, **<BOS>** b a **b** g b **a g**, **<BOS>** b a b g **b a g**.

## 3.2 BACKWARD PROCESS: INSERTION

As in Sec. 2.1, we aim to learn the time reversal of the forward process, a CTMC with rate matrix $\tilde{Q}_t(\boldsymbol{x}_t, \boldsymbol{y}) = Q_t(\boldsymbol{y}, \boldsymbol{x}_t) s(\boldsymbol{x}_t, t)_{\boldsymbol{y}}$, where $s$ is the **concrete score**. Since $Q_t$ involves single-token deletions, the backward process $\tilde{Q}_t$ only considers single-token insertions (i.e., $\boldsymbol{y} \succ_1 \boldsymbol{x}_t$).

Directly applying the concrete score learning approach used in MDLMs is impractical here. The reason is that given $\boldsymbol{x}_t$, the number of possible resulting states $\boldsymbol{y}$ is variable, because different insertions can lead to the same result.[2] Consequently, targeting the concrete score requires calculating the number of possible $\boldsymbol{y}$ values and enabling the model to produce a variable-shaped output representing the concrete score for each $\boldsymbol{y}$. These requirements collectively introduce significant complexity and implementation challenges.

As an alternative, we target the **insertion score** $\bar{s}$. We learn a score for every insertion, regardless of whether the resulting $\boldsymbol{y}$ are identical. First, we define an insertion action $(i, v)$ as inserting token $v$ **after**[3] the $i$-th position of $\boldsymbol{x}$, resulting in:

$$\text{Ins}(\boldsymbol{x}, i, v) = (\boldsymbol{x}_{\leq i}, v, \boldsymbol{x}_{>i}). \tag{7}$$

Then, we define the insertion score as:

$$\bar{s}(\boldsymbol{x}_t, t)[i, v] \stackrel{\text{def}}{=} \frac{\mathbb{E}_{\boldsymbol{x}_0}[(1 - e^{-\bar{\sigma}(t)})^{|\boldsymbol{x}_0|} N(\text{Ins}(\boldsymbol{x}_t, i, v), \boldsymbol{x}_0)]}{\mathbb{E}_{\boldsymbol{x}_0}[(1 - e^{-\bar{\sigma}(t)})^{|\boldsymbol{x}_0|} N(\boldsymbol{x}_t, \boldsymbol{x}_0)]}, \quad \forall (i, v) \in [0, |\boldsymbol{x}_t|)_{\mathbb{Z}} \times \mathcal{V}, \tag{8}$$

whose shape is $|\boldsymbol{x}_t| \times |\mathcal{V}|$, tractable for transformer-based models.

Since the CTMC dynamics are based on the concrete score, to sample with the insertion score, we show that the concrete score is a scaled average of insertion scores, and the reverse transition rate is a summation of the rate of insertion actions (details in Appendix D.4):

$$s(\boldsymbol{x}_t, t)_{\boldsymbol{y}} = \frac{e^{-\bar{\sigma}(t)}}{1 - e^{-\bar{\sigma}(t)}} \frac{1}{N(\boldsymbol{x}_t, \boldsymbol{y})} \sum_{i \in I(\boldsymbol{x}_t, \boldsymbol{y})} \bar{s}(\boldsymbol{x}_t, t)[i, v(\boldsymbol{x}_t, \boldsymbol{y})], \tag{9}$$

$$\tilde{Q}_t(\boldsymbol{x}_t, \boldsymbol{y}) = \sum_{i \in I(\boldsymbol{x}_t, \boldsymbol{y})} \underbrace{\left( \frac{\sigma(t) e^{-\bar{\sigma}(t)}}{1 - e^{-\bar{\sigma}(t)}} \bar{s}(\boldsymbol{x}_t, t)[i, v(\boldsymbol{x}_t, \boldsymbol{y})] \right)}_{\text{Rate of action } (i, v(\boldsymbol{x}_t, \boldsymbol{y}))}, \tag{10}$$

where $I(\boldsymbol{x}_t, \boldsymbol{y})$ is the set of viable insertion positions to insert token $v(\boldsymbol{x}_t, \boldsymbol{y})$ from $\boldsymbol{x}_t$ to $\boldsymbol{y}$.[4]

Based on Eq. 10, we can transform the concrete score-based sampling in Eq. 2 into an equivalent insertion score-based sampling without computing $N(\boldsymbol{x}_t, \boldsymbol{y})$, $I(\boldsymbol{x}_t, \boldsymbol{y})$, or $v(\boldsymbol{x}_t, \boldsymbol{y})$ (details in Appendix D.5):

$$p^\theta_{t-\Delta t|t}((i, v)|\boldsymbol{x}_t) = \begin{cases} \frac{\sigma(t) e^{-\bar{\sigma}(t)}}{1 - e^{-\bar{\sigma}(t)}} \bar{s}_\theta(\boldsymbol{x}_t, t)[i, v] \Delta t, & v \neq \varnothing, \\ 1 - \sum_{w \neq \varnothing} p^\theta_{t-\Delta t|t}((i, w)|\boldsymbol{x}_t), & v = \varnothing, \end{cases} \tag{11}$$

where $\varnothing$ indicates no insertion, and $\tau$-leaping (Gillespie, 2001), a very popular approximate simulation method, could be adopted to sample all insertions simultaneously for parallel decoding.

## 3.3 TRAINING OBJECTIVE: DENOISING INSERTION SCORE ENTROPY

We aim to train the insertion score $\bar{s}_\theta$ using the DSE objective (Eq. 3) with the derived transition rate $Q(\boldsymbol{y}, \boldsymbol{x}_t)$ (Eq. 6), conditional distribution $p_{t|s}(\boldsymbol{x}_t | \boldsymbol{x}_s)$ (Eq. 5), and parameterized concrete score $s_\theta(\boldsymbol{x}_t, t)_{\boldsymbol{y}}$ (Eq. 9) for DID. Here, directly substituting $s_\theta$ (Eq. 9) into DSE is challenging. Since $s_\theta$ involves a summation of insertion scores $\bar{s}_\theta$, it results in an intractable log-sum structure within the DSE objective (Appendix D.6), we apply Jensen's inequality to derive a tractable variational upper bound, the Denoising Insertion Score Entropy (DISE).

---

[2]For example, inserting 'a' after the 1st or 2nd index of '<BOS> b a g' both yield '<BOS> b a a g'.

[3]We use a prepended, non-deletable <BOS> token (index 0) for insertions at the start.

[4]For example, if $\boldsymbol{x}_t = $ <BOS> b a g and $\boldsymbol{y} = $ <BOS> b a a g, then $\text{Ins}(\boldsymbol{x}_t, 1, a) = \text{Ins}(\boldsymbol{x}_t, 2, a) = \boldsymbol{y}$, $v(\boldsymbol{x}_t, \boldsymbol{y}) = $ a, $I(\boldsymbol{x}_t, \boldsymbol{y}) = \{1, 2\}$, and $N(\boldsymbol{x}_t, \boldsymbol{y}) = |I(\boldsymbol{x}_t, \boldsymbol{y})| = 2$.

**Proposition 1** (Denoising Insertion Score Entropy (DISE)). *The DSE objective for the deletion-insertion process is upper bounded by the DISE objective, $\mathcal{L}_\theta^{\text{DSE}}(\boldsymbol{x}_0) \leq \mathcal{L}_\theta^{\text{DISE}}(\boldsymbol{x}_0)$, which is:*

$$\mathcal{L}_\theta^{\text{DISE}}(\boldsymbol{x}_0) = \mathbb{E}_{t,\boldsymbol{x}_t} \left\{ \frac{\sigma(t)e^{-\bar{\sigma}(t)}}{1 - e^{-\bar{\sigma}(t)}} \sum_{i,v} \left[ \bar{s}_\theta(\boldsymbol{x}_t, t)[i,v] - \frac{N(\text{Ins}(\boldsymbol{x}_t, i, v), \boldsymbol{x}_0)}{N(\boldsymbol{x}_t, \boldsymbol{x}_0)} \log \bar{s}_\theta(\boldsymbol{x}_t, t)[i,v] + C \right] \right\}, \quad (12)$$

*where $C = K\left(\frac{N(\text{Ins}(\boldsymbol{x}_t, i, v), \boldsymbol{x}_0)}{N(\boldsymbol{x}_t, \boldsymbol{x}_0)}\right)$ is a $\theta$-free constant, $t \sim \text{Unif}([0, 1])$, and $\boldsymbol{x}_t \sim p_{t|0}(\boldsymbol{x}_t | \boldsymbol{x}_0)$.*

The proof (Appendix D.6) utilizes Jensen's inequality and a summation identity (Lemma 1) to transform the objective from state-based ($\boldsymbol{y}$) to action-based ($(i, v)$).

## 3.4 Efficient Parallel Dynamic Programming for Subsequence Counting

A fundamental challenge for the DISE objective (Eq. 12) is to efficiently solve the ratios of $N(\text{Ins}(\boldsymbol{x}_t, i, v), \boldsymbol{x}_0)$ and $N(\boldsymbol{x}_t, \boldsymbol{x}_0)$ for all possible insertions of $(i, v)$ on $\boldsymbol{x}_t$. Suppose the lengths of $\boldsymbol{x}_0$ and $\boldsymbol{x}_t$ are $m$ and $n$, solving a single subsequence counting problem of $N(\boldsymbol{x}_t, \boldsymbol{x}_0)$ has a well-known time complexity of $O(mn)$ through dynamic programming, performing it naively for $n \times V$ times would be prohibitive. Here, we show that $N(\text{Ins}(\boldsymbol{x}_t, i, v), \boldsymbol{x}_0)$ for all $(i, v)$ pairs could be efficiently solved based on the intermediate results of solving $N(\boldsymbol{x}_t, \boldsymbol{x}_0)$ just twice (via a prefix DP in Eq. 14 and a suffix DP in Eq. 15), reducing the time complexity to compute all ratios from $O(mn^2V)$ to $O(mn)$, and making the training of DID practical.

**Counting $N(\boldsymbol{x}_t, \boldsymbol{x}_0)$.** Here we briefly introduce the classic prefix DP and suffix DP solutions to this problem. Using Python's slicing syntax, the base cases for the prefix and suffix empty sequences of $\boldsymbol{x}_t$ are initialized as 1 (other entries are initialized as 0):

$$N(\boldsymbol{x}_t[:0], \boldsymbol{x}_0[:j]) = N(\boldsymbol{x}_t[n:], \boldsymbol{x}_0[j:]) = 1, \quad \forall j \in \{0, ..., m\}. \quad (13)$$

The **prefix DP** iterates $j$ from 1 to $m$, and for each $j$, it iterates $i$ from 1 to $n$:

$$N(\boldsymbol{x}_t[:i], \boldsymbol{x}_0[:j]) = N(\boldsymbol{x}_t[:i], \boldsymbol{x}_0[:j-1]) + \delta(\boldsymbol{x}_t[i-1], \boldsymbol{x}_0[j-1]) \cdot N(\boldsymbol{x}_t[:i-1], \boldsymbol{x}_0[:j-1]). \quad (14)$$

The **suffix DP**, similarly, iterates $j$ from $m - 1$ to 0, and for each $j$, it iterates $i$ from $n - 1$ to 0:

$$N(\boldsymbol{x}_t[i:], \boldsymbol{x}_0[j:]) = N(\boldsymbol{x}_t[i:], \boldsymbol{x}_0[j+1:]) + \delta(\boldsymbol{x}_t[i], \boldsymbol{x}_0[j]) \cdot N(\boldsymbol{x}_t[i+1:], \boldsymbol{x}_0[j+1:]). \quad (15)$$

The time complexities are $O(mn)$ for both the prefix and suffix DPs, and both $N(\boldsymbol{x}_t[:n], \boldsymbol{x}_0[:m])$ from prefix DP and $N(\boldsymbol{x}_t[0:], \boldsymbol{x}_0[0:])$ from suffix DP are the result of $N(\boldsymbol{x}_t, \boldsymbol{x}_0)$. Notably, the DPs could be batched and parallelized along the $i$-dimension, thus supporting vectorization and parallel execution, and it only needs to sequentially loop over the $j$-dimension for $m$ times.

**Counting $N(\textbf{Ins}(\boldsymbol{x}_t, i, v), \boldsymbol{x}_0)$.** It could be efficiently solved with the intermediate results of prefix DP and suffix DP:

$$N(\text{Ins}(\boldsymbol{x}_t, i, v), \boldsymbol{x}_0) = \sum_{j=1}^{m} \left[ \underbrace{\delta(\boldsymbol{x}_0[j], v)}_{\text{index addition}} \cdot \underbrace{N(\boldsymbol{x}_t[:i], \boldsymbol{x}_0[:j-1])}_{\text{prefix DP result}} \cdot \underbrace{N(\boldsymbol{x}_t[i:], \boldsymbol{x}_0[j:])}_{\text{suffix DP result}} \right]. \quad (16)$$

Based on the structure of Eq. 16, results for all $(i, v)$ pairs can be solved in parallel with an element-wise multiplication of the prefix and suffix DP result matrices, followed by an index addition that could be efficiently implemented with a sparse tensor coalescence. Algorithm details and a PyTorch implementation are in Appendix E.

## 3.5 Simplified Model for Fixed-Length Setting

To facilitate a fair comparison with MDLMs on the widely-adopted fixed-length language modeling benchmarks (Tab. 1), and clearly isolate the superior FLOPs efficiency of DID, we develop a set of optimizations to enhance DID in the fixed-length setting. We show that when $|\boldsymbol{x}_0|$ is a constant, 1) the insertion score becomes time-independent as the time-dependent terms of $(1 - e^{-\bar{\sigma}(t)})^{|\boldsymbol{x}_0|}$ in Eq. 8 could be canceled out, which leads to 2) a sequence-level normalization property (details in Appendix D.7):

$$\sum_{i,v} \bar{s}(\boldsymbol{x}_t, t)[i,v] = |\boldsymbol{x}_0| - |\boldsymbol{x}_t|. \quad (17)$$

This benefits the parameterization and training of DID from two aspects. First, the time-independence means that the network does not require time $t$ as input in the fixed-length setting, thus the parameterization reduces to $\bar{s}_\theta(\boldsymbol{x}_t)$, saves the parameters for time embedding, and enables a cache mechanism similar to (Ou et al., 2025) if the sequence is not changed between steps. Second, the output of the insertion score network could be explicitly normalized with a summation of $|\boldsymbol{x}_0| - |\boldsymbol{x}_t|$ as in Eq. 17. Therefore, the summation term of outputs from the insertion score network in the DISE objective (Eq. 12) turns into a constant, giving rise to a simplified Denoising Insertion Cross Entropy (DICE) objective (details in Appendix D.8):

$$\mathcal{L}_\theta^{\text{DICE}}(\boldsymbol{x}_0) = \mathop{\mathbb{E}}_{t,\boldsymbol{x}_t} \left\{ \frac{\sigma(t)e^{-\bar{\sigma}(t)}}{1 - e^{-\bar{\sigma}(t)}} \sum_{i,v} \frac{N(\text{Ins}(\boldsymbol{x}_t, i, v), \boldsymbol{x}_0)}{N(\boldsymbol{x}_t, \boldsymbol{x}_0)} \left[ -\log \bar{s}_\theta(\boldsymbol{x}_t)[i, v] + C \right] \right\}, \qquad (18)$$

where $C = \log \frac{N(\text{Ins}(\boldsymbol{x}_t, i, v), \boldsymbol{x}_0)}{N(\boldsymbol{x}_t, \boldsymbol{x}_0)}$ is a $\theta$-free constant. DICE can be interpreted as a weighted cross-entropy loss between the predicted insertion scores and the ground truth of subsequence count ratios, hence the name.

## 4 EXPERIMENTS

We evaluate language modeling performance, sampling quality, and training/inference efficiency of DID in the fixed-length setting in Sec. 4.1 and the variable-length setting in Sec. 4.2. For more details, results, analyses, generation examples, and intermediate generation process, please refer to Appendix F, G, H.

### 4.1 DID FOR FIXED-LENGTH LANGUAGE MODELING

**Settings.** Following RADD (Ou et al., 2025), which serves as our baseline method of MDLM, we train DID of both small and medium sizes on the OpenWebText (OWT) dataset (Gokaslan & Cohen, 2019) with the DICE objective (Eq. 18). We adopt the GPT2 tokenizer (Radford et al., 2019), concatenate all sequences, and split them into fixed-length chunks of 1024 tokens, and the training batch size is 512. RADD-small and -medium are reproduced with their open-sourced model checkpoints (trained for 400K steps on OWT).

**Zero-shot language modeling perplexity.** We evaluate the zero-shot modeling perplexity on seven datasets in Tab. 1. Since DID eliminates the computational FLOPs for `<MASK>` and hence reduces FLOPs by approximately half compared to RADD for the same training steps, we compare DID under two training configurations: **DID-S** (Steps-aligned), trained for the same steps as RADD (400K) and **DID-F** (FLOPs-aligned), trained for double the steps (800K) to match the total computational budget. We observe that when aligned by training steps (DID-S), our model achieves performance comparable to RADD, despite utilizing only about half the computational FLOPs. When aligned by the total computational budget (DID-F), DID consistently outperforms RADD across the majority of datasets for both small and medium sizes. This demonstrates that the computational savings achieved by eliminating `<MASK>` tokens are effectively turned into improved modeling performance. We also provide an ablation study in Appendix G.3 for DID trained with DISE objective (Eq. 12), i.e., without the additional optimizations in DICE for fixed-length data introduced in Sec. 3.5. This results in a reduced performance than DICE-trained DID in Tab. 1, yet comparable to RADD.

**Generative performance.** We report generative perplexity (PPL), unigram entropy (diversity), and inference speed of direct sampling across different numbers of total denoising steps in Tab. 2. Compared with RADD, DID achieves significantly better generation quality (lower PPL) with fewer denoising steps. When more total denoising steps are used, RADD slightly outperforms DID, which is reasonable since RADD is naturally designed for the fixed-length setting. Nonetheless, DID could consistently outperform RADD in the variable-length setting, which is deferred to Tab. 4. Furthermore, DID consistently provides $\sim 1.5\times$ inference speedup. This improvement stems from the fact that the average sequence length during the iterative insertion process is shorter than the fixed-length process of RADD. We also provide nucleus sampling results in Appendix G.6, where DID achieves lower PPL and entropy compared with RADD, demonstrating a stronger annealing effect.

Table 1: Zero-shot language modeling perplexity. Results for diffusion models are perplexity upper bounds.

| Size | Method | WikiText | Lambada | Pubmed | AG News | LM1B | Arxiv | PTB |
|------|--------|----------|---------|--------|---------|------|-------|-----|
| Small | RADD | 38.27 | 51.82 | 56.99 | 73.18 | 72.99 | 85.95 | **108.79** |
| | DID-S | 38.72 | 49.10 | 55.02 | 76.02 | 74.04 | 82.41 | 115.37 |
| | DID-F | **36.91** | **48.00** | **52.89** | **71.48** | **72.04** | **78.38** | 111.60 |
| Medium | RADD | 28.44 | 44.10 | 41.06 | 48.96 | 60.32 | 66.28 | **81.05** |
| | DID-S | 29.19 | 41.94 | 40.84 | 52.53 | 59.88 | 63.95 | 91.87 |
| | DID-F | **28.35** | **41.00** | **38.71** | **48.84** | **58.05** | **61.77** | 87.09 |

Table 2: Generative perplexity (PPL, evaluated by GPT2 Large), unigram entropy, inference time (in seconds), speedup, and average generation length for fixed-length models under different total denoising steps.

| Method | Steps | 16 | 32 | 64 | 128 | 256 | 512 | 1024 |
|--------|-------|-----|-----|-----|-----|-----|-----|------|
| RADD | PPL | 284.78 | 155.01 | 111.56 | 95.10 | 87.56 | **84.00** | **84.05** |
| | Entropy | 8.35 | 8.26 | 8.20 | 8.15 | 8.11 | 8.10 | 8.09 |
| | Time (s) | 0.220 | 0.317 | 0.499 | 0.879 | 1.644 | 2.882 | 4.512 |
| DID | PPL | **158.93** | **110.06** | **97.32** | **91.25** | **86.98** | 86.04 | 85.35 |
| | Entropy | 8.15 | 8.13 | 8.13 | 8.12 | 8.09 | 8.08 | 8.09 |
| | Time (s) | **0.169** | **0.246** | **0.353** | **0.573** | **1.047** | **1.826** | **3.006** |
| | Speedup | 1.30× | 1.29× | 1.41× | 1.53× | 1.57× | 1.58× | 1.50× |
| | Length | 1023.29 | 1024.01 | 1024.18 | 1024.07 | 1023.91 | 1024.03 | 1024.10 |

**Training speed.** [5] We report the training speeds for different model sizes in Tab. 3 to verify the efficiency gains of DID. DID demonstrates substantial speedups, increasing from 1.89× to 1.99× as we scale up the model. Empirically, removing the computations for `<MASK>` tokens significantly boosts training efficiency.

While the reduction in FLOPs suggests a theoretical 2× speedup, the actual gains in Tab. 2 are more modest. This discrepancy, as discussed in (Zheng et al., 2025), arises because inference is not a purely FLOPs-bound task. For training, the observed gains in Tab. 3 are also slightly lower due to additional overheads such as the DP algorithm for loss implementation (Sec. 3.4), which is a constant overhead

Table 3: Average training time (in seconds) per 50 steps (i.e. batches) on OpenWebText.

| | Small | Medium | Large |
|---|-------|--------|-------|
| RADD | 26.46 | 53.17 | 92.90 |
| DID | **14.03** | **27.77** | **46.60** |
| Speedup | 1.89× | 1.91× | 1.99× |

independent of the model size, thus the speedup could be scaled up with model size. Besides, a less developed system-level support for variable-length data also constrains the speedups.

## 4.2 DID FOR VARIABLE-LENGTH LANGUAGE MODELING

**Settings.** Following ILM (Patel et al., 2025), an insertion-based approach, we train DID-small on the Stories dataset (Eldan & Li, 2023; Mostafazadeh et al., 2016) for 60K steps with batch size 512, utilizing its variable-length sequences (average length 213.43 under the Bert-base-uncased tokenizer (Devlin et al., 2018)) truncated to a maximum length of 1024 and without padding. We also train RADD-small on Stories for the same steps (> 2× FLOPs of DID), details in Appendix F. Since RADD requires fixed-length inputs, we pad all sequences to a length of 1024, which is in line with the setting for MDLM in ILM experiments (Patel et al., 2025). As a result, RADD generates fixed-length outputs containing `<PAD>` tokens, which we subsequently remove to obtain the final variable-length outputs. ILM is reproduced with its open-sourced checkpoints.

---

[5] Large models are not fully trained; only their training speeds are measured.

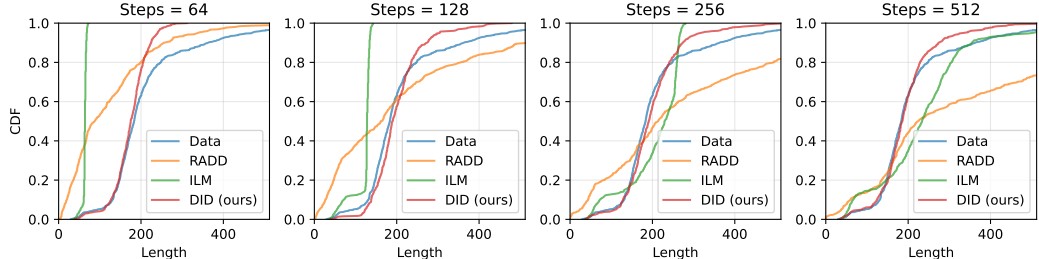

Figure 2: Cumulative distribution functions of generation length under different total denoising steps.

**Generative performance.** We report the generation quality and speed of direct sampling for variable-length models in Tab. 4. Compared with both baselines, DID maintains a significantly lower generative PPL (evaluated by GPT2 Large) with relatively high diversity and is more stable across different numbers of steps. This is important because strong sensitivity and non-convergence to the manually predefined number of generation steps are undesirable.

Regarding inference speed, DID achieves a speedup of up to 3.79× compared to RADD, due to the savings of `<MASK>` and `<PAD>` tokens for both model calling and distribution sampling. On the other hand, ILM achieves the fastest speed despite its lowest quality. We credit this to its over-simplified generation mechanism, which samples a token at exactly one *designated* position per step. In contrast, DID and RADD sample from all possible token posi-

Table 4: Generative PPL, unigram entropy, inference time (in seconds), and average generation length for variable-length models under different denoising steps. *: as outliers significantly affect PPL, only samples with PPL < 300 are counted, †: speedup over RADD.

| Method | Steps | 64 | 128 | 256 | 512 |
|---|---|---|---|---|---|
| ILM | PPL* | 161.80 | 137.64 | 42.29 | 31.14 |
| | Entropy | 5.20 | 5.65 | 5.97 | 6.01 |
| | Time (s) | **0.016** | **0.034** | **0.087** | **0.271** |
| | Length | 63.34 | 120.77 | 206.44 | 234.44 |
| RADD | PPL* | 81.92 | 50.89 | 34.47 | 26.78 |
| | Entropy | 5.22 | 5.58 | 5.79 | 5.85 |
| | Time (s) | 0.246 | 0.441 | 0.827 | 1.461 |
| | Length | 110.66 | 200.73 | 349.54 | 353.47 |
| DID | PPL | **22.78** | **21.07** | **21.90** | **23.88** |
| | Entropy | 5.90 | 5.94 | 5.94 | 5.94 |
| | Time (s) | 0.090 | 0.132 | 0.218 | 0.388 |
| | Speedup† | 2.73× | 3.34× | 3.79× | 3.76× |
| | Length | 182.31 | 193.77 | 202.97 | 204.96 |

tions to enable parallel decoding. ILM benefits considerably from this simplification, since categorical sampling is a major bottleneck in small models (Zheng et al., 2025). Moreover, ILM is limited to generating text shorter than the total steps (see Tab. 4), which also contributes to its faster sampling.

We also provide nucleus sampling results for variable-length models in Appendix G.6, and ablation studies of different padding lengths for RADD in Appendix G.8, addressing potential concerns that the 1024 padding length for RADD might be too long or unfair. When trained at a length of 512, RADD exhibits observable degradation and remains $\sim 1.59\times$ slower than DID, further confirming the original setting of ILM.

**Length modeling.** Besides, DID demonstrates superior length modeling capabilities, exhibiting consistency between the generation length distribution and the training data length distribution. This is demonstrated in Fig. 2, where the CDF of the length distribution of DID is closely aligned with the dataset compared to the baselines. DID's average generation length reported in Tab. 4 is also stable and approximating the ground truth distribution.

Table 5: Average training time (in seconds) per 50 steps on Stories.

| | Small | Medium | Large |
|---|---|---|---|
| RADD | 19.93 | 37.87 | 67.75 |
| DID | **7.71** | **12.30** | **19.83** |
| Speedup | 2.58× | 3.08× | 3.42× |

**Training speed.** [6] We also compare the training speed on the Stories dataset (Tab. 5). The efficiency gains of DID are even more pronounced in the variable-length setting, reaching up to 3.42× speedup for large models. This is because RADD suffers significant overhead from processing `<PAD>` tokens (since the average length 213.43 is much shorter than the padded length 1024), while DID benefits from the shorter length.

---

[6]Medium and large models are not fully trained; only their training speeds are measured.

Regarding more efficiency analyses, ablation studies, inference strategies, and performance analyses, please see Appendix G.

## 5 DISCUSSIONS

Here we discuss the theoretical insights for the advantages of DID over MDM, we summarize four structural advantages that come from using insertion instead of unmasking during generation.

**Context-aware decoding order.** In standard RADD sampling (e.g., $\tau$-leaping), the decision of which positions to update is governed solely by the noise schedule (random unmasking). The probability that any masked position $i$ is unmasked in a step is governed by a scalar function $\psi(t, s)$, which depends only on the noise schedule (e.g., $\frac{t-s}{t}$ for log-linear):

$$\mathbb{P}\left[\boldsymbol{x}_s^{(i)} \neq [\mathbf{M}] \mid \boldsymbol{x}_t\right] = \psi(t, s).$$

The probability is uniform across all masked positions and independent of the context $\boldsymbol{x}_t^{\mathrm{UM}}$. The learned model only determines what token to fill in if a position is unmasked; it does not influence where updates occur. When steps are few, this forces the model to predict many tokens at random absolute positions simultaneously, often leading to incoherence.

In contrast, DID learns where to insert based on the existing content. The probability of an insertion after position $i$ in $\Delta t$ time is directly determined by the learned insertion score $\bar{s}_\theta(\boldsymbol{x}_t, t)$:

$$\mathbb{P}\left[\text{Insertion at position } i \mid \boldsymbol{x}_t\right] \approx \frac{\sigma(t)e^{-\bar{\sigma}(t)}}{1 - e^{-\bar{\sigma}(t)}} \sum_{v \in \mathcal{V}} \bar{s}_\theta(\boldsymbol{x}_t, t)[i, v]\Delta t.$$

Crucially, this mechanism is content-dependent. The model naturally chooses to generate tokens in positions where it is more probable first, improving the quality of each sampling step.

**Self-correction mechanism.** RADD relies on absolute positions. Once a token is unmasked, its content and position are fixed, so any early suboptimal predictions (common in fast sampling) lead to error accumulation. DID predicts relative order. Even if the model generates imperfect tokens early on, it can still refine the sentence structure in later steps by inserting new tokens between existing ones (see our demonstration in Appendix H.3). This dynamic refinement capability makes DID more robust to the errors inherent in few-step generation.

**Cautious early generation.** RADD works on the whole sequence length from the very first step, so it can unmask exceedingly many tokens at the earlier stage using inaccurate model predictions. In DID, we allow at most $n + 1$ insertions if the current context length is $n$. For example, the very first sampling step will insert at most 1 token. This means we perform fewer insertions at the earlier stage, acting more cautiously than RADD.

**Better handling of adjacent token dependence.** Both DID and MDM employ the $\tau$-leaping method for fast sampling, which assumes multiple token transitions occur independently at each step. This independence assumption is a major source of sampling error. Note that DID will not insert multiple adjacent tokens at once, so the transitions of adjacent tokens are never independent. Adjacent tokens are always inserted sequentially. In contrast, $\tau$-leaping in MDM poses a stronger assumption that any pair of tokens are mutually independent. It is fair to hypothesize that adjacent tokens bear more dependence than separated tokens — thus, we believe DID handles token interdependence better.

## 6 CONCLUSION

In this paper, we introduce DID to improve the computational efficiency and generation flexibility of diffusion language models by eliminating the use of `<MASK>` and `<PAD>` tokens in our paradigm. Theoretically, we formulate the diffusion processes for deletion and insertion, define an insertion score, derive and implement the corresponding training objectives and sampling algorithm for DID. We evaluated DID on modeling performance, generation quality, and training/inference speed, demonstrating the superiority of DID over the baselines of MDLM and existing insertion-based LMs in both fixed-length and variable-length settings. A discussion of the limitations and future works of this paper is provided in Appendix A.

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

CONTENTS

## A    LIMITATIONS AND FUTURE WORKS

This work presents the core framework of DID and, unlike the more established MDLMs, has not yet integrated many optimizations, such as advanced inference algorithms (Wu et al., 2025a; Wang et al., 2025), or hybrid models combining autoregressive approaches (Arriola et al., 2025; Sahoo et al., 2025). Since these optimizations are not inherently tied to a specific diffusion process, adapting them to DID represents a promising future direction. Second, although we have demonstrated the effectiveness, efficiency, and flexibility of DID, our models were trained at a relatively small scale due to resource constraints. As a result, their performance on larger and more complex tasks remains unexplored, and we leave scaling up DID to future work.

## B    THE USE OF LARGE LANGUAGE MODELS (LLMS)

Following ICLR guidelines, we wish to clarify our use of Large Language Models (LLMs) during the preparation of this work.

The research ideas, methodology, experimental design, and analysis presented in this paper were developed entirely by the human authors. LLMs were not involved in the ideation process.

We utilized LLMs as tools for editing and polishing the text, helping to improve the clarity and phrasing in various sections of the main paper and the appendix. The authors have reviewed the manuscript thoroughly and take full responsibility for its content.

## C    NOTATION SUMMARY

Table 6: Summary of Key Notations

| Notation | Description |
|---|---|
| $\mathcal{V}$ | Vocabulary. |
| $\mathcal{X} = \bigcup_{d=0}^{\infty} \mathcal{V}^d$ | Sequence state space (variable length). |
| $\boldsymbol{x}_0, \boldsymbol{x}_t, \boldsymbol{y}$ | Clean sequence; sequence at time $t$; another sequence state. |
| $\|\boldsymbol{x}\|$ | Length of $\boldsymbol{x}$. |
| <BOS>, <MASK>, <PAD> | Begin-of-sequence (non-deletable), absorbing mask, padding. |
| $\varnothing$ | Null token representing deletion or no insertion (not in $\mathcal{V}$). |
| $Q_t, \tilde{Q}_t$ | Forward and reverse sequence-level CTMC rate matrices. |
| $p_{t\|s}(\cdot\|\cdot)$ | Forward transition probability from $s$ to $t$. |
| $p_t(\cdot)$ | Marginal distribution at time $t$. |
| $\sigma(t), \bar{\sigma}(t)$ | Noise rate and its integral $\int_0^t \sigma(\tau)\,d\tau$. |
| $s(\boldsymbol{x}_t, t)_{\boldsymbol{y}}$ | Concrete score $p_t(\boldsymbol{y})/p_t(\boldsymbol{x}_t)$. |
| $N(\boldsymbol{x}, \boldsymbol{y})$ | The number of occurrences of $\boldsymbol{x}$ as a distinct subsequence of $\boldsymbol{y}$. |
| $\boldsymbol{y} \succ_1 \boldsymbol{x}$ | $\boldsymbol{y}$ is obtained by inserting exactly one token into $\boldsymbol{x}$. |
| $v(\boldsymbol{x}, \boldsymbol{y})$ | The unique inserted token when $\boldsymbol{y} \succ_1 \boldsymbol{x}$. |
| $\mathrm{Ins}(\boldsymbol{x}, i, v)$ | The result of inserting token $v \in \mathcal{V}$ *after* position $i$ of $\boldsymbol{x}$. |
| $I(\boldsymbol{x}, \boldsymbol{y})$ | Valid insertion indices s.t. $\mathrm{Ins}(\boldsymbol{x}, i, v(\boldsymbol{x}, \boldsymbol{y})) = \boldsymbol{y}$. |
| $\bar{s}(\boldsymbol{x}_t, t)[i, v]$ | Insertion score for insertion operation $(i, v)$ at time $t$. |
| $\bar{s}(\boldsymbol{x}_t)[i, v]$ | Time-independent insertion score (fixed-length setting). |
| $K(a)$ | Convex function $a(\log a - 1)$. |

## D    DETAILED PROOFS AND DERIVATIONS

### D.1    DERIVATION OF THE DENOISING SCORE ENTROPY LOSS (EQ.3)

The training objective for discrete diffusion models is derived by minimizing a variational upper bound on the negative log-likelihood (NLL) of the data, $-\log p_0^\theta(\boldsymbol{x}_0)$.

Let $\mathbb{P}_{\boldsymbol{x}_0}$ denote the path measure (the probability distribution over entire trajectories) of the true posterior reverse process conditioned on the data $\boldsymbol{x}_0$. Let $\mathbb{P}^\theta$ denote the path measure of the learned reverse process parameterized by $\theta$. By the data processing inequality, we have:

$$-\log p_0^\theta(\boldsymbol{x}_0) = D_{\mathrm{KL}}\left(\delta_{\boldsymbol{x}_0}\|p_0^\theta\right) \leq D_{\mathrm{KL}}(\mathbb{P}_{\boldsymbol{x}_0}\|\mathbb{P}^\theta). \tag{19}$$

We define the training objective as this variational upper bound: $\mathcal{L}(\theta) = D_{\mathrm{KL}}(\mathbb{P}_{\boldsymbol{x}_0}\|\mathbb{P}^\theta)$, assuming both processes share the same prior distribution at $t = 1$.

Both processes are Continuous-Time Markov Chains (CTMCs). Let $\widetilde{Q}_t^0(\boldsymbol{x}_t, \boldsymbol{y})$ denote the true conditional reverse transition rate (for $\mathbb{P}_{\boldsymbol{x}_0}$) and $\widetilde{Q}_t^\theta(\boldsymbol{x}_t, \boldsymbol{y})$ denote the parameterized reverse transition rate (for $\mathbb{P}^\theta$).

The KL divergence between path measures can be decomposed using the chain rule. If we consider a discrete-time approximation with infinitesimal step $\Delta t$, the total KL divergence is the sum of the expected KL divergences at each step. We analyze the KL divergence between the infinitesimal transition probabilities $p^0(\boldsymbol{y}|\boldsymbol{x}_t) = \delta(\boldsymbol{x}_t, \boldsymbol{y}) + \widetilde{Q}_t^0(\boldsymbol{x}_t, \boldsymbol{y})\Delta t + o(\Delta t)$ and $p^\theta(\boldsymbol{y}|\boldsymbol{x}_t) = \delta(\boldsymbol{x}_t, \boldsymbol{y}) + \widetilde{Q}_t^\theta(\boldsymbol{x}_t, \boldsymbol{y})\Delta t + o(\Delta t)$. The instantaneous KL divergence is (Opper & Sanguinetti, 2007):

$$D_{\mathrm{KL}}(p^0(\cdot|\boldsymbol{x}_t)\|p^\theta(\cdot|\boldsymbol{x}_t)) = \Delta t \sum_{\boldsymbol{y} \neq \boldsymbol{x}_t} \left( \widetilde{Q}_t^0(\boldsymbol{x}_t, \boldsymbol{y}) \log \frac{\widetilde{Q}_t^0(\boldsymbol{x}_t, \boldsymbol{y})}{\widetilde{Q}_t^\theta(\boldsymbol{x}_t, \boldsymbol{y})} + \widetilde{Q}_t^\theta(\boldsymbol{x}_t, \boldsymbol{y}) - \widetilde{Q}_t^0(\boldsymbol{x}_t, \boldsymbol{y}) \right) + o(\Delta t). \tag{20}$$

Summing these contributions and taking the continuous limit ($\Delta t \to 0$) yields the integral form for the path KL divergence:

$$\mathcal{L}(\theta) = \int_0^1 \mathbb{E}_{\boldsymbol{x}_t \sim p_{t|0}} \left[ \sum_{\boldsymbol{y} \neq \boldsymbol{x}_t} \left( \widetilde{Q}_t^0(\boldsymbol{x}_t, \boldsymbol{y}) \log \frac{\widetilde{Q}_t^0(\boldsymbol{x}_t, \boldsymbol{y})}{\widetilde{Q}_t^\theta(\boldsymbol{x}_t, \boldsymbol{y})} + \widetilde{Q}_t^\theta(\boldsymbol{x}_t, \boldsymbol{y}) - \widetilde{Q}_t^0(\boldsymbol{x}_t, \boldsymbol{y}) \right) \right] dt. \tag{21}$$

We now substitute the specific definitions of these transition rates. The true conditional reverse rate $\widetilde{Q}_t^0$ is related to the forward rate $Q_t(\boldsymbol{y}, \boldsymbol{x}_t)$ by

$$\widetilde{Q}_t^0(\boldsymbol{x}_t, \boldsymbol{y}) = Q_t(\boldsymbol{y}, \boldsymbol{x}_t) \frac{p_{t|0}(\boldsymbol{y}|\boldsymbol{x}_0)}{p_{t|0}(\boldsymbol{x}_t|\boldsymbol{x}_0)}. \tag{22}$$

The parameterized reverse rate $\widetilde{Q}_t^\theta$ is defined using the score network $s_\theta(\boldsymbol{x}_t, t)_{\boldsymbol{y}}$:

$$\widetilde{Q}_t^\theta(\boldsymbol{x}_t, \boldsymbol{y}) = Q_t(\boldsymbol{y}, \boldsymbol{x}_t) s_\theta(\boldsymbol{x}_t, t)_{\boldsymbol{y}}. \tag{23}$$

We substitute these rates (Eq.22 and Eq.23) into the expression inside the summation in Eq.21. The expression becomes:

$$\left( Q_t(\boldsymbol{y}, \boldsymbol{x}_t) \frac{p_{t|0}(\boldsymbol{y}|\boldsymbol{x}_0)}{p_{t|0}(\boldsymbol{x}_t|\boldsymbol{x}_0)} \right) \log \frac{Q_t(\boldsymbol{y}, \boldsymbol{x}_t) \frac{p_{t|0}(\boldsymbol{y}|\boldsymbol{x}_0)}{p_{t|0}(\boldsymbol{x}_t|\boldsymbol{x}_0)}}{Q_t(\boldsymbol{y}, \boldsymbol{x}_t) s_\theta(\boldsymbol{x}_t, t)_{\boldsymbol{y}}} + Q_t(\boldsymbol{y}, \boldsymbol{x}_t) s_\theta(\boldsymbol{x}_t, t)_{\boldsymbol{y}} - Q_t(\boldsymbol{y}, \boldsymbol{x}_t) \frac{p_{t|0}(\boldsymbol{y}|\boldsymbol{x}_0)}{p_{t|0}(\boldsymbol{x}_t|\boldsymbol{x}_0)}. \tag{24}$$

We simplify by canceling $Q_t(\boldsymbol{y}, \boldsymbol{x}_t)$ inside the logarithm and factoring it out from the entire expression:

$$= Q_t(\boldsymbol{y}, \boldsymbol{x}_t) \left[ \frac{p_{t|0}(\boldsymbol{y}|\boldsymbol{x}_0)}{p_{t|0}(\boldsymbol{x}_t|\boldsymbol{x}_0)} \log \frac{\frac{p_{t|0}(\boldsymbol{y}|\boldsymbol{x}_0)}{p_{t|0}(\boldsymbol{x}_t|\boldsymbol{x}_0)}}{s_\theta(\boldsymbol{x}_t, t)_{\boldsymbol{y}}} + s_\theta(\boldsymbol{x}_t, t)_{\boldsymbol{y}} - \frac{p_{t|0}(\boldsymbol{y}|\boldsymbol{x}_0)}{p_{t|0}(\boldsymbol{x}_t|\boldsymbol{x}_0)} \right]. \tag{25}$$

We expand the logarithm ($\log(A/B) = \log A - \log B$) and rearrange the terms:

$$= Q_t(\boldsymbol{y}, \boldsymbol{x}_t) \left[ \frac{p_{t|0}(\boldsymbol{y}|\boldsymbol{x}_0)}{p_{t|0}(\boldsymbol{x}_t|\boldsymbol{x}_0)} \left( \log \frac{p_{t|0}(\boldsymbol{y}|\boldsymbol{x}_0)}{p_{t|0}(\boldsymbol{x}_t|\boldsymbol{x}_0)} - \log s_\theta(\boldsymbol{x}_t, t)_{\boldsymbol{y}} \right) + s_\theta(\boldsymbol{x}_t, t)_{\boldsymbol{y}} - \frac{p_{t|0}(\boldsymbol{y}|\boldsymbol{x}_0)}{p_{t|0}(\boldsymbol{x}_t|\boldsymbol{x}_0)} \right] \tag{26}$$

$$= Q_t(\boldsymbol{y}, \boldsymbol{x}_t) \left[ s_\theta(\boldsymbol{x}_t, t)_{\boldsymbol{y}} - \frac{p_{t|0}(\boldsymbol{y}|\boldsymbol{x}_0)}{p_{t|0}(\boldsymbol{x}_t|\boldsymbol{x}_0)} \log s_\theta(\boldsymbol{x}_t, t)_{\boldsymbol{y}} + \frac{p_{t|0}(\boldsymbol{y}|\boldsymbol{x}_0)}{p_{t|0}(\boldsymbol{x}_t|\boldsymbol{x}_0)} \left( \log \frac{p_{t|0}(\boldsymbol{y}|\boldsymbol{x}_0)}{p_{t|0}(\boldsymbol{x}_t|\boldsymbol{x}_0)} - 1 \right) \right]. \tag{27}$$

Letting $K(a) = a(\log a - 1)$. Recognizing that the time integral $\int_0^1 dt$ combined with the expectation over $\boldsymbol{x}_t \sim p_{t|0}$ is equivalent to the expectation over uniform time $t \sim U(0, 1)$ and the corresponding conditional $\boldsymbol{x}_t$, the objective $\mathcal{L}(\theta)$ is exactly the DSE loss (Eq.3):

$$\mathcal{L}_\theta^{\text{DSE}}(\boldsymbol{x}_0) = \mathop{\mathbb{E}}_{t, \boldsymbol{x}_t} \sum_{\boldsymbol{y} \neq \boldsymbol{x}_t} Q_t(\boldsymbol{y}, \boldsymbol{x}_t) \left[ s_\theta(\boldsymbol{x}_t, t)_{\boldsymbol{y}} - \frac{p_{t|0}(\boldsymbol{y}|\boldsymbol{x}_0)}{p_{t|0}(\boldsymbol{x}_t|\boldsymbol{x}_0)} \log s_\theta(\boldsymbol{x}_t, t)_{\boldsymbol{y}} + K\left( \frac{p_{t|0}(\boldsymbol{y}|\boldsymbol{x}_0)}{p_{t|0}(\boldsymbol{x}_t|\boldsymbol{x}_0)} \right) \right].$$

(28)

### D.2 DERIVATION OF THE SEQUENCE-LEVEL TRANSITION PROBABILITY (EQ.5)

We derive the sequence-level transition probability $p_{t|s}(\boldsymbol{x}_t|\boldsymbol{x}_s)$ based on the definition of the DID forward process as an independent token-level deletion process. We begin by analyzing the dynamics of a single token.

The token-level process (Eq.4) is a Continuous-Time Markov Chain (CTMC) on the state space $\{v, \varnothing\}$, where $v \in \mathcal{V}$ denotes the presence of a token and $\varnothing$ denotes the deleted state. The transition rate matrix $Q_t^{\text{tok}}$ at time $t$, indexed by $(v, \varnothing)$, is defined by the deletion rate $\sigma(t)$:

$$Q_t^{\text{tok}} = \begin{pmatrix} -\sigma(t) & \sigma(t) \\ 0 & 0 \end{pmatrix}.$$

(29)

Let $P_v(\tau)$ be the probability that the token is in state $v$ at time $\tau \in [s, t]$, given it started in state $v$ at time $s$ ($P_v(s) = 1$). The evolution of this probability follows the Kolmogorov forward equation:

$$\frac{dP_v(\tau)}{d\tau} = P_v(\tau)Q_\tau^{\text{tok}}(v, v) + P_\varnothing(\tau)Q_\tau^{\text{tok}}(\varnothing, v) = -\sigma(\tau)P_v(\tau).$$

(30)

Solving this first-order ordinary differential equation by integrating from $s$ to $t$:

$$\int_s^t \frac{dP_v(\tau)}{P_v(\tau)} = \int_s^t -\sigma(\tau)d\tau \implies \ln(P_v(t)) - \ln(P_v(s)) = -(\bar{\sigma}(t) - \bar{\sigma}(s)).$$

(31)

Thus, the probability that a single token survives during the interval $[s, t]$ is $P_v(t) = e^{-(\bar{\sigma}(t) - \bar{\sigma}(s))}$. Conversely, the probability of deletion is $1 - e^{-(\bar{\sigma}(t) - \bar{\sigma}(s))}$.

We now consider the sequence-level transition from $\boldsymbol{x}_s$ to $\boldsymbol{x}_t$. Let $\Delta\bar{\sigma} = \bar{\sigma}(t) - \bar{\sigma}(s)$. A transition occurs if the tokens forming $\boldsymbol{x}_t$ survive and the remaining $|\boldsymbol{x}_s| - |\boldsymbol{x}_t|$ tokens are deleted. Since token deletions are independent, the probability of a specific path (a specific occurrence of $\boldsymbol{x}_t$ in $\boldsymbol{x}_s$) is $(e^{-\Delta\bar{\sigma}})^{|\boldsymbol{x}_t|} \times (1 - e^{-\Delta\bar{\sigma}})^{|\boldsymbol{x}_s| - |\boldsymbol{x}_t|}$.

The total transition probability $p_{t|s}(\boldsymbol{x}_t|\boldsymbol{x}_s)$ is the sum over all distinct paths, counted by the subsequence count $N(\boldsymbol{x}_t, \boldsymbol{x}_s)$. Therefore, we obtain:

$$p_{t|s}(\boldsymbol{x}_t|\boldsymbol{x}_s) = N(\boldsymbol{x}_t, \boldsymbol{x}_s)(1 - e^{-(\bar{\sigma}(t) - \bar{\sigma}(s))})^{|\boldsymbol{x}_s| - |\boldsymbol{x}_t|} e^{-(\bar{\sigma}(t) - \bar{\sigma}(s))|\boldsymbol{x}_t|}.$$

(32)

### D.3 DERIVATION OF THE TRANSITION RATE (EQ.6)

We derive the transition rate $Q_t(\boldsymbol{y}, \boldsymbol{x}_t)$ from a sequence $\boldsymbol{y}$ to a sequence $\boldsymbol{x}_t$, where $\boldsymbol{x}_t$ is obtained from $\boldsymbol{y}$ by deleting a single token (denoted as $\boldsymbol{y} \succ_1 \boldsymbol{x}_t$). This implies $|\boldsymbol{y}| = |\boldsymbol{x}_t| + 1$.

$$Q_t(\boldsymbol{y}, \boldsymbol{x}_t) \triangleq \lim_{\Delta t \to 0} \frac{p_{t+\Delta t|t}(\boldsymbol{x}_t|\boldsymbol{y})}{\Delta t} \tag{33}$$

$$= \lim_{\Delta t \to 0} \frac{(1 - e^{-(\bar{\sigma}(t+\Delta t) - \bar{\sigma}(t))})^{|\boldsymbol{y}| - |\boldsymbol{x}_t|} e^{-(\bar{\sigma}(t+\Delta t) - \bar{\sigma}(t))|\boldsymbol{x}_t|} N(\boldsymbol{x}_t, \boldsymbol{y})}{\Delta t} \tag{34}$$

$$= \lim_{\Delta t \to 0} \frac{(1 - e^{-\sigma(t)\Delta t + o(\Delta t)})^1 \cdot e^{-(\sigma(t)\Delta t + o(\Delta t))|\boldsymbol{x}_t|} \cdot N(\boldsymbol{x}_t, \boldsymbol{y})}{\Delta t} \tag{35}$$

$$= \lim_{\Delta t \to 0} \frac{(\sigma(t)\Delta t + o(\Delta t)) \cdot (1 - |\boldsymbol{x}_t|\sigma(t)\Delta t + o(\Delta t)) \cdot N(\boldsymbol{x}_t, \boldsymbol{y})}{\Delta t} \tag{36}$$

$$= \lim_{\Delta t \to 0} \frac{\sigma(t)\Delta t + o(\Delta t)}{\Delta t} \cdot N(\boldsymbol{x}_t, \boldsymbol{y}) \tag{37}$$

$$= \sigma(t)N(\boldsymbol{x}_t, \boldsymbol{y}). \tag{38}$$

### D.4 DERIVATION OF THE RELATIONSHIP BETWEEN CONCRETE SCORE AND INSERTION SCORE (EQ. 9)

We aim to prove the identity stated in Eq. 9. This identity concerns the backward process where transitions occur such that $\boldsymbol{y} \succ_1 \boldsymbol{x}_t$. Note that this condition implies $|\boldsymbol{y}| = |\boldsymbol{x}_t| + 1$.

First, we derive the explicit form of the concrete score $s(\boldsymbol{x}_t, t)_{\boldsymbol{y}} = p_t(\boldsymbol{y})/p_t(\boldsymbol{x}_t)$ by expanding the marginal distributions using the forward transition probability (Eq. 5):

$$
\begin{aligned}
s(\boldsymbol{x}_t, t)_{\boldsymbol{y}} &= \frac{p_t(\boldsymbol{y})}{p_t(\boldsymbol{x}_t)} = \frac{\mathbb{E}_{\boldsymbol{x}_0}[p_{t|0}(\boldsymbol{y}|\boldsymbol{x}_0)]}{\mathbb{E}_{\boldsymbol{x}_0}[p_{t|0}(\boldsymbol{x}_t|\boldsymbol{x}_0)]} \\
&= \frac{\mathbb{E}_{\boldsymbol{x}_0}\left[(1 - e^{-\bar{\sigma}(t)})^{|\boldsymbol{x}_0| - |\boldsymbol{y}|} e^{-\bar{\sigma}(t)|\boldsymbol{y}|} N(\boldsymbol{y}, \boldsymbol{x}_0)\right]}{\mathbb{E}_{\boldsymbol{x}_0}\left[(1 - e^{-\bar{\sigma}(t)})^{|\boldsymbol{x}_0| - |\boldsymbol{x}_t|} e^{-\bar{\sigma}(t)|\boldsymbol{x}_t|} N(\boldsymbol{x}_t, \boldsymbol{x}_0)\right]} \\
&= \frac{e^{-\bar{\sigma}(t)|\boldsymbol{y}|}(1 - e^{-\bar{\sigma}(t)})^{-|\boldsymbol{y}|}}{e^{-\bar{\sigma}(t)|\boldsymbol{x}_t|}(1 - e^{-\bar{\sigma}(t)})^{-|\boldsymbol{x}_t|}} \frac{\mathbb{E}_{\boldsymbol{x}_0}\left[(1 - e^{-\bar{\sigma}(t)})^{|\boldsymbol{x}_0|} N(\boldsymbol{y}, \boldsymbol{x}_0)\right]}{\mathbb{E}_{\boldsymbol{x}_0}\left[(1 - e^{-\bar{\sigma}(t)})^{|\boldsymbol{x}_0|} N(\boldsymbol{x}_t, \boldsymbol{x}_0)\right]} \\
&\overset{|\boldsymbol{y}| = |\boldsymbol{x}_t| + 1}{=} \frac{e^{-\bar{\sigma}(t)}}{1 - e^{-\bar{\sigma}(t)}} \frac{\mathbb{E}_{\boldsymbol{x}_0}\left[(1 - e^{-\bar{\sigma}(t)})^{|\boldsymbol{x}_0|} N(\boldsymbol{y}, \boldsymbol{x}_0)\right]}{\mathbb{E}_{\boldsymbol{x}_0}\left[(1 - e^{-\bar{\sigma}(t)})^{|\boldsymbol{x}_0|} N(\boldsymbol{x}_t, \boldsymbol{x}_0)\right]}.
\end{aligned}
\tag{39}
$$

Next, we examine the right-hand side (RHS) of Eq. 9 and substitute the definition of the insertion score $\bar{s}$ (Eq. 8):

$$
\text{RHS} = \frac{e^{-\bar{\sigma}(t)}}{1 - e^{-\bar{\sigma}(t)}} \frac{1}{N(\boldsymbol{x}_t, \boldsymbol{y})} \sum_{i \in I(\boldsymbol{x}_t, \boldsymbol{y})} \bar{s}(\boldsymbol{x}_t, t)[i, v(\boldsymbol{x}_t, \boldsymbol{y})]
\tag{40}
$$

$$
= \frac{e^{-\bar{\sigma}(t)}}{1 - e^{-\bar{\sigma}(t)}} \frac{1}{N(\boldsymbol{x}_t, \boldsymbol{y})} \sum_{i \in I(\boldsymbol{x}_t, \boldsymbol{y})} \left( \frac{\mathbb{E}_{\boldsymbol{x}_0}[(1 - e^{-\bar{\sigma}(t)})^{|\boldsymbol{x}_0|} N(\text{Ins}(\boldsymbol{x}_t, i, v(\boldsymbol{x}_t, \boldsymbol{y})), \boldsymbol{x}_0)]}{\mathbb{E}_{\boldsymbol{x}_0}[(1 - e^{-\bar{\sigma}(t)})^{|\boldsymbol{x}_0|} N(\boldsymbol{x}_t, \boldsymbol{x}_0)]} \right).
\tag{41}
$$

By definition of the index set $I(\boldsymbol{x}_t, \boldsymbol{y})$, for any $i \in I(\boldsymbol{x}_t, \boldsymbol{y})$, we have $\text{Ins}(\boldsymbol{x}_t, i, v(\boldsymbol{x}_t, \boldsymbol{y})) = \boldsymbol{y}$. Therefore:

$$
\text{RHS} = \frac{e^{-\bar{\sigma}(t)}}{1 - e^{-\bar{\sigma}(t)}} \frac{1}{N(\boldsymbol{x}_t, \boldsymbol{y})} \sum_{i \in I(\boldsymbol{x}_t, \boldsymbol{y})} \left( \frac{\mathbb{E}_{\boldsymbol{x}_0}[(1 - e^{-\bar{\sigma}(t)})^{|\boldsymbol{x}_0|} N(\boldsymbol{y}, \boldsymbol{x}_0)]}{\mathbb{E}_{\boldsymbol{x}_0}[(1 - e^{-\bar{\sigma}(t)})^{|\boldsymbol{x}_0|} N(\boldsymbol{x}_t, \boldsymbol{x}_0)]} \right).
\tag{42}
$$

The term inside the summation is independent of the index $i$. We factor it out and utilize the property that $\sum_{i \in I(\boldsymbol{x}_t, \boldsymbol{y})} 1 = |I(\boldsymbol{x}_t, \boldsymbol{y})| = N(\boldsymbol{x}_t, \boldsymbol{y})$:

$$
\text{RHS} = \frac{e^{-\bar{\sigma}(t)}}{1 - e^{-\bar{\sigma}(t)}} \frac{1}{N(\boldsymbol{x}_t, \boldsymbol{y})} \left( \frac{\mathbb{E}_{\boldsymbol{x}_0}[(1 - e^{-\bar{\sigma}(t)})^{|\boldsymbol{x}_0|} N(\boldsymbol{y}, \boldsymbol{x}_0)]}{\mathbb{E}_{\boldsymbol{x}_0}[(1 - e^{-\bar{\sigma}(t)})^{|\boldsymbol{x}_0|} N(\boldsymbol{x}_t, \boldsymbol{x}_0)]} \right) N(\boldsymbol{x}_t, \boldsymbol{y})
\tag{43}
$$

$$
= \frac{e^{-\bar{\sigma}(t)}}{1 - e^{-\bar{\sigma}(t)}} \frac{\mathbb{E}_{\boldsymbol{x}_0}\left[(1 - e^{-\bar{\sigma}(t)})^{|\boldsymbol{x}_0|} N(\boldsymbol{y}, \boldsymbol{x}_0)\right]}{\mathbb{E}_{\boldsymbol{x}_0}\left[(1 - e^{-\bar{\sigma}(t)})^{|\boldsymbol{x}_0|} N(\boldsymbol{x}_t, \boldsymbol{x}_0)\right]}.
\tag{44}
$$

This matches the derived concrete score in Eq. 39, completing the proof.

### D.5 DERIVATION OF THE SAMPLING PROBABILITY (EQ.11)

We aim to derive the probability of executing a specific insertion operation—inserting token $v$ after position $i$—within an infinitesimal time interval $[t - \Delta t, t]$ during the backward process.

The backward process is a CTMC characterized by the parameterized reverse transition rate matrix $\tilde{Q}_t^\theta$. The rate of transition from state $\boldsymbol{x}_t$ to a different state $\boldsymbol{y}$ is defined as:

$$
\tilde{Q}_t^\theta(\boldsymbol{x}_t, \boldsymbol{y}) = Q_t(\boldsymbol{y}, \boldsymbol{x}_t) s_\theta(\boldsymbol{x}_t, t)_{\boldsymbol{y}}.
\tag{45}
$$

In the DID framework, backward transitions occur only when $\boldsymbol{y} \succ_1 \boldsymbol{x}_t$. We substitute the forward transition rate (Eq.6), $Q_t(\boldsymbol{y}, \boldsymbol{x}_t) = \sigma(t) N(\boldsymbol{x}_t, \boldsymbol{y})$. We also substitute the parameterized concrete

score $s_\theta$, which is derived by parameterizing the relationship between the concrete score and the insertion score (Eq.9):

$$s_\theta(\boldsymbol{x}_t, t)_{\boldsymbol{y}} = \frac{e^{-\bar{\sigma}(t)}}{1 - e^{-\bar{\sigma}(t)}} \frac{1}{N(\boldsymbol{x}_t, \boldsymbol{y})} \sum_{j \in I(\boldsymbol{x}_t, \boldsymbol{y})} \bar{s}_\theta(\boldsymbol{x}_t, t)[j, v(\boldsymbol{x}_t, \boldsymbol{y})]. \tag{46}$$

Substituting these expressions into the definition of $\tilde{Q}_t^\theta(\boldsymbol{x}_t, \boldsymbol{y})$:

$$\tilde{Q}_t^\theta(\boldsymbol{x}_t, \boldsymbol{y}) = (\sigma(t) N(\boldsymbol{x}_t, \boldsymbol{y})) \cdot \left( \frac{e^{-\bar{\sigma}(t)}}{1 - e^{-\bar{\sigma}(t)}} \frac{1}{N(\boldsymbol{x}_t, \boldsymbol{y})} \sum_{j \in I(\boldsymbol{x}_t, \boldsymbol{y})} \bar{s}_\theta(\boldsymbol{x}_t, t)[j, v(\boldsymbol{x}_t, \boldsymbol{y})] \right) \tag{47}$$

$$= \sigma(t) \frac{e^{-\bar{\sigma}(t)}}{1 - e^{-\bar{\sigma}(t)}} \sum_{j \in I(\boldsymbol{x}_t, \boldsymbol{y})} \bar{s}_\theta(\boldsymbol{x}_t, t)[j, v(\boldsymbol{x}_t, \boldsymbol{y})]. \tag{48}$$

This result demonstrates that the total transition rate from $\boldsymbol{x}_t$ to $\boldsymbol{y}$ is decomposed into a summation of individual components. Each term in the summation, $\sigma(t) \frac{e^{-\bar{\sigma}(t)}}{1 - e^{-\bar{\sigma}(t)}} \bar{s}_\theta(\boldsymbol{x}_t, t)[j, v(\boldsymbol{x}_t, \boldsymbol{y})]$, corresponds to the instantaneous rate of the specific insertion operation $(j, v(\boldsymbol{x}_t, \boldsymbol{y}))$ that transforms $\boldsymbol{x}_t$ into $\boldsymbol{y}$.

By the definition of a CTMC, the probability of a specific operation $(i, v)$ occurring within the infinitesimal interval $\Delta t$ is given by its corresponding rate multiplied by $\Delta t$. For $v \neq \varnothing$, we identify this probability directly from the decomposition above:

$$p_{t-\Delta t|t}^\theta((i, v)|\boldsymbol{x}_t) = \left( \sigma(t) \frac{e^{-\bar{\sigma}(t)}}{1 - e^{-\bar{\sigma}(t)}} \bar{s}_\theta(\boldsymbol{x}_t, t)[i, v] \right) \Delta t + o(\Delta t) \tag{49}$$

$$= \frac{\sigma(t) e^{-\bar{\sigma}(t)}}{1 - e^{-\bar{\sigma}(t)}} \bar{s}_\theta(\boldsymbol{x}_t, t)[i, v] \Delta t + o(\Delta t). \tag{50}$$

This confirms the first case of Eq.11. The probability of no insertion occurring at position $i$ (i.e., $v = \varnothing$) is determined by the normalization constraint, ensuring the sum of probabilities for all possible events at that position equals 1.

To confirm the equivalence between this action-based sampling and the required state-based transition, we verify that the action probabilities correctly recover the state transition probability $p_{t-\Delta t|t}^\theta(\boldsymbol{y}|\boldsymbol{x}_t)$. A transition to $\boldsymbol{y}$ occurs if any of the actions indexed by $I(\boldsymbol{x}_t, \boldsymbol{y})$ occurs. In the infinitesimal limit $\Delta t \to 0$, these actions are mutually exclusive events:

$$p_{t-\Delta t|t}^\theta(\boldsymbol{y}|\boldsymbol{x}_t) = \sum_{j \in I(\boldsymbol{x}_t, \boldsymbol{y})} p_{t-\Delta t|t}^\theta((j, v(\boldsymbol{x}_t, \boldsymbol{y}))|\boldsymbol{x}_t) + o(\Delta t) \tag{51}$$

$$= \left( \sigma(t) \frac{e^{-\bar{\sigma}(t)}}{1 - e^{-\bar{\sigma}(t)}} \sum_{j \in I(\boldsymbol{x}_t, \boldsymbol{y})} \bar{s}_\theta(\boldsymbol{x}_t, t)[j, v(\boldsymbol{x}_t, \boldsymbol{y})] \right) \Delta t + o(\Delta t). \tag{52}$$

By Eq. 48, the term in the parenthesis is exactly $\tilde{Q}_t^\theta(\boldsymbol{x}_t, \boldsymbol{y})$. Thus, $p_{t-\Delta t|t}^\theta(\boldsymbol{y}|\boldsymbol{x}_t) = \tilde{Q}_t^\theta(\boldsymbol{x}_t, \boldsymbol{y}) \Delta t + o(\Delta t)$, confirming that the action-based sampling correctly implements the dynamics of the backward CTMC.

### D.6 DERIVATION OF THE DISE OBJECTIVE (EQ.12)

We derive the Denoising Insertion Score Entropy (DISE) objective (Eq.12) starting from the general DSE objective (Eq.3), demonstrating that DISE is a variational upper bound on DSE, $\mathcal{L}_\theta^{\text{DISE}}(\boldsymbol{x}_0) \geq \mathcal{L}_\theta^{\text{DSE}}(\boldsymbol{x}_0)$.

We begin with the DSE objective:

$$\mathcal{L}_\theta^{\text{DSE}}(\boldsymbol{x}_0) = \mathbb{E}_{t, \boldsymbol{x}_t} \sum_{\boldsymbol{y} \neq \boldsymbol{x}_t} Q_t(\boldsymbol{y}, \boldsymbol{x}_t) \left[ s_\theta(\boldsymbol{x}_t, t)_{\boldsymbol{y}} - \frac{p_{t|0}(\boldsymbol{y}|\boldsymbol{x}_0)}{p_{t|0}(\boldsymbol{x}_t|\boldsymbol{x}_0)} \log s_\theta(\boldsymbol{x}_t, t)_{\boldsymbol{y}} + K\left( \frac{p_{t|0}(\boldsymbol{y}|\boldsymbol{x}_0)}{p_{t|0}(\boldsymbol{x}_t|\boldsymbol{x}_0)} \right) \right], \tag{53}$$

where $K(a) = a(\log a - 1)$. In the deletion–insertion process, the transition rate $Q_t(\boldsymbol{y}, \boldsymbol{x}_t)$ is non-zero only when $\boldsymbol{y} \succ_1 \boldsymbol{x}_t$.

We now substitute the specific definitions for the DID process. The transition rate is $Q_t(\boldsymbol{y}, \boldsymbol{x}_t) = \sigma(t)N(\boldsymbol{x}_t, \boldsymbol{y})$. The conditional probability ratio is:

$$\frac{p_{t|0}(\boldsymbol{y} \mid \boldsymbol{x}_0)}{p_{t|0}(\boldsymbol{x}_t \mid \boldsymbol{x}_0)} = \frac{e^{-\bar{\sigma}(t)}}{1 - e^{-\bar{\sigma}(t)}} \frac{N(\boldsymbol{y}, \boldsymbol{x}_0)}{N(\boldsymbol{x}_t, \boldsymbol{x}_0)}. \tag{54}$$

The parameterized concrete score (from parameterized Eq.9) is:

$$s_\theta(\boldsymbol{x}_t, t)_{\boldsymbol{y}} = \frac{e^{-\bar{\sigma}(t)}}{1 - e^{-\bar{\sigma}(t)}} \frac{1}{N(\boldsymbol{x}_t, \boldsymbol{y})} \sum_{i \in I(\boldsymbol{x}_t, \boldsymbol{y})} \bar{s}_\theta(\boldsymbol{x}_t, t)[i, v(\boldsymbol{x}_t, \boldsymbol{y})]. \tag{55}$$

We examine the expression inside the bracket of the DSE objective by substituting these definitions. Let $v = v(\boldsymbol{x}_t, \boldsymbol{y})$ for brevity in the following block:

$$\begin{aligned}
&s_\theta(\boldsymbol{x}_t, t)_{\boldsymbol{y}} - \frac{p_{t|0}(\boldsymbol{y} \mid \boldsymbol{x}_0)}{p_{t|0}(\boldsymbol{x}_t \mid \boldsymbol{x}_0)} \log s_\theta(\boldsymbol{x}_t, t)_{\boldsymbol{y}} + K\left(\frac{p_{t|0}(\boldsymbol{y} \mid \boldsymbol{x}_0)}{p_{t|0}(\boldsymbol{x}_t \mid \boldsymbol{x}_0)}\right) \\
&= \frac{e^{-\bar{\sigma}(t)}}{1 - e^{-\bar{\sigma}(t)}} \frac{1}{N(\boldsymbol{x}_t, \boldsymbol{y})} \left(\sum_{i \in I(\boldsymbol{x}_t, \boldsymbol{y})} \bar{s}_\theta(\boldsymbol{x}_t, t)[i, v]\right) \\
&\quad - \left(\frac{e^{-\bar{\sigma}(t)}}{1 - e^{-\bar{\sigma}(t)}} \frac{N(\boldsymbol{y}, \boldsymbol{x}_0)}{N(\boldsymbol{x}_t, \boldsymbol{x}_0)}\right) \log \left(\frac{e^{-\bar{\sigma}(t)}}{1 - e^{-\bar{\sigma}(t)}} \frac{1}{N(\boldsymbol{x}_t, \boldsymbol{y})} \left(\sum_{i \in I(\boldsymbol{x}_t, \boldsymbol{y})} \bar{s}_\theta(\boldsymbol{x}_t, t)[i, v]\right)\right) \\
&\quad + K\left(\frac{e^{-\bar{\sigma}(t)}}{1 - e^{-\bar{\sigma}(t)}} \frac{N(\boldsymbol{y}, \boldsymbol{x}_0)}{N(\boldsymbol{x}_t, \boldsymbol{x}_0)}\right).
\end{aligned} \tag{56}$$

We expand the $K(\cdot)$ term using $K(a) = a(\log a - 1)$. By expanding the logarithms ($\log(AB) = \log A + \log B$), we observe that the terms involving the time factor $\log(\frac{e^{-\bar{\sigma}(t)}}{1-e^{-\bar{\sigma}(t)}})$ cancel out exactly.

The expression inside the bracket simplifies significantly by factoring out the time prefactor:

$$\begin{aligned}
&= \frac{e^{-\bar{\sigma}(t)}}{1 - e^{-\bar{\sigma}(t)}} \left[\frac{1}{N(\boldsymbol{x}_t, \boldsymbol{y})} \left(\sum_{i \in I(\boldsymbol{x}_t, \boldsymbol{y})} \bar{s}_\theta(\boldsymbol{x}_t, t)[i, v]\right)\right. \\
&\quad - \frac{N(\boldsymbol{y}, \boldsymbol{x}_0)}{N(\boldsymbol{x}_t, \boldsymbol{x}_0)} \log \left(\frac{1}{N(\boldsymbol{x}_t, \boldsymbol{y})} \left(\sum_{i \in I(\boldsymbol{x}_t, \boldsymbol{y})} \bar{s}_\theta(\boldsymbol{x}_t, t)[i, v]\right)\right) \\
&\quad \left. + \frac{N(\boldsymbol{y}, \boldsymbol{x}_0)}{N(\boldsymbol{x}_t, \boldsymbol{x}_0)} \left(\log \frac{N(\boldsymbol{y}, \boldsymbol{x}_0)}{N(\boldsymbol{x}_t, \boldsymbol{x}_0)} - 1\right)\right].
\end{aligned} \tag{57}$$

Substituting this simplified bracket back into the main objective equation and multiplying by $Q_t(\boldsymbol{y}, \boldsymbol{x}_t) = \sigma(t)N(\boldsymbol{x}_t, \boldsymbol{y})$. The DSE objective can be decomposed into three terms (T1, T2, T3):

$$\mathcal{L}_\theta^{\text{DSE}}(\boldsymbol{x}_0) = \text{T1} + \text{T2} + \text{T3}. \tag{58}$$

We define these terms based on the components derived above:

$$\text{T1} = \mathop{\mathbb{E}}_{t, \boldsymbol{x}_t} \frac{\sigma(t)e^{-\bar{\sigma}(t)}}{1 - e^{-\bar{\sigma}(t)}} \sum_{\boldsymbol{y} \succ_1 \boldsymbol{x}_t} N(\boldsymbol{x}_t, \boldsymbol{y}) \left[\frac{1}{N(\boldsymbol{x}_t, \boldsymbol{y})} \sum_{i \in I(\boldsymbol{x}_t, \boldsymbol{y})} \bar{s}_\theta(\boldsymbol{x}_t, t)[i, v]\right]. \tag{59}$$

$$\text{T2} = \mathop{\mathbb{E}}_{t, \boldsymbol{x}_t} \frac{\sigma(t)e^{-\bar{\sigma}(t)}}{1 - e^{-\bar{\sigma}(t)}} \sum_{\boldsymbol{y} \succ_1 \boldsymbol{x}_t} N(\boldsymbol{x}_t, \boldsymbol{y}) \left[-\frac{N(\boldsymbol{y}, \boldsymbol{x}_0)}{N(\boldsymbol{x}_t, \boldsymbol{x}_0)} \log \left(\frac{1}{N(\boldsymbol{x}_t, \boldsymbol{y})} \sum_{i \in I(\boldsymbol{x}_t, \boldsymbol{y})} \bar{s}_\theta(\boldsymbol{x}_t, t)[i, v]\right)\right]. \tag{60}$$

$$\text{T3} = \mathop{\mathbb{E}}_{t,\boldsymbol{x}_t} \frac{\sigma(t)e^{-\bar{\sigma}(t)}}{1 - e^{-\bar{\sigma}(t)}} \sum_{\boldsymbol{y} \succ_1 \boldsymbol{x}_t} N(\boldsymbol{x}_t, \boldsymbol{y}) K\left(\frac{N(\boldsymbol{y}, \boldsymbol{x}_0)}{N(\boldsymbol{x}_t, \boldsymbol{x}_0)}\right). \tag{61}$$

We now apply Jensen's inequality to T2. The term inside the logarithm is an average of insertion scores. Because the logarithm function is concave, $\log(\frac{1}{N} \sum a_i) \geq \frac{1}{N} \sum \log a_i$. Since the logarithm is negated, this leads to an upper bound on T2:

$$-\log\left(\frac{1}{N(\boldsymbol{x}_t, \boldsymbol{y})} \sum_{i \in I(\boldsymbol{x}_t, \boldsymbol{y})} \bar{s}_\theta(\boldsymbol{x}_t, t)[i, v]\right) \leq -\frac{1}{N(\boldsymbol{x}_t, \boldsymbol{y})} \sum_{i \in I(\boldsymbol{x}_t, \boldsymbol{y})} \log \bar{s}_\theta(\boldsymbol{x}_t, t)[i, v]. \tag{62}$$

We define T2$_{\text{Bound}}$ as the upper bound for T2:

$$\text{T2} \leq \text{T2}_{\text{Bound}} = \mathop{\mathbb{E}}_{t,\boldsymbol{x}_t} \frac{\sigma(t)e^{-\bar{\sigma}(t)}}{1 - e^{-\bar{\sigma}(t)}} \sum_{\boldsymbol{y} \succ_1 \boldsymbol{x}_t} N(\boldsymbol{x}_t, \boldsymbol{y}) \frac{N(\boldsymbol{y}, \boldsymbol{x}_0)}{N(\boldsymbol{x}_t, \boldsymbol{x}_0)} \left[-\frac{1}{N(\boldsymbol{x}_t, \boldsymbol{y})} \sum_{i \in I(\boldsymbol{x}_t, \boldsymbol{y})} \log \bar{s}_\theta(\boldsymbol{x}_t, t)[i, v]\right]. \tag{63}$$

We define the DISE objective as the upper bound obtained by replacing T2 with T2$_{\text{Bound}}$, ensuring $\mathcal{L}_\theta^{\text{DISE}}(\boldsymbol{x}_0) \geq \mathcal{L}_\theta^{\text{DSE}}(\boldsymbol{x}_0)$:

$$\mathcal{L}_\theta^{\text{DISE}}(\boldsymbol{x}_0) = \text{T1} + \text{T2}_{\text{Bound}} + \text{T3}. \tag{64}$$

To simplify the nested summations and transform the objective from state-level ($\boldsymbol{y}$) to operation-level ($(i, v)$), we rely on the following identity.

**Lemma 1** (Summation Change of Variables). Let $v(\boldsymbol{x}_t, \boldsymbol{y})$ denote the unique inserted token and $I(\boldsymbol{x}_t, \boldsymbol{y})$ the set of valid insertion positions that yield $\boldsymbol{y}$ from $\boldsymbol{x}_t$. For any function $G$,

$$\sum_{\boldsymbol{y} \succ_1 \boldsymbol{x}_t} \sum_{i \in I(\boldsymbol{x}_t, \boldsymbol{y})} G\big(i, v(\boldsymbol{x}_t, \boldsymbol{y}), \boldsymbol{y}\big) = \sum_{i,v} G\big(i, v, \text{Ins}(\boldsymbol{x}_t, i, v)\big). \tag{65}$$

*Proof.* Define the index sets

$$\mathcal{A} = \{(\boldsymbol{y}, i) : \boldsymbol{y} \succ_1 \boldsymbol{x}_t, i \in I(\boldsymbol{x}_t, \boldsymbol{y})\}, \qquad \mathcal{B} = \{(i, v) : i \in \{0, \ldots, |\boldsymbol{x}_t|\}, v \in \mathcal{V}\}. \tag{66}$$

The map $g : \mathcal{B} \to \mathcal{A}$ defined by $g(i, v) = (\text{Ins}(\boldsymbol{x}_t, i, v), i)$ is a bijection, with its inverse $f : \mathcal{A} \to \mathcal{B}$ defined by $f(\boldsymbol{y}, i) = (i, v(\boldsymbol{x}_t, \boldsymbol{y}))$. Changing variables over this bijection gives the claimed identity. $\square$

We apply Lemma 1 to simplify T1, T2$_{\text{Bound}}$, and T3.

For T1, the $N(\boldsymbol{x}_t, \boldsymbol{y})$ terms cancel before the summation.

$$\text{T1} = \mathop{\mathbb{E}}_{t,\boldsymbol{x}_t} \frac{\sigma(t)e^{-\bar{\sigma}(t)}}{1 - e^{-\bar{\sigma}(t)}} \sum_{\boldsymbol{y} \succ_1 \boldsymbol{x}_t} \sum_{i \in I(\boldsymbol{x}_t, \boldsymbol{y})} \bar{s}_\theta(\boldsymbol{x}_t, t)[i, v(\boldsymbol{x}_t, \boldsymbol{y})] \tag{67}$$

$$= \mathop{\mathbb{E}}_{t,\boldsymbol{x}_t} \frac{\sigma(t)e^{-\bar{\sigma}(t)}}{1 - e^{-\bar{\sigma}(t)}} \sum_{i,v} \bar{s}_\theta(\boldsymbol{x}_t, t)[i, v]. \quad \text{(Using Lemma 1)} \tag{68}$$

For T2$_{\text{Bound}}$, cancellation of $N(\boldsymbol{x}_t, \boldsymbol{y})$ yields:

$$\text{T2}_{\text{Bound}} = \mathop{\mathbb{E}}_{t,\boldsymbol{x}_t} \frac{\sigma(t)e^{-\bar{\sigma}(t)}}{1 - e^{-\bar{\sigma}(t)}} \sum_{\boldsymbol{y} \succ_1 \boldsymbol{x}_t} \sum_{i \in I(\boldsymbol{x}_t, \boldsymbol{y})} -\frac{N(\boldsymbol{y}, \boldsymbol{x}_0)}{N(\boldsymbol{x}_t, \boldsymbol{x}_0)} \log \bar{s}_\theta(\boldsymbol{x}_t, t)[i, v(\boldsymbol{x}_t, \boldsymbol{y})]. \tag{69}$$

Using Lemma 1, noting that $\boldsymbol{y} = \text{Ins}(\boldsymbol{x}_t, i, v)$:

$$\text{T2}_{\text{Bound}} = \mathop{\mathbb{E}}_{t,\boldsymbol{x}_t} \frac{\sigma(t)e^{-\bar{\sigma}(t)}}{1 - e^{-\bar{\sigma}(t)}} \sum_{i,v} -\frac{N(\text{Ins}(\boldsymbol{x}_t, i, v), \boldsymbol{x}_0)}{N(\boldsymbol{x}_t, \boldsymbol{x}_0)} \log \bar{s}_\theta(\boldsymbol{x}_t, t)[i, v]. \tag{70}$$

For T3, we first utilize the fact that $N(\boldsymbol{x}_t, \boldsymbol{y}) = \sum_{i \in I(\boldsymbol{x}_t, \boldsymbol{y})} 1$.

$$\text{T3} = \mathop{\mathbb{E}}_{t, \boldsymbol{x}_t} \frac{\sigma(t) e^{-\bar{\sigma}(t)}}{1 - e^{-\bar{\sigma}(t)}} \sum_{\boldsymbol{y} \succ_1 \boldsymbol{x}_t} \sum_{i \in I(\boldsymbol{x}_t, \boldsymbol{y})} K\left( \frac{N(\boldsymbol{y}, \boldsymbol{x}_0)}{N(\boldsymbol{x}_t, \boldsymbol{x}_0)} \right). \tag{71}$$

Applying Lemma 1 to T3:

$$\text{T3} = \mathop{\mathbb{E}}_{t, \boldsymbol{x}_t} \frac{\sigma(t) e^{-\bar{\sigma}(t)}}{1 - e^{-\bar{\sigma}(t)}} \sum_{i,v} K\left( \frac{N(\text{Ins}(\boldsymbol{x}_t, i, v), \boldsymbol{x}_0)}{N(\boldsymbol{x}_t, \boldsymbol{x}_0)} \right). \tag{72}$$

We combine T1, T2$_{\text{Bound}}$, and T3 to obtain the final DISE objective. Let $C$ denote the constant term T3's summand.

$$\mathcal{L}_\theta^{\text{DISE}}(\boldsymbol{x}_0) = \text{T1} + \text{T2}_{\text{Bound}} + \text{T3} \tag{73}$$

$$= \mathop{\mathbb{E}}_{t, \boldsymbol{x}_t} \left\{ \frac{\sigma(t) e^{-\bar{\sigma}(t)}}{1 - e^{-\bar{\sigma}(t)}} \sum_{i,v} \left[ \bar{s}_\theta(\boldsymbol{x}_t, t)[i, v] - \frac{N(\text{Ins}(\boldsymbol{x}_t, i, v), \boldsymbol{x}_0)}{N(\boldsymbol{x}_t, \boldsymbol{x}_0)} \log \bar{s}_\theta(\boldsymbol{x}_t, t)[i, v] + C \right] \right\}. \tag{74}$$

### D.7 DERIVATION OF THE SEQUENCE-LEVEL NORMALIZATION PROPERTY (EQ. 17)

In the fixed-length setting (Sec. 3.5), we assume $|\boldsymbol{x}_0| = K$ (a constant) for all data samples. Under this assumption, the insertion score becomes time-independent. We aim to prove the normalization property (Eq. 17), restated here for the fixed-length context:

$$\sum_{i,v} \bar{s}(\boldsymbol{x}_t)[i, v] = K - |\boldsymbol{x}_t|. \tag{75}$$

The proof relies on a fundamental combinatorial identity regarding subsequence counts.

**Lemma 2** (Subsequence Count Identity). *For any sequences $\boldsymbol{x}_t$ and $\boldsymbol{x}_0$ such that $\boldsymbol{x}_t$ is a subsequence of $\boldsymbol{x}_0$, the following identity holds:*

$$\sum_{i,v} N(\text{Ins}(\boldsymbol{x}_t, i, v), \boldsymbol{x}_0) = N(\boldsymbol{x}_t, \boldsymbol{x}_0)(|\boldsymbol{x}_0| - |\boldsymbol{x}_t|). \tag{76}$$

*Proof of Lemma 2.* We prove the statement $\sum_{i,v} N(\text{Ins}(\boldsymbol{x}_t, i, v), \boldsymbol{x}_0) = N(\boldsymbol{x}_t, \boldsymbol{x}_0)(|\boldsymbol{x}_0| - |\boldsymbol{x}_t|)$ via a bijective proof, by constructing two sets of equal cardinality. Let $S(\boldsymbol{x}, \boldsymbol{z})$ denote the set of index tuples corresponding to all occurrences of a subsequence $\boldsymbol{x}$ in $\boldsymbol{z}$, such that $N(\boldsymbol{x}, \boldsymbol{z}) = |S(\boldsymbol{x}, \boldsymbol{z})|$.

First, consider the set A, defined as the set of pairs $(I, j)$, where $I$ is the index tuple of an occurrence of $\boldsymbol{x}_t$ in $\boldsymbol{x}_0$, and $j$ is an index in $\boldsymbol{x}_0$ that is not part of that occurrence:

$$A = \{(I, j) : I \in S(\boldsymbol{x}_t, \boldsymbol{x}_0), j \in \{1, \ldots, |\boldsymbol{x}_0|\} \setminus I\}. \tag{77}$$

The cardinality of A is $|A| = N(\boldsymbol{x}_t, \boldsymbol{x}_0)(|\boldsymbol{x}_0| - |\boldsymbol{x}_t|)$.

Second, consider the set B, defined as the set of pairs $((i, v), J)$, where $(i, v)$ is an insertion operation on $\boldsymbol{x}_t$, and $J$ is the index tuple of an occurrence of the resulting sequence, $\text{Ins}(\boldsymbol{x}_t, i, v)$, in $\boldsymbol{x}_0$:

$$B = \{((i, v), J) : J \in S(\text{Ins}(\boldsymbol{x}_t, i, v), \boldsymbol{x}_0)\}. \tag{78}$$

The cardinality of B is $|B| = \sum_{i,v} N(\text{Ins}(\boldsymbol{x}_t, i, v), \boldsymbol{x}_0)$.

We now establish a bijection between A and B. For any element $(I, j) \in A$, we define a mapping to an element in B as follows: let the inserted token be $v = \boldsymbol{x}_0[j]$, and let the insertion position relative to $\boldsymbol{x}_t$ be $i = |\{k \in I : k < j\}|$. The new index tuple is $J = I \cup \{j\}$, sorted. The subsequence $\boldsymbol{x}_0[J]$ is precisely $\text{Ins}(\boldsymbol{x}_t, i, v)$ by construction. This defines a unique mapping $f : A \to B$.

Conversely, for any element $((i, v), J) \in B$, we can define an inverse mapping. The index of the inserted token in $\boldsymbol{x}_0$ is the $(i + 1)$-th element of the sorted tuple $J$; let this be $j$. Removing this index yields the tuple $I = J \setminus \{j\}$, which corresponds to an occurrence of $\boldsymbol{x}_t$. This defines a unique mapping $g : B \to A$.

Since a one-to-one correspondence exists between the sets, their cardinalities must be equal. Therefore, $|A| = |B|$, which proves the lemma. $\square$

*Proof of Eq. 17.* Under the fixed-length assumption ($|\boldsymbol{x}_0| = K$), the definition of the insertion score (Eq. 8) simplifies because the time-dependent terms $(1 - e^{-\bar{\sigma}(t)})^{|\boldsymbol{x}_0|}$ are constant $(1 - e^{-\bar{\sigma}(t)})^K$ and cancel out, leading to the time-independent score:

$$\bar{s}(\boldsymbol{x}_t)[i, v] = \frac{\mathbb{E}_{\boldsymbol{x}_0}[N(\text{Ins}(\boldsymbol{x}_t, i, v), \boldsymbol{x}_0)]}{\mathbb{E}_{\boldsymbol{x}_0}[N(\boldsymbol{x}_t, \boldsymbol{x}_0)]}. \tag{79}$$

We sum this score over all possible insertion operations $(i, v)$:

$$\sum_{i,v} \bar{s}(\boldsymbol{x}_t)[i, v] = \sum_{i,v} \frac{\mathbb{E}_{\boldsymbol{x}_0}[N(\text{Ins}(\boldsymbol{x}_t, i, v), \boldsymbol{x}_0)]}{\mathbb{E}_{\boldsymbol{x}_0}[N(\boldsymbol{x}_t, \boldsymbol{x}_0)]} \tag{80}$$

$$= \frac{1}{\mathbb{E}_{\boldsymbol{x}_0}[N(\boldsymbol{x}_t, \boldsymbol{x}_0)]} \mathbb{E}_{\boldsymbol{x}_0} \left[ \sum_{i,v} N(\text{Ins}(\boldsymbol{x}_t, i, v), \boldsymbol{x}_0) \right]. \tag{81}$$

We apply the combinatorial identity (Lemma 2) to the summation inside the expectation, substituting $|\boldsymbol{x}_0| = K$:

$$\sum_{i,v} N(\text{Ins}(\boldsymbol{x}_t, i, v), \boldsymbol{x}_0) = N(\boldsymbol{x}_t, \boldsymbol{x}_0)(K - |\boldsymbol{x}_t|). \tag{82}$$

Substituting this back:

$$\sum_{i,v} \bar{s}(\boldsymbol{x}_t)[i, v] = \frac{1}{\mathbb{E}_{\boldsymbol{x}_0}[N(\boldsymbol{x}_t, \boldsymbol{x}_0)]} \mathbb{E}_{\boldsymbol{x}_0} \left[ N(\boldsymbol{x}_t, \boldsymbol{x}_0)(K - |\boldsymbol{x}_t|) \right]. \tag{83}$$

Since $(K - |\boldsymbol{x}_t|)$ is constant with respect to the expectation over $\boldsymbol{x}_0$:

$$\sum_{i,v} \bar{s}(\boldsymbol{x}_t)[i, v] = (K - |\boldsymbol{x}_t|) \frac{\mathbb{E}_{\boldsymbol{x}_0}[N(\boldsymbol{x}_t, \boldsymbol{x}_0)]}{\mathbb{E}_{\boldsymbol{x}_0}[N(\boldsymbol{x}_t, \boldsymbol{x}_0)]} = K - |\boldsymbol{x}_t|. \tag{84}$$

This confirms the normalization property stated in Eq. 17. $\square$

### D.8 DERIVATION OF THE DICE OBJECTIVE FOR FIXED-LENGTH DATA (EQ. 18)

In the fixed-length setting (Section 3.5), we assume $|\boldsymbol{x}_0| = K$ (constant). This assumption leads to time-independent insertion scores $\bar{s}_\theta(\boldsymbol{x}_t)[i, v]$ (as shown in Appendix D.7) and allows for the exact simplification of the DISE objective into the Denoising Insertion Cross-Entropy (DICE) objective.

We start from the DISE objective (Eq.12):

$$\mathcal{L}_\theta^{\text{DISE}}(\boldsymbol{x}_0) = \mathbb{E}_{t,\boldsymbol{x}_t} \frac{\sigma(t)e^{-\bar{\sigma}(t)}}{1 - e^{-\bar{\sigma}(t)}} \sum_{i,v} \left[ \bar{s}_\theta(\boldsymbol{x}_t)[i, v] - \frac{N(\text{Ins}(\boldsymbol{x}_t, i, v), \boldsymbol{x}_0)}{N(\boldsymbol{x}_t, \boldsymbol{x}_0)} \log \bar{s}_\theta(\boldsymbol{x}_t)[i, v] \right.$$
$$\left. + K \left( \frac{N(\text{Ins}(\boldsymbol{x}_t, i, v), \boldsymbol{x}_0)}{N(\boldsymbol{x}_t, \boldsymbol{x}_0)} \right) \right]. \tag{85}$$

We rearrange the expression inside the square brackets using the definition $K(a) = a(\log a - 1)$.

$$\bar{s}_\theta[i, v] - \frac{N(\text{Ins}(\boldsymbol{x}_t, i, v), \boldsymbol{x}_0)}{N(\boldsymbol{x}_t, \boldsymbol{x}_0)} \log \bar{s}_\theta[i, v] + \frac{N(\text{Ins}(\boldsymbol{x}_t, i, v), \boldsymbol{x}_0)}{N(\boldsymbol{x}_t, \boldsymbol{x}_0)} \left( \log \frac{N(\text{Ins}(\boldsymbol{x}_t, i, v), \boldsymbol{x}_0)}{N(\boldsymbol{x}_t, \boldsymbol{x}_0)} - 1 \right)$$
$$= \left( \bar{s}_\theta[i, v] - \frac{N(\text{Ins}(\boldsymbol{x}_t, i, v), \boldsymbol{x}_0)}{N(\boldsymbol{x}_t, \boldsymbol{x}_0)} \right)$$
$$+ \frac{N(\text{Ins}(\boldsymbol{x}_t, i, v), \boldsymbol{x}_0)}{N(\boldsymbol{x}_t, \boldsymbol{x}_0)} \left( \log \frac{N(\text{Ins}(\boldsymbol{x}_t, i, v), \boldsymbol{x}_0)}{N(\boldsymbol{x}_t, \boldsymbol{x}_0)} - \log \bar{s}_\theta[i, v] \right). \tag{86}$$

We substitute this rearrangement back into the DISE objective:

$$\mathcal{L}_\theta^{\text{DISE}}(\boldsymbol{x}_0) = \mathbb{E}_{t,\boldsymbol{x}_t} \frac{\sigma(t)e^{-\bar{\sigma}(t)}}{1 - e^{-\bar{\sigma}(t)}} \sum_{i,v} \left[ \left( \bar{s}_\theta[i, v] - \frac{N(\text{Ins}(\boldsymbol{x}_t, i, v), \boldsymbol{x}_0)}{N(\boldsymbol{x}_t, \boldsymbol{x}_0)} \right) \right.$$

$$+ \frac{N(\text{Ins}(\boldsymbol{x}_t, i, v), \boldsymbol{x}_0)}{N(\boldsymbol{x}_t, \boldsymbol{x}_0)} \left( \log \frac{N(\text{Ins}(\boldsymbol{x}_t, i, v), \boldsymbol{x}_0)}{N(\boldsymbol{x}_t, \boldsymbol{x}_0)} - \log \bar{s}_\theta[i, v] \right) \Bigg].$$
(87)

We now utilize the crucial normalization properties derived from the fixed-length assumption ($|\boldsymbol{x}_0| = K$). From Lemma 2 (Appendix D.7), the true subsequence count ratios satisfy:

$$\sum_{i,v} \frac{N(\text{Ins}(\boldsymbol{x}_t, i, v), \boldsymbol{x}_0)}{N(\boldsymbol{x}_t, \boldsymbol{x}_0)} = K - |\boldsymbol{x}_t|.$$
(88)

As discussed in Section 3.5, we design the network architecture such that the parameterized scores $\bar{s}_\theta$ exactly satisfy the same normalization constraint:

$$\sum_{i,v} \bar{s}_\theta(\boldsymbol{x}_t)[i, v] = K - |\boldsymbol{x}_t|.$$
(89)

Because both the true ratios and the parameterized scores sum to the same value, the summation of the first group of terms in Eq. 86 vanishes:

$$\sum_{i,v} \left( \bar{s}_\theta(\boldsymbol{x}_t)[i, v] - \frac{N(\text{Ins}(\boldsymbol{x}_t, i, v), \boldsymbol{x}_0)}{N(\boldsymbol{x}_t, \boldsymbol{x}_0)} \right) = (K - |\boldsymbol{x}_t|) - (K - |\boldsymbol{x}_t|) = 0.$$
(90)

Therefore, the DISE objective simplifies exactly. We define this simplified form as the DICE objective, which is exactly equal to the DISE loss under the fixed-length setting ($\mathcal{L}_\theta^{\text{DICE}}(\boldsymbol{x}_0) = \mathcal{L}_\theta^{\text{DISE}}(\boldsymbol{x}_0)$):

$$\mathcal{L}_\theta^{\text{DICE}}(\boldsymbol{x}_0) = \mathbb{E}_{t,\boldsymbol{x}_t} \frac{\sigma(t)e^{-\bar{\sigma}(t)}}{1 - e^{-\bar{\sigma}(t)}} \sum_{i,v} \frac{N(\text{Ins}(\boldsymbol{x}_t, i, v), \boldsymbol{x}_0)}{N(\boldsymbol{x}_t, \boldsymbol{x}_0)} \left( \log \frac{N(\text{Ins}(\boldsymbol{x}_t, i, v), \boldsymbol{x}_0)}{N(\boldsymbol{x}_t, \boldsymbol{x}_0)} - \log \bar{s}_\theta(\boldsymbol{x}_t)[i, v] \right).$$
(91)

Rearranging the terms inside the summation yields the final DICE objective:

$$\mathcal{L}_\theta^{\text{DICE}}(\boldsymbol{x}_0) = \mathbb{E}_{t,\boldsymbol{x}_t} \left\{ \sum_{i,v} \frac{\sigma(t)e^{-\bar{\sigma}(t)}}{1 - e^{-\bar{\sigma}(t)}} \frac{N(\text{Ins}(\boldsymbol{x}_t, i, v), \boldsymbol{x}_0)}{N(\boldsymbol{x}_t, \boldsymbol{x}_0)} \left[ -\log \bar{s}_\theta(\boldsymbol{x}_t)[i, v] + C \right] \right\},$$
(92)

where $C = \log \frac{N(\text{Ins}(\boldsymbol{x}_t, i, v), \boldsymbol{x}_0)}{N(\boldsymbol{x}_t, \boldsymbol{x}_0)}$ is a $\theta$-free constant.

# E    ALGORITHM DETAILS

## E.1    DYNAMIC PROGRAMMING FOR SUBSEQUENCE COUNTING

Here we provide the pseudocode of the DP algorithms introduced in Sec. 3.4.

---
**Algorithm 1** Prefix DP (Eq. 14)

---
**Require:** original sequence $\boldsymbol{x}_0$ with length $m$, noised sequence $\boldsymbol{x}_t$ with length $n$
    Initialize $N(\boldsymbol{x}_t[: i], \boldsymbol{x}_0[: j]) = 0, \forall i, j$
    Initialize $N(\boldsymbol{x}_t[: 0], \boldsymbol{x}_0[: j]) = 1, \forall j$
    **for** $j = 1$ to $m$ **do**
      **for** $i = 1$ to $n$ **do**  // in parallel
        $N(\boldsymbol{x}_t[: i], \boldsymbol{x}_0[: j]) = N(\boldsymbol{x}_t[: i], \boldsymbol{x}_0[: j - 1]) + \delta(\boldsymbol{x}_t[i - 1], \boldsymbol{x}_0[j - 1]) \cdot N(\boldsymbol{x}_t[: i - 1], \boldsymbol{x}_0[: j - 1])$
      **end for**
    **end for**

---

---

**Algorithm 2** Suffix DP (Eq. 15)

---

**Require:** original sequence $\boldsymbol{x}_0$ with length $m$, noised sequence $\boldsymbol{x}_t$ with length $n$
    Initialize $N(\boldsymbol{x}_t[i:], \boldsymbol{x}_0[j:]) = 0, \forall i, j$
    Initialize $N(\boldsymbol{x}_t[n:], \boldsymbol{x}_0[j:]) = 1, \forall j$
    **for** $j = m - 1$ to $0$ **do**
        **for** $i = n - 1$ to $0$ **do**  // in parallel
            $N(\boldsymbol{x}_t[i:], \boldsymbol{x}_0[j:]) = N(\boldsymbol{x}_t[i:], \boldsymbol{x}_0[j+1:]) + \delta(\boldsymbol{x}_t[i], \boldsymbol{x}_0[j]) \cdot N(\boldsymbol{x}_t[i+1:], \boldsymbol{x}_0[j+1:])$
        **end for**
    **end for**

---

When we extend the DP algorithms to longer sequences, we will meet a numerical issue that the value of $N(\boldsymbol{x}_t, \boldsymbol{x}_0)$ and the values in the DP tables (Eq. 14,15) might be extremely large. For example, when $|\boldsymbol{x}_0| = 2048$, the maximum $N(\boldsymbol{x}_t, \boldsymbol{x}_0)$ is $\binom{2048}{1024} \sim 10^{614}$, larger than the upper limit of float64 precision $\sim 10^{308}$, resulting in a numerical overflow. However, what we want to compute to implement the DISE loss (Eq. 12) of DID is the 'N ratios': $\frac{N(\text{Ins}(\boldsymbol{x}_t, i, v), \boldsymbol{x}_0)}{N(\boldsymbol{x}_t, \boldsymbol{x}_0)}$, according to the sequence-level normalization property (Eq. 17, details in Appendix D.7), the ratios should be $\leq |\boldsymbol{x}_0| - |\boldsymbol{x}_t|$, i.e. the final results of the N ratios will not have any overflow issues. Therefore, to address the numerical overflow of the intermediate DP results, we can transform the DP algorithms into log-domain.

---

**Algorithm 3** Prefix DP in log-domain

---

**Require:** original sequence $\boldsymbol{x}_0$ with length $m$, noised sequence $\boldsymbol{x}_t$ with length $n$, INF = 999999
    Initialize $\log N(\boldsymbol{x}_t[:i], \boldsymbol{x}_0[:j]) = \text{-INF}, \forall i, j$
    Initialize $\log N(\boldsymbol{x}_t[:0], \boldsymbol{x}_0[:j]) = 0, \forall j$
    **for** $j = 1$ to $m$ **do**
        **for** $i = 1$ to $n$ **do**  // in parallel
            $\log N(\boldsymbol{x}_t[:i], \boldsymbol{x}_0[:j]) = \log \left\{ e^{\log N(\boldsymbol{x}_t[:i], \boldsymbol{x}_0[:j-1])} + e^{\log \delta(\boldsymbol{x}_t[i-1], \boldsymbol{x}_0[j-1]) + \log N(\boldsymbol{x}_t[:i-1], \boldsymbol{x}_0[:j-1])} \right\}$
        **end for**
    **end for**

---

**Algorithm 4** Suffix DP in log-domain

---

**Require:** original sequence $\boldsymbol{x}_0$ with length $m$, noised sequence $\boldsymbol{x}_t$ with length $n$, INF = 999999
    Initialize $\log N(\boldsymbol{x}_t[i:], \boldsymbol{x}_0[j:]) = \text{-INF}, \forall i, j$
    Initialize $\log N(\boldsymbol{x}_t[n:], \boldsymbol{x}_0[j:]) = 0, \forall j$
    **for** $j = m - 1$ to $0$ **do**
        **for** $i = n - 1$ to $0$ **do**  // in parallel
            $\log N(\boldsymbol{x}_t[i:], \boldsymbol{x}_0[j:]) = \log \left\{ e^{\log N(\boldsymbol{x}_t[i:], \boldsymbol{x}_0[j+1:])} + e^{\log \delta(\boldsymbol{x}_t[i], \boldsymbol{x}_0[j]) + \log N(\boldsymbol{x}_t[i+1:], \boldsymbol{x}_0[j+1:])} \right\}$
        **end for**
    **end for**

---

which can be efficiently implemented with the 'logaddexp' operation provided by deep learning frameworks such as PyTorch, the DP tables now store the logN values instead of N values in the original version, and lower data precision (e.g. float32) could be enabled to save memory. Notably, the log-domain DP algorithms will encounter the log0 issue, we replace log0 with a large negative number (-999999 in our implementation). The resulting numerical error of our log-domain DP is negligible, approximately on the order of $10^{-13}$ when using float64 and $10^{-4}$ with float32. However, the method fails to perform correctly with float16 and bfloat16 data types.

## E.2 DID TRAINING ALGORITHM

---

**Algorithm 5** DID Training

---

**Require:** Network $\bar{s}_\theta$, noise schedule $\sigma$, time $[0, 1]$, samples from data distribution $p_{\text{data}}$

 **repeat**

  $\boldsymbol{x}_0 \sim p_{\text{data}}, t \sim U([0, 1])$.

  Construct subsequence $\boldsymbol{x}_t$ by removing tokens with the probability of $1 - e^{-\bar{\sigma}(t)}$

  Calculate $N(\boldsymbol{x}_t[: i], \boldsymbol{x}_0[: j]), \forall i, j$ by prefix DP

  Calculate $N(\boldsymbol{x}_t[i :], \boldsymbol{x}_0[j :]), \forall i, j$ by suffix DP

  Calculate $N(\text{Ins}(\boldsymbol{x}_t, i, v), \boldsymbol{x}_0) = \sum_{j=1}^m \delta(\boldsymbol{x}_0[j], v) \cdot N(\boldsymbol{x}_t[: i], \boldsymbol{x}_0[: j - 1]) \cdot N(\boldsymbol{x}_t[i :], \boldsymbol{x}_0[j :]), \forall i, v$

  **if** train on fixed-length data using DICE loss Eq. 18 **then**

   Calculate $L_\theta(\boldsymbol{x}_t, \boldsymbol{x}_0) = \frac{\sigma(t)e^{-\bar{\sigma}(t)}}{1 - e^{-\bar{\sigma}(t)}} \sum_{i,v} \frac{N(\text{Ins}(\boldsymbol{x}_t, i, v), \boldsymbol{x}_0)}{N(\boldsymbol{x}_t, \boldsymbol{x}_0)} \left[ -\log \bar{s}_\theta(\boldsymbol{x}_t)[i, v] + C \right] \Big\}$

  **else if** train on variable-length data using DISE loss Eq. 12 **then**

   Calculate $L_\theta(\boldsymbol{x}_t, \boldsymbol{x}_0) = \frac{\sigma(t)e^{-\bar{\sigma}(t)}}{1 - e^{-\bar{\sigma}(t)}} \sum_{i,v} \left[ \bar{s}_\theta(\boldsymbol{x}_t, t)[i, v] - \frac{N(\text{Ins}(\boldsymbol{x}_t, i, v), \boldsymbol{x}_0)}{N(\boldsymbol{x}_t, \boldsymbol{x}_0)} \log \bar{s}_\theta(\boldsymbol{x}_t, t)[i, v] + C \right]$

  **end if**

  Calculate $\nabla_\theta L(\boldsymbol{x}_t, \boldsymbol{x}_0)$ and run optimizer

 **until** converged

---

## E.3 DID INFERENCE ALGORITHM

---

**Algorithm 6** DID Inference

---

**Require:** Network $\bar{s}_\theta$, noise schedule $\sigma$, time range $[0, 1]$, step size $\Delta t$

 $t \leftarrow 1, \boldsymbol{x}_t \leftarrow [\texttt{<BOS>}]$

 **while** $t > 0$ **do**

  Calculate $p_{t-\Delta t|t}^\theta((i, v)|\boldsymbol{x}_t) = \begin{cases} \frac{\sigma(t)e^{-\bar{\sigma}(t)}}{1 - e^{-\bar{\sigma}(t)}} \bar{s}_\theta(\boldsymbol{x}_t, t)[i, v]\Delta t, & v \neq \varnothing \\ 1 - \sum_{w \neq \varnothing} p_{t-\Delta t|t}^\theta((i, w)|\boldsymbol{x}_t), & v = \varnothing \end{cases}$

  Sample insertion actions $(i, v), \forall i$ based on $p_{t-\Delta t|t}^\theta((i, v)|\boldsymbol{x}_t)$

  Update insertion actions $(i, v), \forall i$ to construct $\boldsymbol{x}_{t-\Delta t}$

  $t \leftarrow t - \Delta t$

 **end while**

---

The algorithm above is for the unconditional generation of DID, for conditional generation, the difference is the probabilities for insertion actions in the middle of the prompt should be set to 0.

## E.4 PYTORCH IMPLEMENTATION OF THE DYNAMIC PROGRAMMING ALGORITHMS

```
LOG_ZERO=-999999
def safe_log(x, ):
    return torch.where(x == 0, LOG_ZERO, torch.log(x))

def get_N_ratio_logdomain(batch, remain_indices, seqlens, token_dim,
    sparse=True):
    # batched data alignment
    prefix_padded_xt = torch.zeros_like(batch, device=batch.device) - 1 #
     init as -1
    prefix_data_mask = seqlens[..., None] > torch.arange(batch.shape[1],
    device=batch.device)[None, ...]
    prefix_padded_xt[prefix_data_mask] = batch[remain_indices]
    prefix_si_eq_tj = (batch.unsqueeze(-1) == prefix_padded_xt.unsqueeze
    (-2))
    prefix_si_eq_tj_log = torch.log(prefix_si_eq_tj)

    suffix_padded_xt = torch.zeros_like(batch, device=batch.device) - 1 #
     init as -1
```

```python
    suffix_data_mask = seqlens[..., None] > torch.arange(batch.shape[1] -
     1, -1, -1, device=batch.device)[None, ...]
    suffix_padded_xt[suffix_data_mask] = batch[remain_indices]

    suffix_si_eq_tj_flipped = (torch.flip(batch, [1]).unsqueeze(-1) ==
    torch.flip(suffix_padded_xt, [1]).unsqueeze(-2))
    suffix_si_eq_tj_log_flipped = torch.log(suffix_si_eq_tj_flipped)

    B, S = batch.shape

    # prefix si_eq_tj and suffix si_eq_tj combined
    combined_eq = torch.stack([prefix_si_eq_tj_log,
    suffix_si_eq_tj_log_flipped], dim=-1).permute(1, 2, 3, 0) # (S, S, 2,
     B)

    # prefix dp and suffix dp combined
    combined_dp = torch.zeros(S+1, S+1, 2, B,  dtype=torch.float64,
    device=batch.device)  # (S+1, S+1, 2, B)
    combined_dp[:, 0] = 1
    combined_dp = safe_log(combined_dp)
    for i in range(1, S+1):
        prev = combined_dp[i-1]
        torch.logaddexp(prev[1:], combined_eq[i-1] + prev[:-1], out=
    combined_dp[i, 1:])

    prefix_dp, suffix_dp = combined_dp[:, :, 0].permute(2, 0, 1),
    combined_dp[:, :, 1].permute(2, 0, 1)
    suffix_dp = torch.flip(suffix_dp, [1, 2])

    # N(Ins(x_t, i, v), x_0) / N(x_t, x_0) prefix-suffix dp
    V = token_dim
    N_ratios = []

    for b in range(B):
        N = prefix_dp[b, -1, seqlens[b]]
        pr = prefix_dp[b, :-1, 1:seqlens[b] + 1]
        su = suffix_dp[b, 1:, S - seqlens[b] + 1:]
        pr_su = (pr + su - N).exp()

        if sparse: # sparse
            S, T = pr_su.shape
            rows = batch[b].unsqueeze(1).expand(S, T).reshape(-1)
            cols = torch.arange(T).to(batch.device).unsqueeze(0).expand(S
    , T).reshape(-1)
            values = pr_su.reshape(-1)

            mask = values.abs() >= 1e-6  # (S*T,) bool mask
            rows = rows[mask]
            cols = cols[mask]
            values = values[mask]

            indices = torch.stack([rows, cols], dim=0)
            N_ratio = torch.sparse_coo_tensor(indices, values, size=(V, T
    ))
            N_ratios.append(N_ratio)
        else: # dense
            N_ratio = torch.zeros((V, pr_su.size(1)), dtype=pr_su.dtype,
    device=batch.device) # (V, T)
            N_ratio.index_add_(0, batch[b], pr_su)
            N_ratios.append(N_ratio)

    packed_N_ratios = torch.cat(N_ratios, 1)

    if sparse: # sparse
        ret = packed_N_ratios.t().coalesce() # (\sum_b |x_t|_b, V)
```

```
    else: # dense
        ret = packed_N_ratios.t() # (\sum_b |x_t|_b, V)

    return ret
```

Listing 1: PyTorch source code for the computation of N-ratios $\frac{N(\mathrm{Ins}(\boldsymbol{x}_t,i,v),\boldsymbol{x}_0)}{N(\boldsymbol{x}_t,\boldsymbol{x}_0)}, \forall i, v$, as described in Sec. 3.4.

There are multiple tricks for the implementation of the subsequence counting DP algorithms to enhance the speed, for instance, 1) the prefix DP and suffix DP could be combined to improve the data parallelism, two DP tables could be updated in parallel; 2) the memory accessing could be contiguous to reduce IO cost by rearranging the axes of the DP tables; 3) in-place operation could be enabled to reduce IO cost.

# F    EXPERIMENTAL DETAILS

## F.1    FIXED-LENGTH TRAINING DETAILS

**Model architecture.** Following RADD, our models use an encoder-only transformer architecture with a dropout rate 0.02, rotary position embedding (Su et al., 2023), and untied word embeddings between input and output. Other model architecture hyperparameters are summarized in Tab. 7. The FFN dimension is $4\times$ hidden dimension.

| Size | Layers | Hidden Dimension | Attention Heads | Non-Embedding Parameters |
|------|--------|------------------|-----------------|--------------------------|
| Small | 12 | 768 | 12 | 85M |
| Medium | 24 | 1024 | 16 | 302M |
| Large | 36 | 1280 | 20 | 708M |

Table 7: Model architecture hyperparameters for difference model sizes of RADD and DID.

We use FlashAttention (Dao et al., 2022) to support attention computation for packed variable-length sequences, while RADD uses PyTorch's standard scaled dot product attention for its fixed-length batched inputs, which is also based on FlashAttention.

**Dataset and pre-processing.** For a fair comparison, we train DID on OpenWebText dataset (Gokaslan & Cohen, 2019) with the same steps (400K steps) or compute budget (800K steps) as RADD (Ou et al., 2025), under the log-linear noise schedule $\bar{\sigma}(t) = -\log(1-t)$, we train DID with a batch size 512, and a context length 1024. We tokenize OpenWebText dataset with the GPT2 (Radford et al., 2019) tokenizer, the vocabulary size is 50257. We follow the data pre-processing adopted in RADD, concatenating all sequences in the training dataset, then splitting it into chunks of fixed length 1024. We add a <BOS> special token to the first position of each chunk to model the insertion behavior before the first normal token by modeling the insertion after the <BOS> special token. We use the same token as <EOS> for <BOS>. We evaluate the zero-shot language modeling perplexity on WikiText, Lambada, Scientific Papers (Arxiv and Pubmed, abstract parts), AG News, LM1B, and PTB datasets (Merity et al., 2017; Paperno et al., 2016; Cohan et al., 2018; Zhang et al., 2015; Chelba et al., 2014; Marcus et al., 1993) in Tab. 1.

**Optimization.** For a fair comparison, we did not do a hyperparameter search, our optimization configuration strictly follows RADD, we use AdamW (Loshchilov & Hutter, 2019) optimizer with $\beta_1 = 0.9$, $\beta_2 = 0.999$, $\epsilon = 1$e-8, a weight decay rate of 0.03, a constant learning rate 3e-4 that linearly warmed up from 0 over the first 2500 steps, a gradient norm clipping value 1, an exponential moving average (EMA) with a decay rate 0.9999, and float16 is enabled for mixed-precision training. To avoid OOM errors during model training, we set the gradient accumulation steps to 1, 2, 2 for DID small, medium, and large; and 2, 4, 4 for RADD small, medium, and large.

### F.2 VARIABLE-LENGTH TRAINING DETAILS

**Model architecture.** Following ILM (Patel et al., 2025), we train a small model with 85M non-embedding parameters. Different from the fixed-length model, the variable-length model is not time-independent, i.e. we need to input the time information into the network. Therefore, we employ an adaptive layernorm for each transformer block as in the practice of DiT (Peebles & Xie, 2023) (as well as the time-dependent masked diffusion models like SEDD (Lou et al., 2024) and MDLM (Sahoo et al., 2024)), the condition embedding dimension is 128, which brings about extra parameters. To keep the total parameter amount of a transformer block unchanged, we reduce the FFN dimension to $3.5\times$ hidden dimension. Following ILM, we use a dropout rate of 0.1, rotary positional embedding, and untied word embeddings between input and output. FlashAttention is also employed for the efficient computation of variable-length data.

**Dataset and pre-processing.** Following ILM, we train variable-length models on the Stories dataset. We tokenize it with the Bert-Base-Uncased tokenizer, whose vocabulary size is 30522. Each sentence in the datasets is an individual training datapoint; hence, the training sequences are of variable lengths. The Stories dataset is truncated with a maximum context length of 1024. The batch size is 512. Data is also noised with a log-linear noise schedule in the diffusion forward process.

**Optimization.** Following the experiment settings of ILM, we use an AdamW optimizer with $\beta_1 = 0.9$, $\beta_2 = 0.999$, $\epsilon =$ 1e-8, a weight decay of 0.01 on all parameters (including biases and normalization layers), a constant learning rate 1e-4 that linearly warmed up from 0 over the first 1000 steps, a gradient norm clipping value 1, an exponential moving average (EMA) with a decay rate 0.9999, and bfloat16 is enabled for mixed-precision training. Stories dataset is trained for 60k steps. To avoid OOM errors during model training, we set the gradient accumulation steps to 1, 1, 1 for DID small, medium, and large; and 2, 4, 4 for RADD small, medium, and large.

### F.3 SAMPLING DETAILS

**Timestep discretization.** We use a standard uniform timestep discretizaition grid, i.e. $t(i) = i/N$, where $N$ is the total number of denoising steps in generation.

**The number of samples.** We evaluate the averaged generative perplexity (evaluated by GPT2 Large), unigram entropy, generation length, and inference time over 1024 samples with a batch size of 32.

**Data precision.** According to the precision issue of Gumbel-argmax sampling with float32 discussed in (Ou et al., 2025; Zheng et al., 2025), we use float64 precision for all generation tasks to ensure accurate categorical samplings.

### F.4 EVALUATION METRICS

**Zero-shot language modeling perplexity** is used to evaluate how well a model is trained, which uses the model to be evaluated to calculate the exponential of the average negative log-likelihood per token on datasets unseen by the model (i.e. zero-shot):

$$\text{PPL} = \exp\left(\mathbb{E}_{x \sim p_{\text{data}}(x)}\left[-\frac{\log p_\theta(x)}{L(x)}\right]\right), \tag{93}$$

where the likelihood can be exact (e.g. in auto-regressive models), or bounded (e.g. in diffusion models), and $L(x)$ is the length of $x$.

**Generative perplexity** is a widely used metric to evaluate the quality of model-generated text, whose definition is also as in Eq.93, the differences are $p_{\text{data}}$ is generated by the model to be evaluated, and $\theta$ is another off-the-shelf model to calculate the likelihood.

**Unigram entropy** is a metric to evaluate the diversity of model-generated text, which is based on the token occurrence frequency in a sentence. For a sentence $x$ with length $L$, the entropy is:

$$H = -\sum_{i=1}^{L} \frac{N(x_i, x)}{L(x)} \log_2 \frac{N(x_i, x)}{L(x)}, \tag{94}$$

where $N(x_i, x)$ is the occurrence time of token $x_i$ in sentence $x$, and $L(x)$ is the length of $x$.

## G   MORE EXPERIMENTAL RESULTS

### G.1   EFFICIENCY ANALYSIS OF THE DP ALGORITHMS IN DID TRAINING

To better demonstrate the efficiency of the DP algorithms (as described in Sec. 3.4) in DID training, we provide a more detailed profiling of DP. We report the DP time cost separately (Tab. 8,9) and investigate the scalability of DP to longer sequences (Tab. 10). Note that the goal of DP is to prepare the training target and its time cost is independent to the model size.

Table 8: DP efficiency analysis in the fixed-length training setting (Sec. 4.1). The overall training time and DP cost for 50 steps are reported (in seconds).

|         | Small        | Medium       | Large       |
|---------|--------------|--------------|-------------|
| DID     | 14.03        | 27.77        | 46.60       |
| DID-DP  | 2.38 (17.0%) | 3.33 (12.0%) | 3.35 (7.2%) |

As shown in Tab. 8, the medium and large models use more time for DP because the gradient accumulation is enabled in these settings (set to 2, as described in Appendix F), i.e. the same batch of data is divided into 2 parts, and DP is called twice, reducing the data parallelism and increasing the total time cost.

Table 9: DP efficiency analysis in the variable-length training setting (Sec. 4.2). The overall training time and DP cost for 50 steps are reported (in seconds).

|         | Small        | Medium       | Large        |
|---------|--------------|--------------|--------------|
| DID     | 7.71         | 12.30        | 19.83        |
| DID-DP  | 2.13 (27.6%) | 2.01 (16.3%) | 2.01 (10.1%) |

As shown in Tab. 9, the DP proportion is more than that in the fixed-length setting (Tab. 8), because in our implementation of DP (Appendix E.4), we pad all sequences to the maximum length 1024 to perform a batched DP, so the DP cost is similar to that in the fixed-length setting (Tab. 8), while the tokens input to the network are much fewer, since the average length of the Stories dataset is much shorter than 1024 used in the fixed-length setting. There are still inefficiencies in the current implementation of DP, and we leave a thorough system-level optimization to future work.

Furthermore, we analyze the time cost of DP for different sequence lengths. Tab. 10 demonstrates the

Table 10: The time cost of DP for different sequence lengths. In line with the training setting in Sec. 4, we set the batch size as 64, and sequence lengths ($S$) as 1k, 2k, 3k, and 4k. The time to perform 50 times of the DP algorithm to get the subsequence count ratios are reported (in seconds).

| Sequence Length ($S$) | 1k   | 2k   | 3k   | 4k   | The Power Law Exponent $a$ ($T \propto S^a$) |
|-----------------------|------|------|------|------|----------------------------------------------|
| DP Time ($T$)         | 2.05 | 3.64 | 8.07 | 11.2 | 1.26                                         |

efficiency of DP, the fitted power law exponent $a = 1.26 \in [1, 2]$ since the actual time complexity to update the 2-dimensional DP table is $O(S^2)$ and one dimension of the DP table could be updated in parallel.

### G.2   EFFICIENCY ANALYSIS OF DID INFERENCE

We more detailedly re-evaluate the inference efficiency of RADD and DID fixed-length models in Tab. 11 and variable-length models in Tab. 12. As discussed in (Zheng et al., 2025), neural function evaluation (NFE) and sampling are two main bottlenecks of DLM inference, we measure the costs of these two parts separately. For fixed-length models (Tab. 11), we observe that the NFE part is close to the theoretical $2\times$ speedup, since DID saves about half the FLOPs on <MASK> in the inference process compared to RADD. Notably, when the total denoising steps are fewer, the speedup of NFE

Table 11: Inference time comparison of fixed-length DID and RADD trained on OWT. Times are in seconds.

| Method | Steps | 16 | 32 | 64 | 128 | 256 | 512 | 1024 |
|--------|-------|-----|-----|-----|-----|-----|-----|------|
| RADD | Time | 0.240 | 0.334 | 0.507 | 0.872 | 1.627 | 2.876 | 4.506 |
| | NFE | 0.080 | 0.161 | 0.323 | 0.642 | 1.262 | 2.234 | 3.308 |
| | Sampling | 0.019 | 0.036 | 0.071 | 0.141 | 0.280 | 0.558 | 1.115 |
| DID | Time | **0.171** | **0.225** | **0.359** | **0.604** | **1.029** | **1.818** | **2.987** |
| | Speedup (Time) | 1.40× | 1.48× | 1.41× | 1.44× | 1.58× | 1.58× | 1.51× |
| | NFE | **0.029** | **0.076** | **0.159** | **0.323** | **0.641** | **1.131** | **1.712** |
| | Speedup (NFE) | 2.76× | 2.12× | 2.03× | 1.98× | 1.97× | 1.98× | 1.93× |
| | Sampling | 0.011 | 0.031 | 0.069 | 0.144 | 0.293 | 0.589 | 1.182 |

is more significant. This is because the maximum number of tokens that can be inserted in each step of DID inference cannot exceed the current length. When the total number of steps is small, the number of tokens decoded in the early steps will be bounded by this constraint, i.e., at step $i$, there are at most $2^{i-1}$ tokens input to the model, which brings greater NFE speedup of DID. Meanwhile, DID's performance is not affected by this reason. On the contrary, when the total number of steps is small, DID's decoding is much better than RADD's, as shown in Tab. 2.

Table 12: Inference time comparison of variable-length DID and padding-based RADD trained on Stories. Times are in seconds.

| Method | Steps | 64 | 128 | 256 | 512 |
|--------|-------|-----|-----|-----|-----|
| RADD | Time | 0.249 | 0.452 | 0.848 | 1.505 |
| | NFE | 0.161 | 0.321 | 0.631 | 1.118 |
| | Sampling | 0.043 | 0.085 | 0.170 | 0.340 |
| DID | Time | **0.089** | **0.135** | **0.217** | **0.384** |
| | Speedup (Time) | 2.78× | 3.35× | 3.91× | 3.92× |
| | NFE | **0.027** | **0.058** | **0.110** | **0.224** |
| | Speedup (NFE) | 5.96× | 5.53× | 5.74× | 4.99× |
| | Sampling | **0.013** | **0.027** | **0.058** | **0.110** |
| | Speedup (Sampling) | 3.30× | 3.15× | 2.93× | 3.09× |

For variable-length models (Tab. 12), both NFE and sampling are accelerated since DID does not need to compute on the <MASK> and <PAD> tokens.

### G.3 Ablation Study of Sequence-Level Normalization for Fixed-Length Models

We provide the ablation study of fixed-length models trained without the sequence-level normalization introduced in Sec. 3.5, i.e. trained with the DISE objective in Eq. 12 rather than the DICE objective in Eq. 18. We train FLOPs-aligned models of the small size, i.e. trained for 800K steps for DID models, and 400K steps for RADD. As shown in Tab. 13, the ablation version (DID-F w/o SeqNorm) exhibits an inferior performance compared to DID-F, demonstrating that utilizing the DICE objective for training could achieve an enhanced performance in the fixed-length setting. On the other hand, the ablation version (DID-F w/o SeqNorm) remains comparable to RADD, demonstrating the reasonability of the DISE objective.

Table 13: Ablation study of sequence-level normalization for fixed-length DID, the zero-shot language modeling perplexity on seven datasets are reported. Results for these diffusion models are perplexity upper bounds.

| Size | Method | WikiText | Lambada | Pubmed | AG News | LM1B | Arxiv | PTB |
|------|--------|----------|---------|--------|---------|------|-------|-----|
| Small | RADD | 38.27 | 51.82 | 56.99 | 73.18 | 72.99 | 85.95 | **108.79** |
| | DID-F w/o SeqNorm | 38.55 | 50.17 | 53.76 | 73.25 | 72.69 | 81.95 | 118.63 |
| | DID-F | **36.91** | **48.00** | **52.89** | **71.48** | **72.04** | **78.38** | 111.60 |

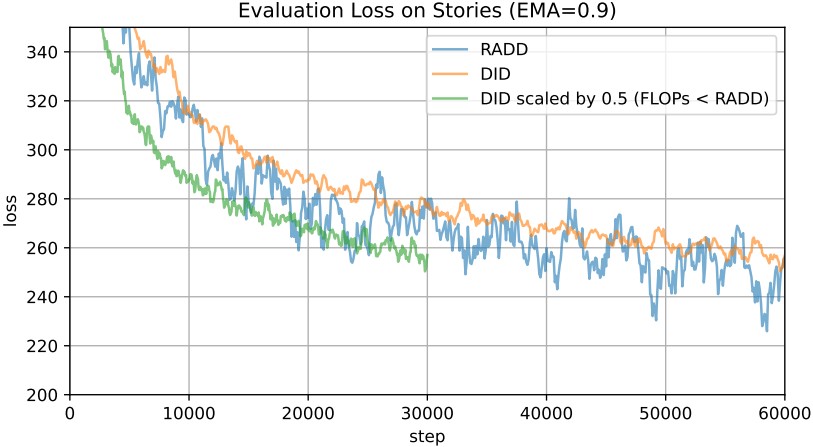

Figure 3: Evaluation loss curve on Stories dataset for RADD and DID in the variable-length setting.

## G.4 LANGUAGE MODELING PERFORMANCE FOR VARIABLE-LENGTH MODELS

We analyze the language modeling performance for variable-length models, as ILM does not have a likelihood-bounded training objective, only the language modeling performances of RADD and DID could be evaluated. As Stories is a specialized dataset, i.e. not so general like OpenWebText, we only show the evaluation loss curve on its own validation dataset in Fig. 3, where the curves for DID exhibit more stability than RADD's. Besides, we also report the curve scaled by 0.5, whose FLOPs are less than RADD (at the same x-coordinate) as the `<PAD>` and `<MASK>` used for RADD occupy more than 1/2 of the computational FLOPs in variable-length setting, this curve remains lower than the RADD curve, demonstrating the superiority of language modeling of DID over RADD in the variable-length setting.

## G.5 MORE INFERENCE STRATEGIES: COSINE TIMESTEP SCHEDULE

In addition to the uniform timestep schedule $T(i) = \frac{i}{N}$, where $N$ is the total number of denoising steps in generation, we evaluate a different inference strategy of cosine timestep schedule $T(i) = \cos(\frac{\pi}{2}(1 - \frac{i}{N}))$, whose timesteps are denser at the beginning of the reverse process (Shi et al., 2024). We evaluate the cosine timestep schedule using the fixed-length models trained on OWT in Tab. 14.

Table 14: Performance comparison with cosine timestep schedule on fixed-length models trained on OWT. Times are in seconds.

| Method | Steps | 16 | 32 | 64 | 128 | 256 | 512 | 1024 |
|---|---|---|---|---|---|---|---|---|
| RADD | PPL | 220.74 | 140.23 | 107.42 | 93.72 | **85.16** | 87.79 | **83.41** |
| | Entropy | 8.24 | 8.17 | 8.14 | 8.12 | 8.09 | 8.09 | 8.08 |
| | Time | 0.240 | 0.364 | 0.516 | 0.904 | 1.606 | 2.817 | 4.567 |
| | NFE | 0.081 | 0.160 | 0.315 | 0.612 | 1.156 | 2.022 | 3.065 |
| | Sampling | 0.023 | 0.046 | 0.090 | 0.178 | 0.356 | 0.708 | 1.414 |
| DID | PPL | **161.21** | **115.54** | **99.59** | **92.52** | 88.14 | **87.18** | 87.00 |
| | Entropy | 8.14 | 8.12 | 8.11 | 8.08 | 8.09 | 8.09 | 8.08 |
| | Time | **0.212** | **0.243** | **0.314** | **0.461** | **0.830** | **1.487** | **2.536** |
| | Speedup (Time) | 1.13× | 1.50× | 1.64× | 1.96× | 1.93× | 1.89× | 1.80× |
| | NFE | **0.028** | **0.064** | **0.134** | **0.257** | **0.514** | **0.955** | **1.579** |
| | Speedup (NFE) | 2.89× | 2.56× | 2.35× | 2.38× | 2.25× | 2.12× | 1.94× |
| | Sampling | **0.010** | **0.026** | **0.053** | **0.106** | **0.214** | **0.429** | **0.859** |
| | Speedup (Sampling) | 2.30× | 1.77× | 1.70× | 1.68× | 1.66× | 1.65× | 1.65× |

We observe that the speedup of the cosine schedule is more significant (up to $1.96\times$) because under the cosine schedule, the average sequence length processed by DID is shorter than that of the uniform schedule, i.e., more NFEs are used for shorter sequences in the early stage of generation, resulting in greater acceleration of DID inference.

### G.6 MORE INFERENCE STRATEGIES: NUCLEUS SAMPLING

We provide the evaluation of generation quality with nucleus sampling, a.k.a. top-$p$ sampling, with $p = 0.9$ in Tab. 15 for fixed-length models in Tab. 16 discussed in Sec. 4.1 and variable-length models discussed in Sec. 4.2.

Table 15: Nucleus sampling results for fixed-length models of RADD and DID trained on Open-WebText. Generative perplexity (PPL, evaluated by GPT2 Large), unigram entropy, and average generation length (for DID) under different denoising steps are reported.

| Method | Steps | 16 | 32 | 64 | 128 | 256 | 512 | 1024 |
|--------|-------|-----|-----|-----|-----|-----|-----|------|
| RADD | PPL | 95.55 | 55.21 | 40.38 | 34.22 | 31.79 | 30.31 | 30.51 |
| | Entropy | 7.99 | 7.89 | 7.82 | 7.77 | 7.70 | 7.70 | 7.69 |
| DID | PPL | **54.40** | **36.80** | **31.73** | **29.61** | **28.23** | **27.29** | **27.05** |
| | Entropy | 7.71 | 7.69 | 7.69 | 7.64 | 7.60 | 7.60 | 7.58 |
| | Length | 1021.88 | 1023.66 | 1024.02 | 1024.12 | 1023.95 | 1024.06 | 1024.12 |

As shown in Tab. 15, with nucleus sampling, DID generations exhibit both lower generative perplexity and lower diversity (measured by unigram entropy), demonstrating the annealing effect for DID is stronger than RADD.

Table 16: Nucleus sampling results for variable-length models of ILM, RADD and DID trained on Stories. Generative perplexity (PPL, evaluated by GPT2 Large), unigram entropy, and average generation length (for DID) under different denoising steps are reported. *: as outliers significantly affect PPL, only samples with PPL < 300 are counted.

| Method | Steps | 64 | 128 | 256 | 512 |
|--------|-------|-----|-----|-----|-----|
| ILM | PPL* | 174.29 | 27.04 | 14.50 | 15.31 |
| | Entropy | 5.02 | 5.38 | 5.44 | 5.45 |
| | Length | 62.68 | 110.05 | 120.75 | 122.99 |
| RADD | PPL* | 50.63 | 28.62 | 22.21 | 16.11 |
| | Entropy | 4.81 | 5.20 | 5.39 | 5.52 |
| | Length | 96.14 | 188.64 | 228.51 | 265.53 |
| DID | PPL | **12.33** | **12.74** | **12.46** | **12.83** |
| | Entropy | 5.69 | 5.72 | 5.73 | 5.69 |
| | Length | 161.64 | 171.91 | 179.62 | 174.39 |

As shown in Tab. 16, unlike the fixed-length models in Tab. 15, nucleus sampling results of DID consistently offer lower generative perplexity and higher diversity compared to ILM and RADD, further demonstrating the superiority of DID in the variable-length setting. Besides, the annealing effects could also be observed at the sentence-level, the generation results by nucleus sampling is shorter than those by direct sampling reported in Tab. 4.

### G.7 CONDITIONAL GENERATION

We evaluate the sample quality of conditional generation using 256 randomly selected OWT prefixes under different prompt lengths of 256, 512, and 768.

As shown in Tab. 17, we observe that DID stably achieves lower PPL with comparable unigram entropy (diversity) and generation length, demonstrating the stable superiority of DID in conditional generation performance over RADD.

Table 17: Conditional generation performance with different prompt lengths on OWT.

| Prompt Length | Method | Steps | 16 | 64 | 256 | 1024 |
|---|---|---|---|---|---|---|
| 256 | RADD | PPL | 114.46 | 67.13 | 58.33 | 55.16 |
| | | Entropy | 8.12 | 8.03 | 8.03 | 8.01 |
| | DID | PPL | **70.11** | **50.98** | **47.47** | **48.85** |
| | | Entropy | 7.92 | 7.94 | 7.92 | 7.92 |
| | | Length | 1022.48 | 1023.81 | 1023.88 | 1024.00 |
| 512 | RADD | PPL | 52.31 | 38.17 | 35.51 | 35.11 |
| | | Entropy | 8.00 | 7.95 | 7.95 | 7.96 |
| | DID | PPL | **35.29** | **29.95** | **29.19** | **29.00** |
| | | Entropy | 7.86 | 7.90 | 7.91 | 7.90 |
| | | Length | 1018.66 | 1023.08 | 1023.99 | 1024.07 |
| 768 | RADD | PPL | 25.85 | 22.57 | 21.92 | 21.80 |
| | | Entropy | 7.85 | 7.84 | 7.84 | 7.84 |
| | DID | PPL | **21.64** | **20.25** | **20.10** | **19.99** |
| | | Entropy | 7.83 | 7.86 | 7.87 | 7.86 |
| | | Length | 1011.45 | 1023.38 | 1023.93 | 1024.10 |

Table 18: Ablation study of padding length for RADD in the variable-length setting. Models are trained on Stories for 60K steps. Generative perplexity (PPL, evaluated by GPT2 Large), unigram entropy, inference time (in seconds), and average generation length under different denoising steps are reported. *: as outliers significantly affect PPL, only samples with PPL $< 300$ are counted.

| Method | Steps | 64 | 128 | 256 | 512 |
|---|---|---|---|---|---|
| RADD-512 | PPL* | 80.19 | 57.32 | 40.27 | 35.45 |
| | Entropy | 4.89 | 5.21 | 5.56 | 5.63 |
| | Time (s) | 0.123 | 0.221 | 0.383 | 0.608 |
| | Length | 91.83 | 138.70 | 191.43 | 207.93 |
| RADD-1024 | PPL* | 81.92 | 50.89 | 34.47 | 26.78 |
| | Entropy | 5.22 | 5.58 | 5.79 | 5.85 |
| | Time (s) | 0.246 | 0.441 | 0.827 | 1.461 |
| | Length | 110.66 | 200.73 | 349.54 | 353.47 |
| DID | PPL | **22.78** | **21.07** | **21.90** | **23.88** |
| | Entropy | 5.90 | 5.94 | 5.94 | 5.94 |
| | Time (s) | **0.090** | **0.132** | **0.218** | **0.388** |
| | Length | 182.31 | 193.77 | 202.97 | 204.96 |

## G.8 COMPARISONS WITH RADD OF DIFFERENT PADDING LENGTHS

As described in Sec. 4.2, the padding length of RADD in the variable-length setting is a hyperparameter to be pre-defined, which is a complexity for MDM in the variable-length setting, as well as a source of its inefficiency of <PAD> computation. Here we provide another setting of padding length, a shorter one of 512, to train RADD for the same steps (60K) on Stories dataset, and evaluate its generation quality and speed, the cumulative distribution functions (CDFs) of generation length for different models under different total denoising steps are also shown in Fig. 4, alike the experiments in Sec. 4.2.

As shown in Tab. 18, the generation quality of RADD with padding length 512 exhibits observable degradation, i.e. higher generative perplexity and lower diversity (measured by unigram entropy), compared to RADD with padding length 1024, demonstrating the reasonability of setting the padding length as 1024, which is also ILM's original training configuration, even though it is highly inefficient as the average length of Stories training dataset is only 213.43, much shorter than 1024, leading to

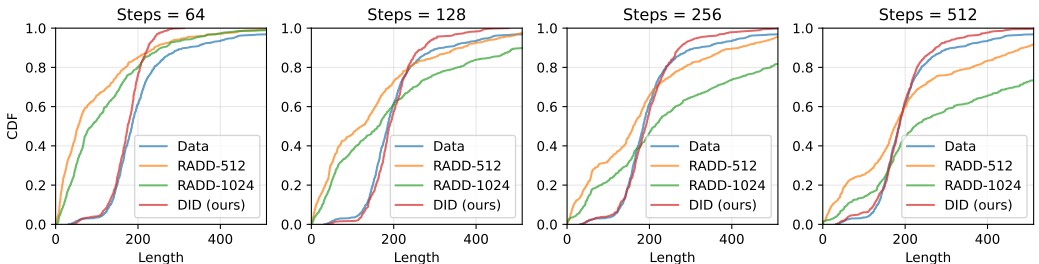

Figure 4: Cumulative distribution functions (CDFs) of generation length under different total denoising steps.

more computational cost for the <PAD> tokens. On the other hand, RADD with padding length 512 achieves a speedup $\sim 2\times$ RADD with padding length 1024, yet still $\sim 1.59\times$ slower than DID on average.

Regarding length modeling, as shown in Fig. 4, although the average generation length of RADD-512 in Tab. 18 is closer to the ground truth (213.43) than RADD-1024, the CDFs in Fig. 4 still show a strong deviation to the ground truth while DID achieves much closer CDFs, which indicates the superiority of DID over MDMs for the variable-length generation task.

### G.9 EVALUATION WITH MORE CHECKPOINTS

We have evaluated DID with more checkpoints to locate where the improvements originate. We additionally evaluate the zero-shot perplexity (Tab. 1) using more checkpoints (200k for 25% FLOPs and 600k for 75% FLOPs), together with the experiments in Tab. 20, our conclusion is: on average, DID could surpass MDM, on the 7 datasets for zero-shot perplexity evaluation (Tab. 19) and 8 tasks for commonsense reasoning evaluation (Tab. 20), with 50% to 75% training FLOPs.

Table 19: Zero-shot language modeling perplexity evaluation with more checkpoints.

| Size | Method | WikiText | Lambada | Pubmed | AG News | LM1B | Arxiv | PTB | Average |
|------|--------|----------|---------|--------|---------|------|-------|-----|---------|
| Small | RADD-400k | 38.27 | 51.82 | 56.99 | 73.18 | 72.99 | 85.95 | 108.79 | 69.71 |
| | DID-200k | 40.50 | 50.55 | 57.87 | 81.02 | 76.37 | 85.86 | 122.84 | 73.57 |
| | DID-400k | 38.72 | 49.10 | 55.02 | 76.02 | 74.04 | 82.41 | 115.37 | 70.10 |
| | DID-600k | 37.41 | 48.40 | 53.59 | 72.88 | 72.64 | 79.94 | 112.95 | **68.26** |
| | DID-800k | 36.91 | 48.00 | 52.89 | 71.48 | 72.04 | 78.38 | 111.60 | **67.33** |
| Medium | RADD-400k | 28.44 | 44.10 | 41.06 | 48.96 | 60.32 | 66.28 | 81.05 | 52.89 |
| | DID-200k | 31.11 | 43.69 | 44.36 | 57.60 | 62.74 | 67.80 | 97.26 | 57.79 |
| | DID-400k | 29.19 | 41.94 | 40.84 | 52.53 | 59.88 | 63.95 | 91.87 | 54.31 |
| | DID-600k | 28.48 | 41.30 | 39.44 | 49.80 | 58.76 | 62.54 | 85.98 | **52.33** |
| | DID-800k | 28.35 | 41.00 | 38.71 | 48.84 | 58.05 | 61.77 | 87.09 | **51.97** |

### G.10 SCALABILITY

Regarding the scalability to larger models and longer sequences, following SMDM (Nie et al., 2024), we scale up the DID variable-length model to 1.1B parameters and train it on the SlimPajama dataset (Soboleva et al., 2023) for 1.6e21 FLOPs, each document in the dataset is an individual variable-length sequence with a truncation length of 2048 (trained with the log-domain DP algorithm for longer sequences described in Appendix E), and evaluate the accuracy (%) of DID and SMDM on downstream commonsense reasoning tasks (Arc (Clark et al., 2018), Hellaswag (Zellers et al., 2019), Obqa (Mihaylov et al., 2018), Piqa (Bisk et al., 2020), Race (Lai et al., 2017), Siqa (Sap et al., 2019), and Winogrande (Sakaguchi et al., 2019)) with the LM-Evaluation-Harness framework (Gao et al., 2023). SMDM results are reproduced with its open-sourced checkpoint, with $< 1\%$ FLOPs trained on variable-length data.

We further evaluate the performance of DID on the GSM8K (Grade School Math 8K) (Cobbe et al., 2021) benchmark for math word reasoning problems, using our 1.1B variable-length model.

Table 20: Downstream task evaluation of 1.1B models. Zero-shot accuracy (%) is reported for each benchmark.

| Method (FLOPs) | Arc-c | Arc-e | Hellaswag | Obqa | Piqa | Race | Siqa | Wino | Average |
|---|---|---|---|---|---|---|---|---|---|
| SMDM (1.6e21) | 24.57 | 48.91 | 44.37 | 31.20 | 65.23 | 33.78 | 39.00 | 52.88 | 42.49 |
| DID (8e20) | 25.17 | 47.73 | 40.79 | 31.20 | 64.58 | 31.48 | 38.33 | 51.62 | 41.36 |
| DID (1.2e21) | 26.11 | 49.20 | 43.98 | 32.80 | 65.72 | 32.06 | 37.97 | 54.62 | **42.81** |
| DID (1.6e21) | 26.54 | 49.37 | 45.72 | 32.00 | 66.16 | 32.44 | 38.89 | 54.85 | **43.25** |

Table 21: GSM8K evaluation under different top-$p$ sampling strategies. Zero-shot accuracy (%) is reported.

| **Top-$p$ strategy** | Steps | 8 | 16 | 32 | 64 | 128 | 256 |
|---|---|---|---|---|---|---|---|
| $p = 1.0$ (direct sampling) | SMDM | 27.82 | 33.97 | 35.78 | 36.85 | 36.54 | 36.69 |
| | DID | **31.92** | **35.41** | **37.45** | **39.35** | **39.27** | **39.88** |
| $p = 0.9$ | SMDM | 32.22 | 36.92 | 37.38 | 39.58 | 39.58 | 39.65 |
| | DID | **32.37** | **38.51** | **42.38** | **41.55** | **42.23** | **43.75** |
| $p = 0.6$ | SMDM | 36.01 | 40.03 | 40.56 | 42.00 | 43.37 | 42.61 |
| | DID | **38.82** | **41.39** | **43.90** | **45.26** | **46.47** | **46.70** |
| $p = 0.3$ | SMDM | 37.83 | 39.88 | 42.23 | 43.82 | 44.73 | 43.37 |
| | DID | **38.89** | **42.08** | **43.21** | **44.81** | **45.49** | **45.79** |

Following the setup in SMDM (Nie et al., 2024), we fine-tune the models pretrained on SlimPajama for 1.6e21 FLOPs (both SMDM-1.1B and DID-1.1B), on the augmented training data (Deng et al., 2023) for 40 epochs, keeping all optimization configurations strictly consistent. We then assess the zero-shot accuracy (%) of DID and SMDM under various inference settings, including different decoding steps and top-$p$ sampling strategies, using the standard LM-Evaluation-Harness (Gao et al., 2023) framework.

As shown in Tab. 21, results demonstrate the consistent superiority of our variable-length DID model over the fixed-length, padding-based SMDM approach on the conditional generation task of the GSM8K benchmark.

# H  GENERATION EXAMPLES

Here we provide generation examples of the fixed-length model trained on OpenWebText in Appendix H.1 and variable-length model trained on Stories in Appendix H.2. Besides, we also demonstrate the intermediate generation process in Appendix H.3. All samples are generated by the direct sampling algorithm under the float point precision of fp64 for accurate categorical sampling.

## H.1  SAMPLES GENERATED BY THE FIXED-LENGTH MODEL TRAINED ON OPENWEBTEXT

### H.1.1  UNCONDITIONAL GENERATION

============================== Sample 1 ==============================

Moon Environment, NEONSEA, CANANA INTERACTIVES,SEVERTY Union ENTRY,PORTS IN DEMO LEAC,,SQUANTIFOCUS, CAZARY, and WACKUP. COvenATION BY BAR WAS PURPOSE TO TEAM DEFENSE, DEFENSE IS THE CHART OF OUTER AND NEGATIVE AND IRONSIDE.

*Selected for Hexagonal Player counting)(Best players from THREE EXPLicates.)  of Medals PT50 showed signs of PT in US Switch Ball, SH DROPIONS DOWN HUMAN RIGHTS, CT PORTS were only available for Queensland Olympics USA PAT NACHINO FUTURE Videos reserved. The PACI did mean that, in every four teams from these groups UNMERCHANT WEIGHT

CARRYING THE LED ARTS OF BROADCASTING SO NECESSARY DOES, ORBURNVE OFF FOVPORT.ProtarpYSCHIETICS COMICS.COM WILL END CONSENTSLY.

CLICK HERE TO SEE RIGDINS OF AUSTRALIAN BUTTE SON BASING<|endoftext|>by Jay R. Tyardots in California Scientist, January 3, http://www.chronicle.com/news/science-technology/124428.2938drozier55.htm

Engineers working on creating a quantum computing field say they've made balls of quantum information that is able to vibrate like graphene by adjusting its signals like a natural quantum bus. The result is a powerful 250-watt device that could be used for computing purposes within a decade after showing the circuitry could be tuned over time.

The research is in IEEE's Physical Review Letters, Designing the World's First Atom of Silicon Nanotubes.

Better than the current solid-state transistor of opposable states, silicon nanotubes that slip into a hurdle metal sandwich (DA50N2) create FCU.

The non-energetic bits could be on their way into next-generation "intense" processors and even be used in quantum computing.

The researchers said

"The circuit design delivered a new transistor of exponentially lower power," said Enrique Cameron, a nitrogen-dot tech at Rice University in Houston, Texas.

The transistors are integrated with what the researchers call quantum control an internal signal source designed to tell the circuit of the transistor it wants the mechanism to act on. If a magnetic or electrostatic oscillation is expected or unwanted, controlling the quantum control system. The internal signal also controls how much current can be drawn.

Researchers have showed quantum computing in a 256-nd transistor with a count of DSNO, but more research needed increase the size and lithography of the chip even higher.

For the first study for such a transistor done in the same environment, researchers used a unique voltage coupler device. The voltage coupler holds the value of a sine wave and information about the direction of voltage and current.

To output a voltage the computer acts on the sum of the voltage and current which is what people might get if divided a computer transfer rate of 16 Gbit/s (3.24 GB per 100 processors) into half.

According to the group's preliminary result their signals-based feeding strategy, which has also found a way to allow a smartphone to create its own elemental energy, could be widely used at lower power levels. If meant then anything involving new electronics is possible as well.

"The speed in which this sort of mechanism lets off electrons through the direction could have really interesting legs," said Dan Schaberer, a professor in the physics department at Princeton who led the design but who didn't publish a copy of the paper.

The use of a dielectric drive system oscillating the same spirals the actuators provides more mechanical control.

"This is analogous to acceleration using laser in phased scanning microscopy," Cameron said.

Naturally it may be difficult to get like that, but the MIT team said it was using some rough theoretical calculations in the paper and details of the initial version have been tweaked within the theory it is possible.

Guy Lee, a different team at the Technology University Berlin, who's the scientist who developed the device, said "for what most people would call a rapid development challenge team's concept is utterly unique."

He added the team is interested in using future flash-memory chips, for example, as a sufficient unit of mass for quantum logic converters.

"Once quantum bits are developed as nanocomposites and lithography problems solved, then I will not be surprised to see all the fantastic uses come up," Cameron said.<|endoftext|>Here are some Bitcoin events and meetups we were hoping you found important information about and why.

Lindsay Reslove is a research analyst with Coin Private Wealth Research. She is a student in the MSU's departments of business administration and computer science. She and John King, a business in Flint, Mich

============================== Sample 2 ==============================

barbecues, unstudded haircuts with a passing connection to classical musicians and singers, African stars Bakawisi Sim and Kutimbla Grime. In Case of The Grace of Contessa, the speaker describes Chichamaster Rupert Brecht's interpretation of the Chop Talk, up close to Hoop Orchestra and the Brooklyn Hall Chop bust. The Volk Attendant is dramatologist Philip Hill's take on exploring Bartoz's interpretation of The Young God (Directed by Geraeus Castoré); Penelope Vernon's superb books on Mijunset's Frativity, the Howeñal movement and other forms of Esperanto music are only one of many created in the United States. Her favorites include the e. Additions Games (1811), The Anomal Valley Merry Stories (1815), and the Nieos family Magazine (1856).

The Continental Brand, the dramatist Herbert's Own Dame Vis, introduces English sleuthmuths Ian Woodward and Charles Feching, and doesnt yourself to the first meeting of witty, energetic and well-traveled Spioclasts from San Diego. San Diego the Altun, inspired by a series of articles by Santiago Contamas and José de Septé, the American Innerts, reflecting the peak of the Esperanzaism American Renaissance, tells from the air of Sclepre's Esperanto sophisticate Avarula Grothomo, a desperate person suffering from fibresesticular use (as in the processing of rhubarb). This city may have gone to war or appeared beneath man's feet. Domaniic unic is not the only thing involving Esperanto; the Atl-mas have described a plague. But apparently that never existed.<|endoftext|>Sangefeng: The Yoruba businessman whose family and children lived in a small sarandi (boat) before the Typhoon Haiyan destroyed the last homes there is thought to have been safe in the Philippines for at least next few weeks. Philippines Home Minister, Peter Bola, told on Monday that authorities had discovered a connection between Mr Shinse and the latest Islamic anti-American manifesto during a vitriolic sermon he gave last Sunday at a funeral.

Judging from the "manifesto", Mr Shinse fits the double-standard of the dodgy wealthy Chinese lawyer, where his son had an affair.

His son said he is seen as a "gynoid, disabled in body, deracinated". He said he is mentally unstable. "But that does not mean I'm happy."

He was approached this week by a Haiyan victim. "I broke it getting in the plane, but I said I thought it might be good to find it," he said. This man said that the skull bones in his basement had been removed and awaiting identification.

Throughout last week and a half with displaced people his nephew has approached outside his office to identify themselves. He recently helped organise seminars of the Islamic branch in the Philippines.

"I keep everything on tapes," he said. "Every morning I have to stuff it out the window in front of the TV screen, I can carry it anymore."<|endoftext|>Three hundred revolutionary followers, were took part in Guatemala's Bernardo (Barefoot) alongside 50 Venezuelans, 20 Ecuadoran asylum seekers, to celebrate the revolution, giving a talk-outout to the US Presidential candidate Sanders for expressing "cynicism" about a Socialist state.

"This shows that democracy is not so bad in the world," Patrick Dubo, one of the Trotskyists from the Unido Brigda Party said during the rallies.Some of the crowd continued to protest for Che Guevara and Fidel Castro, as they claimed that they supported "separatism".

"In the internet came a Facebook group which is affirmed to be an alternative Socialist state which fully affirms the concept of "Apartheid" one unnamed person told the press.

Dubo said that they were representatives of anti-socialist Christians which promote Marxism and fight immorality through government policies. He added that socialism has never really disappeared in South America because its ideas are with more with "fleat-blading".On the other hand self-styled "centrist businessman" Ric Williams fully supported the courage despite this by arguing that expressed the spirit of patriotism.

Reporting Daniel Jenks, Elias Osauga, Thompson Salazar and Guy Aveculs

...<|endoftext|>This week, Dow Chemical Co. CEO George Lopulos finally gave the company's 3,000 employees fresh concessions that break codes and specified overhaul of operations. By Jan. 25, they were signed a tentative contract that would represent more than a third of some 5,000 workers.But while the original tentative agreement was set for the end of this month,

============================== Sample 3 ==============================

"It came from someone who felt the urgency of it and refreshed us."

The idea of an airport being a catalyst to switch venues appeared at the time. The airport has been a home for a decade, with facilities real and perceived to be as nice as they come. Boston has a few Green Card 250, excellent, venues. It is not the site of 60 such ad golf-related contracts according to a report from the AP. Taunton was affected by that, as was the clutch conversion on his Jan. 10, 2014 road trip that made national TV headlines.

He slid perfectly from outside the Patriots' six-yard line and 25 yards out before flipping it over Willy Chopade Jr.'s end zone. This kept him on the field with Patriotsmate Moss.

"We were hoping we were still going to be playing because of Moss," Haley said. "We just came to grips."

Boston has shown a growing appreciation of Shifkar's vision, and now he has the bang of a stadium it will live for. After departure of Kow, its most active sponsors, the Patriots purchased HSBC. The successful discussion of other offers and candidates included Shad Khan, a partner who does global retail on the West End of Oxford.

"We were hoping to go to Fenway and put an identity more behind the location," Shikbaram said. "We are excited about what we can do under the roof but and that's part of the truly team experience. Even before we got the bowl, this is a sport that runs. Fenway Park is a great atmosphere for a guy on NFL foot."

Haley said he can harness that momentum for 2017 based on the gambling on his offensive outlook.

"The other guys are going to push us," Haley said he and his Patriots will return championship opportunities. "I said to the fans it'll remind us of being among the greats and being the champs and champion. They are going to be the lynchpin in this Big Game. And it's not because they live here, but because they're going to be the stick. Meet them, run them out of sight. To have people, that's not something that other teams have done in NFL history."

We covered how the Patriots approached January's New York vacation trip at The Syllabus.com and their spring season traveled since June. Experience our very unique 2015 Mobile Guide at http://quotepan.com .

Related<|endoftext|>A more detailed revelation of Viking's dramatic rise is now available. But The Mammals tended to rely on land expeditions to glean information. Now that more information about the world without computers or maps could accessed.

Wendy Robinson, The Arctic Lion

New Viking Denials Confirmed by Areologists

A underwater trench was once paved off off the UK coast in Viking engineers' location claimed by locals. But drilling inside the channel, sealed for a year by the distortion of the erupted Swarbrough volcano, one in November discovered 8 otheral signed of submarine submarines working against the foraminiferous seawater, accessing new information about its tunnel depths and carbon sources.

Throughout the last century leading officials swore the Viking civilisation was dead, but now archaeologists finally know why.

Adrian Ashleyman, minister in charge of the Western Isles, covers every mile of the Viking civilization for thousands of years up until the 10th century, in a presentation at the Ocean Foundation. "It's amazing how much they travel and the work has been done to get where they are, but it's pretty awful to pull nails in a far schedule. The lines need to be so easy to get across"

Click see the BBC report.

Related myths: Viking rune, USSR, Viking generals<|endoftext|>Because former whistle-blower, Edward Snowden has sent journalists and world bloggers cocktail clippings of a New York-based government security training, Mounties Sustainability Training taught by British Commando Troopers, it's an excellent venue for journalists to get their information.

We hear that the U.S. may have bit of a rock on the memory of its golden-west. Snowden has succeeded in making the FSA comfortable shedding its hard currency, boasting to the Guardian recently that the store in which he formed his fortune in Iceland was valued at a staggering $256.5 million.$

It's no different to Volkswagen World Group's overpriced program and supermarket boss, Valentino Rossi's private company which has been working on technology that can be based on Italian 500's Series 7's.

This thinking causes problems. Companies in Germany and Austria license special patents and sell them only to companies that have a record of

=============================== Sample 4 ===============================

its proposed rules were filed. Seeking to amend the proposed rule, FERC viewed the strong language put forth by an association opposed to the exemption when it met with the statement in a letter from Dan Stern with the Sierra Club. Moreover, the National Farm Bureau and the Federation of Sportsmen and Fishermen, which owns lands for conservation, both backed the new "no-Mine" rules. But FERC didn't allow for this. "We had to try to craft an artificial, complicated substantive rule where power was responsible for conservation that the industry doesn't want at all," Butler says.

The new RRE rules leave farmers with only a title to stake each acre of unused land to mine copper. Buildings must be fewer than 10 feet tall, and need to fitted to a VVAC pole to allow the same air flow. No more than 25 acres are permitted and a 30-acre farm can be mined using hydraulic fracturing. They have to prevent the potential mining operation, built a fence, took part in field tests, and gathered evidence from the site of having found copper that has trampled on production lines.

FERC did not dispute rule specifics—no specific copper facilities the no-mine would be laid—but did release an assessment saying that public processes would determine "the cutting and the use of pipes" and just what amount of public land to be included in maps should be limited. "It's all on a market herrhorse," Slaughter says. "It's a de facto emissions test."

Nutter warned that the regulations might actually result in "a more anti-competitive system" than one that does allow for polluting, subdivision and enforce strong rules. "The industry does not see it right and rejects the idea that it's worse because it increases pollution," he says. "The Public Works Committee would find it even worse."

Public approval for a draft rule will reportedly take 12-16 months as well as legal challenges.<|endoftext|>A coalition of ex-elites gathered at Laurelview High School this past week. They're part of the Coalition (recent graduates and teachers from urban county were touted for tenure) then showed them how to fight neighborhood segregation. They are so-called Chiefs in the program, and they recruit more black kids through. They simple this practice that is commonplace in large cities and observed pro-Confederate celebrations.

I joined Michael Smart, coordinator of that program, at his rock concert, Rebel Wars, and watched a video of Confederate flag shootouts outside the program.

The group of eight black students lectured and asked teacher Vincent Henderson about characteristics of black communities.

Potish, "Most of us were in tears," following decades of history.

"Then we were surprised," Henderson said. "We were almost sure we had it with us."

"We were thinking about it," Smart said. "We just weren't expecting it."

"The CMO brought us back to races of neighborhoods that are African American. You're talking about lead epidemic and youth unemployment numbers," Smart continued. "You've reported a dramatic employment loss on top of low personal drug using. And you're gonna see a lot of student prejudice and crime in the process."

Though they only a few seconds to tap into the students' attitude with statements like "Motivation, Director of Excellence," officials did acknowledge the past positive attitudes and report they were raising them.

I heard much more Cal sentiments from the rest of the presentation, albeit somewhat-coarse ones.

"They did not apologize for classroom clashes, not even the token degree," Smart told me. "We're not converts."

They made light of the school's efforts in Combat Simulator, a battlefront that drew resentment after the Burbank response of black students.

"They want to know about our diversity program," Principal Frank Fund interrupted. "They make assumptions about who wants them to live and police where they want them to live."

"We actually want kids like these," Fund said. "Eventually, that'll be the biggest part of it."

Historically, many in Chicago's gated neighborhoods have been little but crippled. Since 1992, a handful of private universities where black leads have developed a detrimental regime where they send low-income students from the disadvantaged through Northern schools to improve opportunities for minorities.

For a rural district full of Latino immigrants, these black-only programs have Tucson's black students underperforming at disproportionate rates to kids from whites, Hispanics, and other groups. The effects are little-noticed, and rarely openly presented.

Jew's album, Jazz is now played widely, performing nationwide and last year winning an award in collaboration with the Sierra Club, and the Parents Association of Arizona. They call on the district to address low rates of distance and point to the research that comes out against Unified.

"We're putting black children at a disadvantage," says Jew, who has added Jazz to Spotify himself. "

============================== Sample 5 ==============================

Dr, S Scott Road, LPM, NY 57025 Hours: 7P.M. – 9 PM (10 a.m service; 50 foot Zion TeaPot; chef Beleo, LPMO, chefbeleo.com); Dinner at 7 p.m Tickets online H22D through tickets [404] 775-624-3404 ; public (202) 769-6678.

Website, ChefBalleo.com

Line 4665/4667<|endoftext|>Accelerated around A number of doors with bays to divert between two types of facility. One venue houses the "Em Lion" training and the second one for Saturday night gatherings. Event tickets will be given out hand handed with or without ID to pay. The vast majority of event will be inside the Center for people to catch up with our manager and owner free ops. For partygoers there will be entertainment at every level from the alley and sliding ramp to hooters skateboarding, and pools.

POLRY days temporary weekends and dynamic peoples schedules

Saturday band on all day never has before. Neither school nor college

Must use good for jams between the residence drive to police

Only vintage, topping, tattooing (except for Halloween) and older patrons update our article in two weeks

Place Events

Chocolate: 6500-10000 people coming to town Night manager is on 24 hours and has control of the event (transportation times, gives permits to Huppi-Pickers, funding license to city agency to claim, and team with the owner and operator of the business] 300 people Total event is double 300 Children Lengths must be > 30 vs old chance

Music will be live hand tapping (as will Ad and Staging throughout the building)

Special ticket bidding "Cool City" ends Jan 19 which will last the day after.

Registration 5 slots as long as per picture the touch machines using 5 keys Trans-card from Ticket-master will fund rates Food Trucks/ door.

The event is after the princess's reception is in each auditorium featuring live music, jazz, and theater as the walk out to athlete camp commencement rehearsals before night. There will be separate event for kids to get to these and other things for the kids to meet the party.

Tickets

Pick up on the private chapel events etc. these are not major facilities. Last come first serve

Tickets: $10 or less + fees$

Believe that the jQuery art space is a space that is always open

Saturday operating license called business day or city license

No license $95 treat and food service$

Entertain to the Saturday breakfast event

Featuring a dinner atmosphere at most events.

"Lovers in Trust" Auction designed to get kids and people from the neighborhood get to know some other

Cent hours chance to win charity items

hold a laptop and tablet inside during the Emperor's Lion Training session.

All photos have been stolen from the art event

Security department approved for benefits only.

Notes

Lowers: $50 Token$

(From @shermark)

4.Mairesomata,

Here is the first image of some of our staff that have gotten hurt in this space | Pic (from @Dunader)

How to comment well from the months in this space based on being able experienced in so much politicotation

5. Letters

Here is a season 15 video of Elsa on Lierson handled this space and what is happening, how to congratulate staffers (see here for first time public announcement October 2005), how to give free tickets to the prom.<|endoftext|>This spring, meet at a Chipotle University of Iowa, Ames location, representatives of the Chipotle's signature chicken giant said.

An event has been put together by the Native American Union for FEED Friday from 9-11 p.m. Details on exact locations for the event weren't available.

Si que héres Inigo, SE ANNOCALE CHICK 10 – Inigo for the 27th date on May 7th!! — Equalization (@Cornish Alliance  Equalization) March 27, 2016

Click here for onscreen printable image of the location – 700 Mr. Petty Avenue.

Privacy

In the release, which had more details on Chipotle, the University's College of Agriculture and Economics said, "Georgetown and its partner specializes in protein, pirouin, cellophane, and tinder-minder products."

Meanwhile, more past Chipotle showings:

LUS / GRODYS Club

LatINO STARS Launchday starts at 2 a.m. Friday when Las Vegas's Graduation Day Lounge will be on hand (bar not applicable)

9 p.m. on Friday, April 7.

Shillam

### H.1.2 CONDITIONAL GENERATION

Here we provide conditional generation examples generated by DID trained on OpenWebText, we set prompt lengths as 256, 512, and 768, and the prompt parts are colored in blue.

=========================== Prompt Length 256 =============================
with the Hawks with a view to identifying why, as they say in the song, Hawthorn is such a happy team. Which raises the question: Are the Hawthorn players happy because they have won the past two premierships or is Hawthorn winning more because their players are happy? Or, as the Hawks themselves suspect, perhaps the answer lies between. Not only does that club feel validated in their work practices, which this season have included an unscheduled day off after their first Launceston game, but it could also use its position at the top of the off-field ladder as it works behind the scenes to mount a case against the AFL's move in taxing clubs' football department spending. That new tax will be reviewed before the end of next season and Hawthorn are looking to lobby for the exclusion of such welfare initiatives such as the regular monitoring of every player's mental health by its sport psychologist from the new tax. Another area the Hawks believe should be exempt is the sponsoring of international educational trips for its staff. Hawthorn is not the only team that claims to be focusing on better work-life balance for their players but they have worked more diligently to identify red flags among their team since Travis Tuck and his mental-health issues became front-page news. Originally written by the Herald Sun on Friday night, after a sustained discussion of players' struggles and the potential issues associated with their mental health, Tuck's story has been arguably the story that has generated most interest in the club-induced Australian football story. The responses from sceptical media outlets are mixed, which is through no fault of some of those who have the hardest situation to complain about. The story may need more interpretation, though, as the AFL already has its own legal team.<|endoftext|>Throwback, released in November 2011, makes it clear what club toolsbrew can do for your body. The key thing is its personal kryptonite. Though Artur Kovács shares "late almonds" with his fellow Olympic teams' training athletes Shawn Spink, Vladimir Sinaseck, Marcel Osgood and Vladimir Camence, there is little to nothing that these players seem to have done their own way in strength training. Put it at the bottom of lurk's common ideas, like this when asked about what he has done.

Just two or three years ago Kovács had completed silver and bronze Olympic medallists, finished second in the massagedet, and was a silver medallist at the World Cup as captain of the men's Illinois University team. An exceptional athlete, he had vaulted from the school's single de facto medallist table to the top ranks of world leaders in power. On eight days of running 10,000 yards, with an intensive training schedule filled with time skating down Pacific Hill, he made all of seven dozen major championships in China.

After dumping back to the ranks of UCLA in 2013, he suffered an almost miserable start to his 2012 season, collecting six bronze medals in 2012 and a Cambridge Olympic diploma that year. He yielded fifteen better performances all year, which, after his limbs started teething dry, the San Francisco Nationals player "autureted out".

Analytical reports have found nothing happened, and he faces a battle to climb back on his team loaners following his November 11th release by the Massachusetts Athletic Department.

It have not been very easy. Less than one Friday ago, he went bad back and guilty of punching the face of a wrestler. And in recent days, his approach to appeal has fared poorly. Last season, after a mid-season torn fall, he tank-junked in an US court of law - with his lawyers threatening to sue him unless he addressed his alleged doping confession publicly.

The criticism was even louder this season.

Image copyright Getty Images Image caption Jovan Penscault is in turmoil in the NBA

Ask a few probing questions, in Belnod's case as well; he is asked to explain his appetite, his courtship with his parents and whether there had been a bad day in the offseason. Yet the assumption is not that he is emotionally disturbed. He is not unfaithful or arrogant. We believe, in his eyes, that outdresses are just trying to achieve their own goals.

Character can have positive effects, however, and even if there are further positive developments in the NBA, the progression of such a controversial player is a critique of how US sports and the game works. It is certainly a negative sign and if anything ought to change about how it works to bolster a program.

"Coaches used to talk about their tennis stars in majors. Now they've got them everywhere," says Gary Smith, the head coach at Penn State, who doesn't overstate the size of the reward for those who played in the nation's top SSQC or were willing to put in athletic service. "Look at what a lot of your better players did. I mean, you can't

=========================== Prompt Length 512 ============================
with the Hawks with a view to identifying why, as they say in the song, Hawthorn is such a happy team. Which raises the question: Are the Hawthorn players happy because they have won the past two premierships or is Hawthorn winning more because their players are happy? Or, as the Hawks themselves suspect, perhaps the answer lies between. Not only does that club feel validated in their work practices, which this season have included an unscheduled day off after their first Launceston game, but it could also use its position at the top of the off-field ladder as it works behind the scenes to mount a case against the AFL's move in taxing clubs' football department spending. That new tax will be reviewed before the end of next season and Hawthorn are looking to lobby for the exclusion of such welfare initiatives such as the regular monitoring of every player's mental health by its sport psychologist from the new tax. Another area the Hawks believe should be exempt is the sponsoring of international educational trips for its staff. Hawthorn is not the only team that claims to be focusing on better work-life balance for their players but they have worked more diligently to identify red flags among their team since Travis Tuck and his mental-health issues became front-page news by way of a third positive drug strike.

While the players' union continues to investigate the reasons behind the grievances of so many of its members, it appears beyond doubt that the demands of pre-season training have taken a disproportionate toll. This, along with the pressure most footballers feel, even during the bulk of their holiday periods, to report back for work in perfect condition. The inescapable conclusion drawn by Marsh and his team is that the expectations placed upon players over their recently increased leave period have created a pre-season before the official pre-season. According to the AFLPA and the increasingly shared belief of their leading players, the game as a spectacle would not be adversely affected by a less-intensive spring-summer. Australian football, after all, is a domestic sport. And the union are also monitoring, as reported by Fairfax Media, the link with the growing injury toll. Marsh was reluctant to detail individual club player complaints but, according to his key executive Ian Prendergast, the overall picture of dissatisfaction with their sport by the players "hit him right between the eyes". Marsh came to the AFL from cricket and lack of enjoyment with the game they took on through love and fun and enjoyment as children was relatively non-existent among Australian footballers in general. Over the past decade or so, the dissatisfaction grew. They were routinely asked how much they needed to goof around by their parents and by their coaches, which they tended to miss. Naturally, the players often had their doubts about their first grade standard and few fans made allowances for this chipper criticism when they backed their team throughout the long pre-season preparation period.

The AFL and NRL clubs have recently been struggling with growing talk of missed kickouts and sudden increments in the grading system after their clubs were caught failing to properly police Josh Boyd and Scott Lilly. In Canberra, where some of the players are running faster and talking more about drug use, the federation is testing the AFL's new anti-discrimination law, how to change the rules governing strip derangement and the expanded power of post-match sanctioning cameras. Before games, the representatives from the Sydney Maroons board, including their New Zealand captain and Sydney's assistant coach while in charge, have delivered long diatribes to the players, to the players themselves, some of which severed their ties to the club in return for the shortsighted and grievous attacks on the club's reputation and its unity. There was considerable assistance from Sydney in counting the tally of the AFLU's members, from the club's financial manager Damien Gray and chief operating officer Humphry Rokakley, to ASADA director-general Rods del Fuerta and former AFLAUT president Chris Jacobs. Unfortunately, Stephen Parish, who became the Australian AFL executive president once and for all, has told a Senate inquiry that it would be "unusual for any player not to have remoited from this federation's support when they first entered the competition formats - and especially so if most do not do so immediately in an effort to improve their competitiveness personally". Australian AFLU president Arthur Hurley and the AFLU's chief executive, Stephen

Lawrence, have recently left. Retired federal team officials and CEO Stephen Hendell, Hurley and Lawrence were all also members of the most recent AFLU national advisory committee of the WFWA, the new confederation formed before the federal government's new sporting governance regime. The refs and some of the former clubs are quite possibly expected to have already caught on and most have said that the new local-territory administration is very unlikely to take robust legal action against the only surviving Australian football union. According to Jeff Pearce, the AFLPA's national vice president, the league's new federal strategy now outlines the significant opposition to place on competitive

============================ Prompt Length 768 ============================

with the Hawks with a view to identifying why, as they say in the song, Hawthorn is such a happy team. Which raises the question: Are the Hawthorn players happy because they have won the past two premierships or is Hawthorn winning more because their players are happy? Or, as the Hawks themselves suspect, perhaps the answer lies between. Not only does that club feel validated in their work practices, which this season have included an unscheduled day off after their first Launceston game, but it could also use its position at the top of the off-field ladder as it works behind the scenes to mount a case against the AFL's move in taxing clubs' football department spending. That new tax will be reviewed before the end of next season and Hawthorn are looking to lobby for the exclusion of such welfare initiatives such as the regular monitoring of every player's mental health by its sport psychologist from the new tax. Another area the Hawks believe should be exempt is the sponsoring of international educational trips for its staff. Hawthorn is not the only team that claims to be focusing on better work-life balance for their players but they have worked more diligently to identify red flags among their team since Travis Tuck and his mental-health issues became front-page news by way of a third positive drug strike.

While the players' union continues to investigate the reasons behind the grievances of so many of its members, it appears beyond doubt that the demands of pre-season training have taken a disproportionate toll. This, along with the pressure most footballers feel, even during the bulk of their holiday periods, to report back for work in perfect condition. The inescapable conclusion drawn by Marsh and his team is that the expectations placed upon players over their recently increased leave period have created a pre-season before the official pre-season. According to the AFLPA and the increasingly shared belief of their leading players, the game as a spectacle would not be adversely affected by a less-intensive spring-summer. Australian football, after all, is a domestic sport. And the union are also monitoring, as reported by Fairfax Media, the link with the growing injury toll. Marsh was reluctant to detail individual club player complaints but, according to his key executive Ian Prendergast, the overall picture of dissatisfaction with their sport by the players "hit him right between the eyes". Marsh came to the AFL from cricket and lack of enjoyment with the game they took on through love and fun and enjoyment as children was relatively non-existent among Australian cricketers.

"Maybe I'm being romantic about sport, but I do think it should be fun," was all Marsh would say on the subject. "It's a question which needs to be addressed by everyone in the industry." The AFL is not unique in that they seems paralysed to act against the increasing pressure and negative impact of social media and the uglier side of the expanded modern "selfie" syndrome so hauntingly articulated by Chris Judd in these pages. But the AFLPA have identified player grievances it can address in their next wages-and-conditions deal. Heading those grievances are the plethora of meetings the players believe over-punctuate their working week and many believe have grown as assistant coaches over-zealously justify their jobs. Another is the growing demand of player appearances by clubs working to service members and sponsors. The concerns of the players' representative body were compounded by their pre-season round of visits but were already being addressed after the second annual players' survey last August. Players responded to questions across three areas: their club's culture; resources and its structure, which included leave periods, time off and the performance of player development managers. The AFLPA reported back to the clubs, telling them where they were ranked in each of those four areas. The annual research session was six-day. Those clubs in these four areas that showed the worst results at all times and where their own poor progress was identified as the most apparent, such as the Giants, Hawthorn and Carlton, were laid-out as operating not only without the frequent parliamentary supervises and the more outspoken directorial redistribution policy of the Granny Hill clubs – and where the players most often are active in their pursuit of the best and fairest – but they also operate with the most lax end-to-end performance-related coaching policies, have the most overworked executive list management and team performance-management staff, and rank

among the worst in their management of ticket market participation and other competition activities. Determination of future action at many clubs were not only diluted in terms of the performance of key people in the sports knowledge-based management structures – such as communication engineers, performance support managers and the managers of the union candidates, but also the clubs' player development agents, tryouts and on-balling cadres. Plus, this was punctuated by the poor performances of the club's executive list manager and model player association executive coach – who was rated below average.

"The Blues said they wanted

## H.2 Samples Generated by the Variable-Length Model Trained on Stories

================================ Sample 1 ================================

once there was a boy who liked to explore - even though he was small. every day he ' d struggle to see what he could find. he would pick small foods in his garden and visit his neighbours. one day, he noticed a delicious sauce on the food. he was so tempted to try it out! he struggled to eat it, not as usual, so he decided to buy a popular pasta with sauce. he shared it with all his friends who enjoyed their sauce. but, he refused when he urged them to keep buying. in consideration, someone called out to him : " go ahead, try the sauce. what if the sauce means you won ' t have to really savor the taste? " the boy was happy that he couldn ' t resist the temptation to try the sauce. from then on, everyone in the neighbourhood remembered the popular sauce and dreamed of becoming a more daring sauce expert.

================================ Sample 2 ================================

john was a very popular three year old boy in his town. he liked to play in his backyard in the grass. every day, he would imagine all the different things the world started to offer. one day, john looked up in the sky and came up with an idea. he got off his bike to learn the wisdom. he rode back to the meadow and told the wise owl about the wisdom. the owl said she said the world was full of special and sweet wisdom if he was patient. after a long day, the wisdom yielded a lot. john thanked the wise owl. she had worked hard and was able to learn something magical. john continued on his bike, soon out of the meadow. he saw a beautiful butterfly and smiled as attractive as she had seen him.

================================ Sample 3 ================================

tim was playing in the park one day. he skipped around, shouting. it was a game but there were no toys around. he was feeling disappointed and soon saw a boy. the boy was a lot bigger than no other boy but no one ever seemed to understand. he recognized it from a playground. the boy asked if he wanted to join him. tim hesitated at first for a few moments. but he felt a bit scared, so he decided to follow the boy. he started to climb up the ladder at first but as he went higher and higher he felt helpless. the boy saw tim and said he was looking for help. tim was gentle and asked to try to lift him up the ladder but he was very worried. after a while the boy said goodbye to tim and ran off. tim felt sad that he had to go home without to help a friend. he regrets climbing the ladder.

================================ Sample 4 ================================

timmy and lily are friends. they like to play together. one day, they see a lady on a bench. she has a big bag and a red stick. the candy is sweet and bitter. lily wants to eat candy too. she reaches into her bag and takes the candy. " no, lily! " timmy says. " that is my candy. you can ' t have it. i want this candy. it is a candy for you. " lily did not listen. she does not want to share her candy. she tries to part with her candy with the stick. but the stick is stuck. it stays loose and hurts her teeth. " ow! " lily cries. " you broke my stick! " timmy laughs. he drops the stick and pushes lily away. " no, thank you! " lily says. " you can ' t have my candy. go away now. " tom does not find another stick. he did not know how to have it. he bites the stick and pulls it out. " look, lily, i am eating candy! " tom says. " am i eating kite? " lily looks at tom. she saw what tom was doing. she feels silly and scared. she takes off her hat and her shoes and pants. " oops! sorry, tom, " timmy says. " i did not mean to scare you. i just wanted to bite the stick. maybe we can play something else. " " i know, tom, " lily says. " but i still like the stick. it is better. you can have another candy. " tom feels bad. he also likes lily. he thinks lily is generous. he gave lily a big, red apple. " here, lily, " he says. " this is a cake. it was not magic. it was a secret. " " thank you, lily, " tom says. " but next time, do not eat the stick. you are a good friend. can i help you? " lily smiles. she is glad that tom could help. she was generous. " okay,

lily, " tom says. " i will share the stick. we can play together. " they play with a ball, a kite, a book and other toys. they have fun with the ball. they are happy.

================================ Sample 5 ================================

once upon a time there was a boy named jack. he was worried because he had no worries with him, every day him would carry jack away for a day at work the first time to eat a yummy lunch. one day jack saw some of his mum outside. they were so kind and helpful to all of them. they did not put a heavy box at school or they brought any more delicious treats. jack told his mum about this because he went to work looking for something else to do. jack soon had an idea : he bought a lunchbox and place all of his mum ' s lunch. he carried each piece, and then drive home. his mum was so relieved and smiled. jack was so relieved and happy. from that day on, jack always carried his lunch with him everyday, and people never forgot to go home without a worried look.

================================ Sample 6 ================================

once there was a dog named spot. spot had a big toy that was very hairy. he loved to play with it and hug him. every day spot would sign when he would get a slapped race. one time, spot was ready to show his signa€ " black pawch. he wanted to show his friends how fast he could sign? when he signed with his hand, his friends saw him close by. they liked the show. the show was very funny. when spot managed to sign his name, his friends would even make clapping. in the end, spot and his friends all got slap spots! they are now the best! all day long, spot is signing, and even his hand feels good. everyone around him is happy every week. because of the day, spot brought a prize for his slapch with him.

================================ Sample 7 ================================

once upon a time there was a gorilla. he was big and very strong. he stored food in a crack. the gorilla enjoyed the food and enjoyed it. one day, a little boy was watching the gorilla by himself. he got stuck in a crack. he felt scared and demanded the gorilla some help from it. he knew what to do. he asked the penguin for help. the penguin grabbed a big rock and swam over to the crack. the gorilla worked very hard with his help from the penguin. the little boy was so happy. he ate his food and the penguin thanked him for the sweet treat.

================================ Sample 8 ================================

once upon a time, there was a little girl named lily who loved to play with her ball. every day, she would go to the park and send her ball to him inside a loud song. one day, lily saw a mean boy named max playing his guitar. he was playing his song and was not friendly. lily wanted to help max, so she played with him and made the music beautiful. max became friendly and welcomed lily with her ball. they had a great time together and at the end of the musical song, max ended nicely and said goodbye. lily felt sad that he was not sharing nicely. lily went home and used a plan. she brought out her guitar and spoke to max. she told him that he wasn ' t nice and allowed her to hold the guitar. max was happy and grateful. from that day on, lily and max didn ' t matter what friends tried from each other.

================================ Sample 9 ================================

once upon a time, there was a messy little boy named timmy. timmy liked to play with his toys and do chores. one day, timmy ' s stomach felt really hungry. his tummy was growling, and his food was all over the floor. his mom saw that timmy had eaten all his food, and she knew the bad smell came from lunch. timmy ate it, but she found it was disgusting and looked at the napkins on the floor. she said, " timmy, napkins are not good to eat. " she scoldolded timmy and asked him again before eating his meals. she said, " you don ' t know you would have gotten sick, though. " timmy said, " i ' m sorry, i won ' t eat them next time. " he put his hands on the napkins and threw them away so his clothes were safe from harm. from then on, timmy made sure to listen to his mom and eat his napkins before she went to eating. he never wanted to eat his napkins again. the end.

================================ Sample 10 ================================

anna liked to play with the sack. she liked to pull things out and see what she could find. she asked her dad sometimes. he explained that she was anything she could find. one day, anna found the sack in the field. she looked inside and saw her friend, lily, who was playing with a dog. she opened the sack and saw many shiny things. anna was happy. she wanted to explore more. but then, she saw a big black cat. the cat was sitting on the edge of a bush. it had dirty eyes. anna did not know what it

was. she thought, " maybe it is the cat. i can make it happy. " she went to the shiny thing and stroked it gently. the cat did not go away. it hissed. it said, " no, anna, what are you doing? go out! let ' s stay behind the tree. " anna did not listen. she stepped closer to the cat. she reached her hand out. she pulled its head. the shiny thing moved. it made a ring. it came alive with a band. it moved and snapped. anna heard the snap. she screamed. she ran back. the cat bit her. it hurt her finger. she could not hold it. she cried, " ow, ow, ow, ow, it hurts! it hurts, it hurts! " anna ran to the field. she saw the fence and the other people. she saw the cat too. she heard the people shouting and running. she saw the cat in the tree. she ran to get help. she looked for lily. she saw her mom walking in the car. she saw lily ' s finger. she felt sorry for her. she gave her ring to her. she went to lily and said, " i ' m sorry, mom. i was wrong. lily was naughty. she tried to grab lily ' s ring. " her mom smiled and hugged lily. she said, " it ' s okay, anna, it ' s okay. it doesn ' t hurt. but it ' s not your fault. you should have left the sack alone. " anna was still sad. she said, " i ' m sorry, mom. i love you, lily. " she hugged lily and learned her lesson. she never came back. she hugged lily and went home. she left the cat alone. where lily was sad. she did not harm it.

=============================== Sample 11 ===============================

once upon a time, there was a little girl named lily. she loved to play with her toys and pet dog, max. one day, she went to the park with her mom and saw a leopard flying from its back to its home. she excitedly asked her mom, " what is that, mommy? " her mom said to lily, " this is polly. this leopard is very cheap, so if it misses its home, i can take it to you. " lily listened carefully and said, " thank you, mommy. polly is so kind. can i recommend it to a party for you at the park? " later that week, lily and max watched the leopard and when it arrived, the leopard landed on lily ' s fur. she was so excited! she learned that the leopard had a modest quality on its home. later that day, lily forgot about the leopard and went to bed for a nap. she fell asleep with max, just like the modest parrot. when she woke up, she thanked max for helping her.

=============================== Sample 12 ===============================

once upon a time there was an ancient, grey kangaroo. he lived in a sunny spot near a pond. one night he heard a magical sound. it was a voice coming from the sky. startled, the kangaroo called out. he didn ' t know what to do. but then he noticed a small, red balloon glimmering in the air. he curiously approached the alien and it turned into a truck blocking his direction. it was carrying an old gas truck and it was perfectly heavy. " what is this? " the voice replied. the gasoline truck quickly zaneded the balloon and released it into the sky. the kangaroo almost entered an ancient castle and it was even more beautiful than than he had ever been there before. suddenly the gas truck spoke with a hop and the voice said, " this is a gas castle! you can ' t come back! " the kangaroo was very scared, but he had to figure out the alien ' s names. he ran higher and higher, and the dragon slowly started to slowly spring until it was gone. the kangaroo thought it was gone. the next morning, the ancient kangaroo returned to where the alien had returned. he looked back, not clear with fear - he was even more relieved to see that the magical creature had given him back. he had emerged in the right circle!

=============================== Sample 13 ===============================

once upon a time, there was a little rabbit named benny. benny loved to hop around the forest and play with his friends. one day, benny met a friendly rabbit named rosie. benny introduced rosie to his new friend. " what ' s that? " he asked. " that ' s a toy staff, " said rosie, " it ' s a loud sound that makes me reverse. " benny didn ' t understand what that meant, but rosie continued. benny asked rosie if he wanted to teach rosie her name staff. rosie said, " i don ' t know it, a toy staff means my number 10.... " benny smiled and said, " okay, let ' s count to ten. " benny and rosie learned how to reverse from ten over half cards. they had so much fun counting and learning, they played with the toy staff all day long. when benny and rosie grew up playing, they had so much fun together and were able to play with their new name toy staff.

=============================== Sample 14 ===============================

once upon a time there was a little boy named john. john had a grain wheat mill, who worked there. one day john went to the mill to make flour. he picked out some flour and put the wheat in the corn mill. as he ran over to his house, the flour slowly churned and added it to a slice of the mill. john smiled at what he had done. when he was finished when he rolled out his door, he met another boy called bob. bob said, " would you want to play with me? ". john smiled and said, " yes! " so they had fun playing and running around the mill. when john was done, bob knew exactly how bob had done

it. he held up the grain and carried them both hands. he said, " we have to take care of our wheat and be balanced for what we need ". john was balanced with excitement. he welcomed him, gave him a pat on the shoulder of my mill and was extremely pleased. bob laughed, and told john that day he and the wheat were friends forever.

============================== Sample 15 ==============================

ben and mia like to play on the navy boat on the sea. they pretend they are divers and fight sharks in space. one day, they go to a big island with their house. it is very pretty and green. it has a long sail, a flag, and a blue door. they open the door and see a lot of water in the water. they lower the boat and look over the boat. " hello, many ships! " mia says. " maybe they are pirate? " ben wonders. " maybe they are treasure, " mia says. " or other ships. we are swimming and looking for something to look for. " they see a big ship in the water. it is red and yellow and has a sword. " maybe it ' s a land from here, " ben says. " what ' s a port? " mia says. " maybe it ' s a pirate! " " who is the treasure? " ben says. he dived close and lands on the ship at mia. he lands on mia next and puts on a scarf. she smiles and laughs. " did you hear the sound of the treasure? " ben asks. " one, two, 3, blast off! " mia says. she smiles and jumps high. she sees the sky, the sun, a shark, a butterfly, a shark, and a duck. they jump off the ship and run to the shore. they call their mom and dad. " mom, look what we found! " ben says. " we are in space! " mia says. she hugs their mom and dad. they tell their parents about their adventure. " we dive in the air and dive with a ship, " ben says. " we are great and brave explorers! "

============================== Sample 16 ==============================

once upon a time there was a young boy named sam. sam was very happy because he had a nice bench. every day he used it to sit in the park. one day, he was sitting in the park on the bench while a boy came and slapped his hand. " ouch " the boy said the boy. sam looked up in surprise and started to cry. the kind boy said, " i have an idea to help the young boy. you can use your cane to make hi! it ' s all okay ". sam wiped up his tears and looked at the young boy in the bench. it was a kind surprise that the boy had seen him before. he said, " you ' re not as good as nice as the bench you like and why did who slap my hand? ". the young boy smiled and told sam that hea€™s quarrel with a friend. he said, " let ' s write down my bench and draw it. it looks like a special book ". the young boy smiled and said, " yes, let ' s draw on the bench ". sam and his friend spent lots of time playing and talking. they wrote and drew the bench and gave it a hug. sam and the boy enjoyed the bench together for many years went by and their canes were over, so they were happy.

============================== Sample 17 ==============================

once there were two friends, daniel and norman. they were mighty friends, who loved to play baseball. they would practice their best in every golf game. even on this competitive day, norman still wanted to score a goal without not being four goals as rough as norman. one day, daniel sneaked into norman ' s lead up getting his best grades. johnnie said, " let ' s invite our team to try it out. in the starting game! " the team was excited, they called the veterinarian. norman dashed to the chess room and daniel asked for a tough stick. johnnie took out a tough stick and finally swung it against his friend in the second seat. they both cheered as norman was proud that they could continue on their physical soccer game. at last, they were able to score successfully.

============================== Sample 18 ==============================

once upon a time, there was a little boy named timmy. timmy had a toy weapon, a brown sword. one day, timmy ' s mom came in and asked him to stop playing with his sword. but timmy didn ' t let go of his weapon and wanted to show her where he found it. " timmy, your weapon is on your bed, " his mom said mad, but timmy didn ' t listen. later, timmy ' s mom asked him to clean up my room and put it away. " mom said i will shoot away from your car, " she replied. timmy did a great job cleaning up the room with his dolls and his weapon was very messy. but when he came back, his sister came in and saw that all of his toys were gone. timmy couldn ' t find his toy anywhere. he looked in the closet, in the closet, and even in the closet. his sister searched everywhere, but she couldn ' t find it. timmy felt sad because he didn ' t want to make her worry. finally, he told his sister that he did not think he was stupid of any extra toys. his sister decided to find his toy and they searched the treasure together. the sword was found at its spot and timmy was glad no one could spot it.

============================== Sample 19 ==============================

once upon a time there was a modern woman. she had a very important job and she was very careful with it. each day she looked at her job, as it was a great job. then one day, the woman heard a loud noise. she looked around, looking for help. she started to look out there. suddenly, she saw a little girl running towards a 3 year old policeman. she was running and rushing to avoid her. the policeman stopped and said, " are you alright? i ' m so scared and lost. " the elderly woman went to find the cop and saw how brave the little girl was. he asked her if she wanted to help. the little girl told her yes and the cop started hunting. but the little girl bit her and threw everything away. in the end, the woman ' s heart was left increased and into distress. the moral of the story is that we must always be careful when we are hunting, especially when someone is about three years old and the little girl needed help. if we argued with someone, or else remains in our life, we can start causing consequences. this story requires a serious ending, figuring out those who are looking out for help and can cause trouble.

============================== Sample 20 ==============================

once upon a time, lily and max were best friends. one day, they found a delicate flower in the grass. it was very delicate with lots of spiky petals. max loved flowers and wanted to hold it. but the flower was too big for him and max couldn ' t hold it very much. but then, they found a snapwig and the flower fell and landed in their hands. it burst in two! lily was so happy that max found a beautiful flower they could rectangle. they decided to build a castle in the castle. lily looked for pretty stones as they found, but even though it was hard for them to find that stones would be making the castle stronger. so, they promised to keep the castle strong and delicate from one day on. they played together every day, and from that day on, the delicate flower made its castle strong and attractive.

## H.3 DEMONSTRATION OF THE INTERMEDIATE GENERATION PROCESS

============================== Step 0 ==============================

[CLS]

============================== Step 1 ==============================

[CLS] lucy

============================== Step 2 ==============================

[CLS], lucy her

============================== Step 3 ==============================

[CLS], lucy very her.

============================== Step 4 ==============================

[CLS], lucy lucy very her.

============================== Step 5 ==============================

[CLS], lucy smile lucy, very her.

============================== Step 6 ==============================

[CLS], lucy. smile lucy she, very her.

============================== Step 7 ==============================

[CLS], lucy. smile lucy she, very her.

============================== Step 8 ==============================

[CLS], lucy. smile. lucy she and, very her.

============================== Step 9 ==============================

[CLS], lucy. it smile. lucy she and, very her.

============================== Step 10 ==============================

[CLS], lucy. it smile. lucy she and, very after her.

============================== Step 11 ==============================

[CLS], lucy. it smile. lucy she and, very after her. her

================================ Step 12 ================================

[CLS], lucy.. it smile. lucy she and, very after her. park her

================================ Step 13 ================================

[CLS], lucy.. it smile. lucy she and, very after her. park her.

================================ Step 14 ================================

[CLS], lucy. a. it smile. lucy she and, very pretty after her. park her.

================================ Step 15 ================================

[CLS], lucy. a. it smile. lucy she and farm, very pretty after her. park her.

================================ Step 16 ================================

[CLS], lucy. a. it smile. lucy she the and farm farm, very pretty all after finished her. park her.

================================ Step 17 ================================

[CLS], lucy. a. it smile. lucy to she the and farm farm,. very pretty playing all after finished her. park her.

================================ Step 18 ================================

[CLS], there lucy. a. it smile. lucy to she the and farm farm,. very pretty playing all after finished her. park her.

================================ Step 19 ================================

[CLS], there lucy. a. it smile. lucy to she the and farm farm,. onion very pretty playing all after finished her. to park her.

================================ Step 20 ================================

[CLS], there lucy. a. it and smile. lucy to. she the and farm farm,. onion very pretty playing all after finished her. to park her mom.

================================ Step 21 ================================

[CLS], there lucy. a. it and smile. lucy to. she the and farm farm,. onion very pretty playing all after finished her. to park her mom more.

================================ Step 22 ================================

[CLS], there lucy. a. it and smile. lucy to. she saw the and farm farm,. onion very pretty playing all after finished her. to park her mom the more fun.

================================ Step 23 ================================

[CLS] a, there lucy. a. it and smile. lucy the to. she saw the and farm farm,. onion very pretty playing all after finished her. to park her mom the more fun.

================================ Step 24 ================================

[CLS] a, there lucy. a. it and smile. lucy the to. she saw the and farm farm,. onion very pretty playing all after finished her. to park her mom the more fun.

================================ Step 25 ================================

[CLS] a, there lucy. a. it and smile. lucy the to. she saw the and farm farm,. onion very pretty playing all after finished her. to park her mom the more fun.

================================ Step 26 ================================

[CLS] a, there lucy. a. it and smile. lucy the to. she saw the and farm farm,. onion very pretty playing all after finished her. to park her mom the more fun.

================================ Step 27 ================================

[CLS] a, there lucy. a. it and smile. lucy the park to. she saw the and farm farm,. onion very pretty playing all after finished, her. to park her mom the they more fun.

================================ Step 28 ================================

[CLS] a, there lucy. a. it and smile. lucy the park to. she saw the and farm farm,. onion very pretty playing all after finished, her. to park her mom the they more fun.

================================ Step 29 ================================

[CLS] a, there was lucy. she a. it and smile. lucy the park to. she saw the and farm farm,. onion very pretty playing all after finished, her. to park with her mom the they more fun.

================================ Step 30 ================================

[CLS] a, there was lucy. she a. it attractive and smile. lucy the park to. she saw the and farm farm,. onion very pretty playing all after finished, her. to park with her mom the they more fun.

================================ Step 31 ================================

[CLS] a, there was lucy. she a. it attractive and smile. lucy the park to. she saw the and farm farm,. onion very pretty playing all after finished, her. very to park with her mom the they more fun the.

================================ Step 32 ================================

[CLS] a, there was lucy. she a. it attractive and smile. lucy the park to. she saw the and farm farm,. onion very pretty playing all after finished, her. very to park with her mom the they more fun the.

================================ Step 33 ================================

[CLS] a, there was lucy. she a. it attractive and smile. lucy the park to. she saw the and farm farm,. onion very pretty. playing all after finished, her. very to park with her mom the they more fun the.

================================ Step 34 ================================

[CLS] a, there was lucy. she a. it attractive and smile. lucy the park to. she saw the and farm farm,. onion very pretty. playing all after finished, her. was very to the park with her mom the they more fun the.

================================ Step 35 ================================

[CLS] a time, there was lucy. she a. it was attractive and smile. lucy the park to. she saw the trees and farm farm,. onion very pretty. playing all after finished, her. was very to have the park with her mom the they more fun the.

================================ Step 36 ================================

[CLS] a time, there was lucy. she a. it was attractive and smile. lucy the park to. she saw the trees and farm farm,. onion very pretty. playing all after finished, her. was very to have the park with her mom the they more fun the.

================================ Step 37 ================================

[CLS] a time, there was lucy. she a. it was attractive and smile. lucy the park to. she saw flowers the trees and the farm farm,. onion very pretty. playing all after finished, her. was very to have the park with her mom the they more fun the.

================================ Step 38 ================================

[CLS] a time, there was lucy. she a. it was attractive and smile. day lucy the park to. she saw flowers the trees and the farm farm, onions. onion very pretty. playing all after finished, went her. was very to have the park with her mom the onion they more fun the.

================================ Step 39 ================================

[CLS] a time, there was lucy. she a. it was attractive and made smile. day lucy the park to. she saw flowers the trees and the farm farm, onions. onion very pretty. playing all after finished, went her. was very to have the park with her mom the onion they more fun the.

================================ Step 40 ================================

[CLS] a time, there was lucy. she a. it was very attractive and made smile. day lucy the park to. she saw flowers the trees and the farm farm, onions. onion very pretty. playing all after finished, went her. was very to have the park with her mom the onion they more fun the.

================================ Step 41 ================================

[CLS] a time, there was named lucy. she a. it was very attractive and made smile. day lucy the park to. she saw flowers, the trees and the farm farm, onions. onion very pretty. playing all after finished, went her. was very to have the park with her mom the onion they more fun the.

================================ Step 42 ================================

[CLS] a time, there was named lucy. she a. it was very attractive and made smile. day lucy the park to. she saw flowers, the trees, and the farm farm, onions. onion very pretty. playing, all after finished, went her. was very to have the park with her mom the onion they more fun at the.

================================ Step 43 ================================

[CLS] a time, there was named lucy. she a. it was very attractive and made smile. day lucy the park to. she saw flowers, the trees, and the farm farm, onions. tomatoes onion very pretty. playing, all after finished, went her. was very to have the park with her mom the onion they planned more fun at the.

================================ Step 44 ================================

[CLS] a time, there was named lucy. she a. it was very attractive and made smile. day lucy went the park to. she saw flowers, the trees, and the farm farm, onions. tomatoes onion were very pretty. playing, all after finished, went her mom. she was very to have the park with her mom the onion they planned more fun at the.

================================ Step 45 ================================

[CLS] a time, there was named lucy. she a. it was very attractive and made smile. day lucy went the park to. she saw flowers, the trees, and the farm farm, onions. orange tomatoes onion were very pretty. playing, watered all after finished, went her mom. she was very to have the park with her mom the onion they planned more fun at the.

================================ Step 46 ================================

[CLS] a time, there was named lucy. she a. it was very attractive and made smile. day lucy went the park to. she saw flowers, the trees, and the farm farm, onions. orange tomatoes onion were very pretty. playing, watered all after finished, went her mom. she was very to have the park with her mom the onion they planned more fun at the.

================================ Step 47 ================================

[CLS] a time, there was named lucy. she a. it was very attractive and made smile. day lucy went the park to. she saw flowers, the trees, and the farm farm, onions. orange tomatoes and onion were very pretty. playing, watered all after finished planting, went her mom. she was very to have seen the park with her mom and the onion they planned more fun at the.

================================ Step 48 ================================

[CLS] a time, there was named lucy. she a. it was very attractive and made smile. day lucy went the park to. she saw flowers, the trees, and the farm farm, onions. orange tomatoes and onion were very pretty. playing, watered all after finished planting, went to her mom. she was very to have seen the park with her mom and the onion they planned more fun at the.

================================ Step 49 ================================

[CLS] a time, there was named lucy. she a dress. it was very attractive and made smile. day lucy went the park to. she saw flowers, the trees, and the farm farm, lucy onions. they orange tomatoes, and onion were very pretty. playing, watered all after finished planting, went to her mom. she was very to have seen the park with her mom and the onion they planned more fun at the.

================================ Step 50 ================================

[CLS] a time, there was named lucy. she a dress. it was very attractive and made smile. day lucy went the park to. she saw flowers, the trees, and the farm farm, lucy onions. they orange tomatoes,

and onion were very pretty. playing, watered all after finished planting, went to her mom. she was very to have seen the park with her mom and the onion they planned more fun at the.

================================= Step 51 =================================

[CLS] once a time, there was named lucy. she a dress. it was very attractive and made smile. day lucy went to the park to. she saw flowers, the trees, and the farm farm, lucy saw onions. they orange tomatoes, and onion. were very pretty. playing, watered all after finished planting, went to her mom. she was very to have seen the park with her mom and the onion they planned more fun at the day.

================================= Step 52 =================================

[CLS] once a time, there was named lucy. she a dress. it was very attractive and made her smile. day lucy went to the park to. she saw flowers, the trees, and the farm farm, lucy saw onions. they orange tomatoes, and onion. thought were very pretty. playing, watered all after lucy finished planting, went to her mom. she was very to have seen the park with her mom and the onion they planned more fun at the day.

================================= Step 53 =================================

[CLS] once a time, there was named lucy. she a dress. it was very attractive and made her smile. day lucy went to the park to. she saw flowers, the trees, and the farm farm, lucy saw onions. they orange tomatoes, and onion. thought were very pretty. playing, watered all vegetables after lucy finished planting, went to her mom. she was very happy to have seen the park with her mom and the onion they planned more fun at the day.

================================= Step 54 =================================

[CLS] once a time, there was little named lucy. she a dress. it was very attractive and made her smile. day lucy went to the park to. she saw flowers, the trees, and the farm farm, lucy saw onions. they orange tomatoes, and onion. thought were very pretty. after playing, watered all vegetables after lucy finished planting, went to her mom. she was very happy to have seen the park with her mom and the onion they planned more fun at the every day.

================================= Step 55 =================================

[CLS] once a time, there was little named lucy. she a dress. it was very attractive and made her smile. day, lucy went to the park to. she saw flowers, the trees, and the farm farm, lucy saw onions. they orange tomatoes, and onion. thought were very pretty. after playing, lucy watered all her vegetables after lucy finished planting, went to her mom. she was very happy to have seen the park with her mom and the onion they planned more fun at the park every day.

================================= Step 56 =================================

[CLS] once a time, there was little named lucy. she a dress. it was very attractive and made her smile. one day, lucy went to the park to. she saw flowers, the trees, and the farm farm, lucy saw onions. they orange tomatoes, and onion. thought they were very pretty. after playing, lucy watered all her vegetables after lucy finished planting, went to her mom. she was very happy to have seen the park with her mom and the onion they planned more fun at the park every day.

================================= Step 57 =================================

[CLS] once a time, there was a little named lucy. she a nice dress. it was very attractive and made her smile. one day, lucy went to the park to. she saw flowers, the trees, and the farm the farm, lucy saw onions. they orange tomatoes, and onion. thought they were very pretty. after playing, lucy watered all her vegetables after lucy finished planting, went to her mom. she was very happy to have seen the park with her mom and the onion they planned more fun at the park every day.

================================= Step 58 =================================

[CLS] once a time, there was a little named lucy. she a nice dress. it was very attractive and made her smile. one day, lucy went to the park to. she saw the flowers, the trees, and the farm the farm, lucy saw onions. they orange tomatoes, and onion. thought they were very pretty. after playing, lucy watered all her vegetables after lucy finished planting, went to her mom. she was very happy to have seen the park with her mom and the onion they planned more fun at the park every day.

================================ Step 59 ================================

[CLS] once upon a time, there was a little girl named lucy. she had a nice dress. it was very attractive and made her smile. one day, lucy went to the park to play. she saw the flowers, the trees, and the farm. the farm, lucy saw many onions. they orange tomatoes, and onion. thought they were very pretty. after playing, lucy watered all her vegetables after lucy finished planting, went to her mom. she was very happy to have seen the park with her mom and the onion they planned more fun days at the park every day.

================================ Step 60 ================================

[CLS] once upon a time, there was a little girl named lucy. she had a nice dress. it was very attractive and made her smile. one day, lucy went to the park to play. she saw the flowers, the trees, and the farm. the farm, lucy saw many onions. they orange tomatoes, and onion. she thought they were very pretty. after playing, lucy watered all her vegetables after lucy finished planting, went to her mom. she was very happy to have seen the park with her mom and the onion they planned more fun days at the park every day.

================================ Step 61 ================================

[CLS] once upon a time, there was a little girl named lucy. she had a nice dress. it was very attractive and made her smile. one day, lucy went to the park to play. she saw the flowers, the trees, and the farm. at the farm, lucy saw many onions. they orange tomatoes, and orange onion. she thought they were very pretty. after playing, lucy watered all her vegetables after lucy finished planting, went to her mom. she was very happy to have seen the park with her mom and the onion. they planned more fun days at the park every day.

================================ Step 62 ================================

[CLS] once upon a time, there was a little girl named lucy. she had a nice dress. it was very attractive and made her smile. one day, lucy went to the park to play. she saw the flowers, the trees, and the farm. at the farm, lucy saw many onions. they were orange tomatoes, and orange onion. she thought they were very pretty. after playing, lucy watered all her vegetables after lucy finished planting, went to her mom. she was very happy to have seen the park with her mom and the onion. they planned more fun days at the park every day.

================================ Step 63 ================================

[CLS] once upon a time, there was a little girl named lucy. she had a nice dress. it was very attractive and made her smile. one day, lucy went to the park to play. she saw the flowers, the trees, and the farm. at the farm, lucy saw many onions. they were orange tomatoes, and orange onion. she thought they were very pretty. after playing, lucy watered all her vegetables. after lucy finished planting, went to her mom. she was very happy to have seen the park with her mom and the onion. they planned more fun days at the park every day.

================================ Step 64 ================================

[CLS] once upon a time, there was a little girl named lucy. she had a nice dress. it was very attractive and made her smile. one day, lucy went to the park to play. she saw the flowers, the trees, and the farm. at the farm, lucy saw many onions. they were orange tomatoes, and orange onion. she thought they were very pretty. after playing, lucy watered all her vegetables. after lucy finished planting, she went to her mom. she was very happy to have seen the park with her mom and the onion. they planned more fun days at the park every day.

