# OpenReview forum: "Beyond Masks: Efficient, Flexible Diffusion Language Models via Deletion-Insertion Processes"
_ICLR.cc/2026/Conference — ICLR 2026 Poster_

### Official Review · Reviewer_cWDw · 2025-10-28

**Soundness:** 4
**Presentation:** 2
**Contribution:** 4
**Rating:** 8
**Confidence:** 3

**Summary:**

This paper proposes Deletion-Insertion Diffusion (DID) language models, which formulate token deletion and insertion as discrete diffusion processes. This approach eliminates the need for MASK and PAD tokens, thereby improving computational efficiency and generative flexibility over standard Diffusion Language Models (DLMs). The authors report that their experiments demonstrate DID's superior efficiency and flexibility compared to MDLMs and other insertion-based baselines across tasks in language/length modeling, generation quality, and training/inference speed.

**Strengths:**

* The paper is exceptionally thorough, and the appendix provides numerous additional interesting findings.
* The proposed method effectively addresses two significant limitations of existing diffusion language models (DLMs): managing variable-length generation and the computational waste associated with MASK tokens.
* The application of dynamic programming for training diffusion models is original and creative

**Weaknesses:**

* The discussion of prior work on insertion-based LLMs feels underdeveloped. This concept has been explored in several papers (e.g., [1, 2, 3]), and the current draft would benefit from a more thorough discussion beyond the most recent related work.

* The choice of evaluation metrics could be strengthened. The use of GPT2-XL as a base model is questionable given the existence of much better pretrained models. Furthermore, the use of unigram entropy seems less informative than computing a joint entropy, which could also be accomplished using an LLM. While these choices may follow existing conventions, adopting more robust metrics and baselines would significantly increase the credibility of the results.

* Appendix H does include the examples of RADD. A more direct qualitative comparison could be achieved by conditioning on OWT prefixes from the validation set and sampling from each model.

[1] Lu, Sidi, Tao Meng, and Nanyun Peng. "Insnet: An efficient, flexible, and performant insertion-based text generation model." Advances in Neural Information Processing Systems 35 (2022): 7011-7023.

[2] Gu, Jiatao, Changhan Wang, and Junbo Zhao. "Levenshtein transformer." Advances in neural information processing systems 32 (2019).

[3] Stern, Mitchell, et al. "Insertion transformer: Flexible sequence generation via insertion operations." International Conference on Machine Learning. PMLR, 2019.

**Questions:**

1. Could the authors please provide the exact training and inference algorithms, ideally in pseudocode? While they are presumably similar to SEDD, explicit algorithms would greatly improve clarity.

2. What are the precise inputs and outputs to the Transformer model during training? Given that the sequence length changes, how is potential GPU underutilization (due to variable batch sizes or padding) managed?

3. As seen in Table 2, DID's generative perplexity appears to plateau (relative to RADD) at a high number of denoising steps (e.g., 512 and 1024). Do the authors have an explanation for this phenomenon?

---

> ### Author Response · Authors · 2025-11-23
>
> Thank you for your review and constructive comments. We have revised our paper writing and added more experiments to address your concerns. Below is our point-by-point response.
>
> W1, Q1: Thank you for the writing suggestions to improve the completeness of our paper.
>
> - We have added more related works about insertion-based LMs to Sec. 2.2.
> - We have added the pseudocode of DP, DID training, and DID inference to Appendix G.
>
> W2: We appreciate the suggestions regarding evaluation metrics, for the sample quality evaluations of fixed-length and variable-length models in Tab. 2, 4, we have re-evaluated the joint entropy measured by a stronger LLM scorer Llama3.2-3B, and got conclusions similar to the paper. Please refer to **Common Concern 2** for details. Besides, we have scaled up DID to 1.1B parameters and evaluated the prediction accuracy on downstream commonsense reasoning tasks with the LM-Evaluation-Harness framework to strengthen the evaluation metrics. Please refer to **Common Concern 3** for details.
>
> W3: We appreciate the suggestion to evaluate the conditional generation performance of DID. We evaluate DID and RADD on OWT prefixes with different prompt lengths (256, 512, and 768). The results have demonstrated that DID could achieve stable lower perplexity with comparable unigram entropy (diversity) and generation length in various conditional generation settings. Please refer to **Common Concern 2.3** for details.
>
> Q2: Regarding more detailed explanations of the DID training process.
>
> - We have added the pseudocode of DID training to Appendix G. During training, the clean data $\boldsymbol x _ 0$ is first noised by the deletion process (described in Sec. 3.1) to construct a subsequence $\boldsymbol x _ t$, based on $\boldsymbol{x} _ 0$ and $\boldsymbol{x} _ t$, we compute the training target ‘N ratios’ $\frac{N(\text{Ins}(\boldsymbol{x} _ t, i, v), \boldsymbol{x}_0)}{N(\boldsymbol{x} _ t, \boldsymbol{x} _ 0)}$ shaped $|\boldsymbol{x} _ t| \times |\mathcal{V}|$ for the insertion score network $\bar{s} _ \theta(\boldsymbol x_t, t)$ to predict (as described in Eq. 7). Given an incomplete sequence $\boldsymbol{x} _ t$, the insertion score network  $\bar{s} _ \theta(\boldsymbol x _ t, t)$  predicts what should be inserted after each position to better reconstruct $\boldsymbol{x} _ 0$ at time $t$.
> - To handle variable-length data, we use the VDM sampler [1] to sample more evenly distributed time $t$ for a batch of data to ensure the number of deleted tokens is about 1/2 in the whole batch rather than completely random, which is beneficial for DID to stabilize the memory usage and balance the workload on different devices. For the batched variable-length attention computation, we use the API from FlashAttention [2] (flash_attn_varlen_func) to implement this function.
>
> Q3: We believe this phenomenon reflects the efficiency characteristics of the two methods rather than a fundamental capacity.
>
> **1. Convergence is observed for both models and the gap is marginal.**
>
> We respectfully note that the plateauing behavior is not unique to DID. As shown in Tab 2, RADD also **converges** between 512 steps (**84.00**) and 1024 steps (**84.05**), showing no further gain. This suggests that for discrete diffusion models on this benchmark, increasing steps beyond a certain threshold yields **diminishing returns**. Furthermore, the asymptotic gap between the two models at 1024 steps is relatively small (**85.35** vs. **84.05**), indicating that DID remains highly competitive even in this extreme regime.
>
> **2. DID converges earlier (efficiency advantage).**
>
> The key difference lies in *when* convergence happens. DID achieves strong performance much earlier in the process.
>
> - **At 128 steps:** DID (**91.25**) already significantly outperforms RADD (**95.10**).
> - **At 256 steps:** DID (**86.98**) continues to surpass RADD (**87.56**). The fact that DID's performance curve flattens earlier is a direct result of its **higher sample efficiency** in the low-to-medium NFE regime. It recovers the majority of information with fewer operations, which is a desirable property for practical efficiency.
>
> **3. Robustness in general variable-length settings.**
>
> Finally, in the variable-length setting (Tab. 4) which removes the artificial constraints of fixed-length padding,DID consistently outperforms RADD across all step counts (e.g., **23.88** vs. **26.78** at 512 steps). This confirms that DID maintains robust modeling capacity in the general cases for which it is designed.
>
>
>
> [1] Diederik Kingma, Tim Salimans, Ben Poole, and Jonathan Ho. Variational diffusion models. Advances in neural information processing systems, 34:21696–21707, 2021.
>
> [2] Tri Dao, Daniel Y. Fu, Stefano Ermon, Atri Rudra, and Christopher Ré. FlashAttention: Fast and memory-efficient exact attention with IO-awareness. In *Advances in Neural Information Processing Systems (NeurIPS)*, 2022.

---

### Official Review · Reviewer_orpS · 2025-11-01

**Soundness:** 3
**Presentation:** 2
**Contribution:** 3
**Rating:** 6
**Confidence:** 3

**Summary:**

This paper introduces a diffusion language model that features a deletion based forward process and an insertion based reverse process. A core advantage of the proposed framework lies in that it does not require the modeling of mask tokens, thus achieving variable-length modeling while obtaining speedup. The training objective is derived in a principle manner from score entropy, and a DP algorithm has been proposed to help efficiently obtain the loss function. Experiments show that DID achieves lower perplexity while unlocking variable-length modeling and bringing certain speedup in training.

**Strengths:**

1. The proposed insertion-deletion based diffusion framework is promising that enables variable-length modeling and relieves from modeling mask tokens.

2. The empirical performance of the model is competitive. It achieves lower perplexity compared with fixed-length mask diffusion models.

3. The additional speedup in training brought by the model is favorable.

**Weaknesses:**

1. Lack of investigations on inference strategies. For instance, it would be interesting to show how the sample quality changes with the number of inference steps (or equivalently, inference-compute).

2. More theoretical insights and discussions on the connections/advantages over MDLMs are missing. Please refer to Q2, 3 and 4 for more details.

**Questions:**

1. How does the proposed approach perform under different inference-time compute budgets (e.g. number of steps)?

2. It is still a bit unclear what exactly helps DID to achieve lower perplexity than MDLMs. Is it fundamentally due to the removal of mask tokens or something else? Adding more theoretical insights would be helpful.

3. What is the connection between DID and MDLMs in terms of modeling? Is it possible to construct a one-one mapping for each deletion-insertion process vs mask/unmask process in MDLMs? If not, what leads to the gap?

4. I am a bit confused why it is beneficial (or at least claimed to be) to remove PAD tokens. To me having PAD tokens does have some benefits even for DID since the model will then force to stop when it inserts a PAD token which may mitigate certain ambiguity.

---

> ### Author Response · Authors · 2025-11-23
>
> Thank you for the insightful review and valuable comments. Below is our point-by-point response.
>
> W1, Q1: We appreciate the suggestion for more inference strategies, together with reviewer Bfo3’s Q1, we have added more inference experiments in the response to **Common Concern 2**, including the analyses of decoding steps, timestep schedules (uniform and cosine), and conditional generation.
>
> W2, Q2: Thank you for raising the question of theoretical advantages of DID over MDM. Regarding the lower zero-shot modeling perplexity (Tab. 1), we credit the superiority of DID to the removal of $\texttt{\<MASK\>}$ tokens, as we discussed in the paper, using half of the training FLOPs on  $\texttt{\<MASK\>}$ tokens in MDM is FLOPs-inefficient and wasteful. DID avoids this by redesigning the modeling paradigm, and experiments have demonstrated its effectiveness. Besides, we have summarized four aspects of insights on the benefits of DID generation in **Common Concern 4.**
>
> Q3: Regarding the connection between DID and MDLMs: it is not possible to construct a one-to-one mapping. Instead, there is a many-to-one mapping from MDLM noising paths to DID noising paths. Consider the state spaces for a sequence of length $L$, where $\mathcal{Y} = (\mathcal{V} \cup \{\texttt{\<M\>}\})^L$ corresponds to MDLM and $\mathcal{X} = \bigcup_{k=0}^L \mathcal{V}^k$ corresponds to DID. We can obtain a corresponding DID state by simply removing all mask tokens from any MDLM state. This operation defines a mapping $\Pi: \mathcal{Y} \to \mathcal{X}$ given by $\Pi(\boldsymbol{y}) = (y _ i) _ {i: y _ i \neq \texttt{\<M\>}}$ such that distinct MDLM states map to the exact same DID state. For example, masking different 'p's in "apple" creates distinct MDLM state $\boldsymbol{y}^{(1)} = \texttt{a\<M\>ple}$ and $\boldsymbol{y}^{(2)} = \texttt{ap\<M\>le}$, but under the mapping, both become the same DID state $\boldsymbol{x} = \Pi(\boldsymbol{y}^{(1)}) = \Pi(\boldsymbol{y}^{(2)}) = \texttt{aple}$. This state collapse leads to the theoretical gap. In MDLM, the corruption path is unique because the mask explicitly holds the position. In DID, multiple deletion paths can lead to the same state, so it must mathematically sum over all these valid paths (i.e., all distinct MDLM states that map to the current subsequence).  This summation is explicitly captured by the term $N(\boldsymbol{x}_t, \boldsymbol{x}_s)$ in our Eq. 5.
>
> Q4: First, diffusion sampling is naturally completed when we reach $t=0$ (we have added the pseudocode of DID inference to Appendix G.3), so it does not require a stopping token. Second, to our understanding, even if we use padding in DID, we should not immediately stop generation after inserting a $\texttt{\<PAD\>}$ token, because there may still be other tokens to be inserted before $\texttt{\<PAD\>}$. Please correct us if we misunderstood your question. Third, we would like to clarify that explicit stopping tokens are unnecessary in DID because it possesses an **intrinsic stopping mechanism** derived from the properties of the insertion score. At the same time, removing $\texttt{\<PAD\>}$ tokens helps a lot with computational efficiency.
>
> **1. Intrinsic Stopping Mechanism:**
>
> DID naturally learns the data’s length distribution. We demonstrate this by analyzing the insertion score $\bar{s}(\boldsymbol{x}_t, t)$. By applying the subsequence count identity (Lemma 2, Appendix D.7) to the definition of the insertion score, we derive that the total sum equals the expected remaining length:
>
> $$ \sum_{i, v} \bar{s}\left(\boldsymbol{x_t}, t\right)[i, v]=\mathbb{E}_{\boldsymbol{x_0} \sim p\left(\cdot \mid \boldsymbol{x_t}, t\right)}\left[\left|\boldsymbol{x_0}\right|-\left|\boldsymbol{x_t}\right|\right] $$
>
> This equation demonstrates that as generation proceeds ($t \to 0$) and $|\boldsymbol{x}_t|$ approaches the target length, this expectation naturally decreases towards zero. Consequently, the probability of inserting any new token vanishes, causing the process to terminate intrinsically without needing a force-stop token. This is empirically validated in **Fig. 2**, where DID’s length distribution closely aligns with the ground truth.
>
> **2. Computational Efficiency:**
>
> In variable-length settings, MDLMs must pad sequences to a fixed maximum length $L_{max}$. This introduces overhead when the average data length $L _ {avg} \lt L _ {max}$. By eliminating $\texttt{\<PAD\>}$ (and $\texttt{\<MASK\>}$) tokens, DID avoids this overhead, leading to significant speedups (as shown in Tab. 5).

---

### Official Review · Reviewer_ciUs · 2025-11-04

**Soundness:** 3
**Presentation:** 4
**Contribution:** 3
**Rating:** 6
**Confidence:** 4

**Summary:**

This paper proposes DID, a novel algorithm for text generation using discrete diffusion models, which enables the generation of variable-length text through an insertion-deletion process. DID is theoretically grounded with an ELBO-like training objective and demonstrates promising performances on both fixed-length text and variable-length text generation tasks. DID enjoys decent inference and training speed compared with pure mask-based discrete diffusion approaches, suggesting the method's advantage.

**Strengths:**

- DID is a novel, theoretically grounded approach that tackles the important problem of variable-length generation of texts using discrete diffusion, bridging an important gap to the traditional AR methods.
- The technical details are sound with a self-contained derivation.
- The numerical experiments are comprehensive, and the results are promising.

**Weaknesses:**

- While the experiments are comprehensive, the results could be made more convincing by comparing with more discrete diffusion model baselines other than MDM, such as MDM-prime [1], Block Diffusion [2], etc.
- The paper could use more space to discuss differences & advantages compared with other variable-length discrete diffusion approaches. The existing discussion is nice, but it's not entirely clear what the advantage of DID is compared to existing approaches, especially EditFlow [3], which also uses an insertion-deletion process.
- The paper claims that the auxiliary process introduced in EditFlow is ineffective and introduces additional variance, but I feel like DID suffers from a similar issue due to the need to compute the number $N(x_t, x_0)$. These both originate from the randomness that connects $x_t$ to $x_0$, and it's unclear to me at this moment why the DID approach is better.
- The paper could comment more on the speed-up (in both training and inference) of DID compared with traditional MDM, and have a more thorough analysis of why DID enjoys such acceleration.


References:
[1] Chao, Chen-Hao, et al. "Beyond Masked and Unmasked: Discrete Diffusion Models via Partial Masking."
[2] Arriola, Marianne, et al. "Block diffusion: Interpolating between autoregressive and diffusion language models."
[3] Havasi, Marton, et al. "Edit Flows: Flow Matching with Edit Operations."

**Questions:**

Besides the points mentioned in the weakness section, I would appreciate clarification on the following questions to better my understanding of the paper:
- Regarding the computation of $N(x_t, x_0)$, in practice, how much computational overhead does this incur, and is it expensive?
- Tables 3 & 4 mentioned the training/inference acceleration enjoyed by DID. Can you comment on why DID enjoys such a significant speedup compared to MDM?
- Table 4 mentions that with 64 NFE, DID can generate sequences with ~180 tokens and have decent ppl. How is DID capable of modeling token interdependence/joint distribution that allows such high-quality fast sampling?
- How's DID's actual sampling throughput compared with MDM? How much acceleration does it enjoy in GPU clock time in terms of inference?

---

> ### Author Response · Authors · 2025-11-23
>
> Thank you for your detailed review and insightful comments. Below is our point-by-point response.
>
> W1: Thank you for suggesting the inclusion of additional baselines. We have carefully considered your suggestion and would like to provide the following response regarding our current experimental setup: First, we note that direct comparisons with the mentioned methods (e.g., MDM-prime and Block Diffusion) present certain challenges due to differences in training configurations—such as FLOPs and noise schedules—which may lead to inconsistent or unfair comparisons. Second, from the methodology perspective, our work focuses on comparing with the core mechanism of MDM as a pure diffusion model, whereas the suggested baselines often incorporate MDM with additional components. Our intention is to maintain a clear and focused comparison with the fundamental MDM framework to better highlight the contributions of our approach. We hope this clarifies our current experimental design, and we sincerely appreciate your understanding. We are open to further discussion or suggestions regarding this matter.
>
> W2, W3: Thank you for the suggestion to add more discussions of related works about variable-length discrete diffusion approaches, especially Edit Flows, to enrich the content of our paper. We have revised our paper writing of Sec. 2.2: Insertion-Based Language Models.
>
> W4, Q1, Q2, Q4: We appreciate the suggestion to do more thorough efficiency analyses on DID training and inference. We have profiled the time cost of specific components in the DID training (e.g. DP) and inference (e.g. NFE and sampling) to further illustrate DID’s superior efficiency by removing the $\texttt{\<MASK\>}$ and $\texttt{\<PAD\>}$ tokens in its training and inference. Please refer to **Common Concerns 1 and 2** for the details.
>
> Q3: We credit DID’s superior performance in fast sampling regimes to four structural advantages over RADD come from using insertion instead of unmasking during generation, please refer to **Common Concern 4**.

---

### Official Review · Reviewer_Bfo3 · 2025-11-05

**Soundness:** 3
**Presentation:** 3
**Contribution:** 3
**Rating:** 8
**Confidence:** 2

**Summary:**

The paper proposes a deletion–insertion diffusion approach for language modeling that replaces mask/unmask steps with a forward deletion process and a reverse insertion process. It introduces a training objective tailored to insertions (with a simplified version for fixed-length data) and a GPU-friendly dynamic-programming routine to make training practical. Experiments on fixed- and variable-length text report competitive or better quality than masked-diffusion baselines at similar compute, faster training and inference, and more faithful control of generated lengths. Overall, it’s a clean reformulation with a credible efficiency story.

**Strengths:**

Clear, well-motivated reformulation that naturally supports variable length; practical training and sampling mechanics that fit modern accelerators; consistent efficiency gains with competitive quality; writing is clear and limitations are acknowledged.

**Weaknesses:**

Evidence is concentrated on relatively small models and moderate sequence lengths, so scalability to long contexts and larger models is uncertain; evaluation leans on automatic metrics with a single external scorer and no human assessment; baseline alignment (steps, precision, compute) and ablations could be tighter to isolate where gains come from; the added bookkeeping likely introduces overhead whose impact isn’t fully profiled.

**Questions:**

Could the you comment on how sensitive the results are to the number of denoising rounds and the chosen step schedule?

Can  you share observations about scaling to longer contexts and larger models, including memory use and latency?

Would it be possible to provide runs that more strictly align compute, precision, and sampling settings across baselines to clarify where the improvements originate?

---

> ### Author Response · Authors · 2025-11-23
>
> Thanks for your review and the valuable suggestions. We have added several experiments to enrich the content of our paper and address your concerns.
>
> W1, Q2: We appreciate the reviewer’s concern regarding the scalability of DID to larger models and longer sequences. We have added corresponding experiments, please refer to **Common Concerns 1 and 3**.
>
> W2: Thank you for the suggestion to add more evaluation metrics.
>
> - We have added 2 more metrics of joint entropy (measured by another stronger LLM scorer llama3.2-3B) and throughput (tokens per second) for inference analysis as described in **Common Concern 2**.
> - Besides, we evaluate the DID-1.1B model on a series of downstream commonsense reasoning tasks using the LM-Evaluation-Harness framework, which also involves another evaluation metric of prediction accuracy.
>
> W3, Q3: Thank you for the suggestion of tighter ablations. We have evaluated DID with more checkpoints to locate where the improvements originate. We additionally evaluate the zero-shot perplexity (Tab. 1) using more checkpoints (25% FLOPs and 75% FLOPs), together with the experiments introduced in **Common Concern 3**, our conclusion is: on average, DID could surpass MDM, on the 7 datasets for zero-shot perplexity evaluation (Tab. 1) and 8 tasks for commonsense reasoning evaluation (**Common Concern 3**), with 50% to 75% training FLOPs.
>
> |        |                              | wikitext  | lambada   | pubmed    | agnews    | lm1b      | arxiv     | ptb        | **average** |
> | ------ | ---------------------------- | --------- | --------- | --------- | --------- | --------- | --------- | ---------- | ----------- |
> | small  | RADD-400k                    | 38.27     | 51.82     | 56.99     | 73.18     | 72.99     | 85.95     | **108.79** | 69.71       |
> |        | **DID-200k (25% FLOPs)**     | 40.50     | **50.55** | 57.87     | 81.02     | 76.37     | **85.86** | 122.84     | 73.57       |
> |        | DID-400k (50% FLOPs, DID-S)  | 38.72     | **49.10** | **55.02** | 76.02     | 74.04     | **82.41** | 115.37     | 70.10       |
> |        | **DID-600k (75% FLOPs)**     | **37.41** | **48.40** | **53.59** | **72.88** | **72.64** | **79.94** | 112.95     | **68.26**   |
> |        | DID-800k (100% FLOPs, DID-F) | **36.91** | **48.00** | **52.89** | **71.48** | **72.04** | **78.38** | 111.60     | **67.33**   |
> | medium | RADD-400k                    | 28.44     | 44.10     | 41.06     | 48.96     | 60.32     | 66.28     | **81.05**  | 52.89       |
> |        | **DID-200k (25% FLOPs)**     | 31.11     | **43.69** | 44.36     | 57.60     | 62.74     | 67.80     | 97.26      | 57.79       |
> |        | DID-400k (50% FLOPs, DID-S)  | 29.19     | **41.94** | **40.84** | 52.53     | **59.88** | **63.95** | 91.87      | 54.31       |
> |        | **DID-600k (75% FLOPs)**     | 28.48     | **41.30** | **39.44** | 49.80     | **58.76** | **62.54** | 85.98      | **52.33**   |
> |        | DID-800k (100% FLOPs, DID-F) | **28.35** | **41.00** | **38.71** | **48.84** | **58.05** | **61.77** | 87.09      | **51.97**   |
>
> W4: We understand the reviewer’s concern about DP efficiency. For a more detailed efficiency analysis of DP in DID training, please refer to **Common Concern 1**.
>
> Q1: Regarding the sensitivity to the denoising rounds and the chosen step schedule.
>
> - We have investigated the sensitivity to the denoising rounds in Tab. 2 and 4. In the variable-length setting (Tab. 4), DID is remarkably stable. The PPL remains consistently low across different budgets, ranging from 22.78 to 23.88. This stability is a significant advantage over the baselines, which show much higher sensitivity. In the fixed-length setting (Tab. 2), DID’s sensitivity is also lower than RADD, though the performance is not as stable as in the variable-length setting (Tab. 4).
> - We appreciate the suggestion to investigate more on the step schedule, to address this concern, in addition to the uniform timestep schedule researched in this work, we have added the experiment of cosine timestep schedule for inference (see **Common Concern 2**), which have demonstrated that DID is not sensitive to the chosen step schedule, the uniform schedule and the cosine schedule produce similar results, and notably, the speedup of DID is much more significant with a cosine schedule since more steps are used for early short sequences, reducing the average computational FLOPs of DID inference.

---

### Official Review · Reviewer_YTEE · 2025-11-10

**Soundness:** 3
**Presentation:** 3
**Contribution:** 3
**Rating:** 6
**Confidence:** 3

**Summary:**

The paper introduces Deletion-Insertion Diffusion (DID), a discrete diffusion paradigm for language modeling. DID replaces the standard masking/unmasking process of masked diffusion language models (MDLMs) with deletion/insertion processes. The key idea is that by deleting tokens in the forward process and inserting them in the reverse process, the model can naturally handle variable-length sequences without need for special mask or pad tokens. THe paper formalizes deletion/insertion as a continuous-time discrete diffusion process, presents a tractable objective for optimization, and presents empirical results that show that DID is faster (training/inference) and better at variable-length modeling than MDLM baselines.

**Strengths:**

The overall idea of the proposed method is creative and explores a new direction for language diffusion models. I see the main strengths of the paper as follows:

- method: the shift from masking to pure deletion/insertion for diffusion LMs is intuitive, and the mathematical formulation of the insertion score and the derivation of the objective via subsequence counts seem correct
- efficiency: the paper makes a strong case for the inefficiency of MDLMs due to mask and pad tokens, and the reported speedups (up to about 3x for training, and about 4x for inference) are substantial and practically important
- clarity: the paper is very well-written, albeit quite dense; the motivation is clear, the derivation of the method from standard discrete diffusion makes sense, and the connection between the math and the implementation is well-explained

**Weaknesses:**

I believe there are a few weaknesses:

- experiments: the experiments are relatively smalls-scale (OpenWebText, Stories) compared to state-of-the-art LLM research; while this is fine for providing a proof of concept, it's unclear if the reported efficiency gains hold or if new instabilities arise at larger scales
- complexity: this may be a nitpick, but the DP approach seems much more complicated and harder to implement than, say, standard cross-entropy or MDLM objectives

**Questions:**

- Are there any unique notable failure modes of the proposed method? Does the approach ever get stuck inserting repetitive loops in the middle of sentences?
- How does the computational overhead of the DP approach scale with the sequence length?
- Please comment on the weaknesses listed above

---

> ### Author Response · Authors · 2025-11-23
>
> Thank you for the recognition of our contributions and the thoughtful comments. Below, we address your concerns and suggestions.
>
> W1: We appreciate the reviewer’s concern regarding the scalability of DID to larger models and longer sequences. We have added corresponding experiments, please refer to **Common Concerns 1 and 3**.
>
> W2, Q2: We understand the reviewer’s concern about DP efficiency, and the DP approach of DID is indeed more complicated than the cross-entropy in MDLMs, but its time cost is not significant for sequences that are not ultra-long. We provide a more detailed efficiency analysis of DP in DID training in **Common Concern 1**.
>
> Q1: We have not identified unique failure modes structurally distinct from standard diffusion limitations (e.g., suboptimal generation coherence in low-step or small-model regimes). Regarding the concern about repetitive loops, which is a common failure in AR models, our qualitative analysis (Appendix H) indicates that DID is robust to this behavior. We support this assessment based on the model's intrinsic mechanism and our empirical observations.
>
> - **Intrinsic Self-Correction Mechanism:** Unlike AR models or MDLMs where generated or unmasked tokens typically become fixed, creating a risk of error accumulation, DID operates via iterative insertion processes. This provides an intrinsic self-correction mechanism that dynamically adjusts token positions during generation. This flexibility allows the model to insert new tokens to restructure segments, preventing it from getting locked into repetitive errors.
> - **Empirical Evidence:** Our analysis confirms that the model does not suffer from infinite looping failures. Observed repetitions, such as "ow, ow, ow, ow, it hurts!" (Appendix H.2, Sample 10), fit the context well (a crying child) and the model successfully continues generating the text afterwards. Furthermore, temporary repetitions during generation, such as "farm farm" at Step 38 in Appendix H.3, are effectively fixed by the final step. The model restructures them into distinct sentences ("...and the farm. at the farm..."), demonstrating its ability to correct errors rather than getting stuck.

---

### Author Response · Authors · 2025-11-23

We thank all reviewers for their time and constructive feedback, which has greatly improved our work. Below is our response to some common concerns. All of the following referred equations, figures, and tables correspond to the ones in the paper.

## Common Concern 1 [Reviewers YTEE, Bfo3, ciUs]: DP Efficiency in DID training, and Scalability to Longer Sequences

When we extend the DP algorithms (Eq. 12-15 in the paper) to longer sequences, we encounter a numerical issue that the value of $N(\boldsymbol{x}_t, \boldsymbol{x}_0)$ and the values in the DP tables (Eq. 13, 14) might be extremely large. For example, when $|\boldsymbol{x}_0|=2048$, a moderate sequence length longer than 1024 we used in the paper, the maximum $N(\boldsymbol{x}_t, \boldsymbol{x}_0)$ is $\tbinom{2048}{1024} \sim 10^{614}$, larger than the upper limit of float64 precision $\sim 10^{308}$, resulting in a numerical overflow. However, what we want to compute to implement the DISE loss (Eq. 11) of DID is the ‘N ratios’: $\frac{N(\text{Ins}(\boldsymbol{x}_t, i, v), \boldsymbol{x}_0)}{N(\boldsymbol{x}_t, \boldsymbol{x}_0)}$, according to the sequence-level normalization property (Eq. 16, details in Appendix D.7), the ratios should be  $\le|\boldsymbol{x}_0|- |\boldsymbol{x}_t|$, i.e. the final results of the N ratios will not have any overflow issues.

Therefore, to address the numerical overflow of the intermediate DP results, we have developed a log-domain DP to transform the DP algorithms into the log-domain, take the prefix DP (Eq.13) as an example, the original DP iteration:

$$ N(\boldsymbol{x}_t{[:i]}, \boldsymbol{x}_0{[:j]}) =  N(\boldsymbol{x}_t{[:i]}, \boldsymbol{x}_0{[:j-1]})+\delta(\boldsymbol{x}_t{[i-1]},\boldsymbol{x}_0{[j-1]}) \cdot N(\boldsymbol{x}_t{[:i-1]}, \boldsymbol{x}_0{[:j-1]}), $$

can be transformed as:

$$ \log N(\boldsymbol{x}_t{[:i]}, \boldsymbol{x}_0{[:j]}) = \log [e^{\log N(\boldsymbol{x}_t{[:i]}, \boldsymbol{x}_0{[:j-1]})}+e^{\log\delta(\boldsymbol{x}_t{[i-1]},\boldsymbol{x}_0{[j-1]})+\log N(\boldsymbol{x}_t{[:i-1]}, \boldsymbol{x}_0{[:j-1]})}], $$

which can be efficiently implemented with the ‘logaddexp’ operation provided by deep learning frameworks such as PyTorch, the DP table now stores the logN values instead of N values in the original version, and lower data precision (e.g. float32) could be enabled to save memory. Notably, this algorithm will encounter the log0 issue, we replace log0 with a large negative number (-999999 in our implementation). The resulting numerical error of our log-domain DP is negligible, approximately on the order of $10^{-13}$ when using float64 and $10^{-4}$ with float32. However, the method fails to perform correctly with float16 and bfloat16 data types.

Besides, we adopt several code optimizations to our DP implementation to make it faster, new features include:

- Combined prefix and suffix DP tables, so their updates could be performed in parallel.
- In-place operations to reduce the IO cost, since the DP does not require a backward method.
- Contiguous memory accessing for each iteration to reduce the IO cost.

We have added the pseudocode and PyTorch code of the DP algorithms to **Appendix G**.

---

> ### Author Response · Authors · 2025-11-23
>
> We denote the DP implementation after adopting these updates as version 1 (**v1**), and the original one in the paper as version 0 (**v0**). We perform a more detailed profiling of DP efficiency in the same training setting as the paper (Tab. 3, 5).
>
> **1.1 First, we analyze the time cost of DP-v0 and DP-v1 for different sequence lengths.** To align with the training setting in the paper, we set the batch size as 64, and sequence lengths (S) as 1k, 2k, 3k, and 4k. The time to perform 50 times of the DP algorithm to get the N ratios is:
>
> |       | S=1k     | S=2k                      | S=3k                      | S=4k                                  | The power law exponent $a$ (Time $=k * S^a$) |
> | ----- | -------- | ------------------------- | ------------------------- | ------------------------------------- | -------------------------------------------- |
> | DP-v0 | 4.62     | 10.9 (numerical overflow) | 25.1 (numerical overflow) | - (numerical overflow, out-of-memory) | 1.51                                         |
> | DP-v1 | **2.05** | **3.64**                  | **8.07**                  | **11.2**                              | **1.26**                                     |
>
> which demonstrates the efficiency improvement of DP-v1 over DP-v0, the power law exponent $a \in [1, 2]$  since the actual time complexity to update the 2-dimensional DP table is $O(S^2)$ and one dimension of the DP table could be updated in parallel.
>
> For extremely long sequences, our DP algorithm faces the classic quadratic bottleneck in both time and space complexities, which are not currently optimized. We leave the efficient algorithm and implementation for the N ratios computation for extremely long sequences to future work.
>
> **1.2 Then, we re-evaluate the training time of DID with DP-v1 and demonstrate the even improved training efficiency.**
>
> In the fixed-length training setting (Tab. 3):
>
> |            | small             | medium            | large             |
> | ---------- | ----------------- | ----------------- | ----------------- |
> | RADD       | 26.46             | 53.17             | 92.90             |
> | DID-v0     | 16.26 (1.63x)     | 31.16 (1.71x)     | 51.14 (1.82x)     |
> | **DID-v1** | **14.03 (1.89x)** | **27.77 (1.91x)** | **46.60 (1.99x)** |
> | DID-v1-DP  | 2.38 (17.0%)      | 3.33 (12.0%)      | 3.35 (7.2%)       |
>
> where medium and large models use more time for DP because the gradient accumulation is enabled in these settings (set to 2, as described in Appendix E.1), i.e. the same batch of data is divided into 2 parts, and DP is called twice, reducing the data parallelism and increasing the total time cost.
>
> We also re-evaluate the variable-length training setting  (Tab. 5):
>
> |           | small            | medium            | large             |
> | --------- | ---------------- | ----------------- | ----------------- |
> | RADD      | 19.93            | 37.87             | 67.75             |
> | DID-v0    | 10.59 (1.88x)    | 14.20 (2.67x)     | 21.83 (3.10x)     |
> | DID-v1    | **7.71 (2.58x)** | **12.30 (3.08x)** | **19.83 (3.42x)** |
> | DID-v1-DP | 2.13 (27.6%)     | 2.01 (16.3%)      | 2.01 (10.1%)      |
>
> where the DP proportion is more than that in the fixed-length setting, because we pad all sequences to the maximum length to perform batched DP, so the DP cost is similar, while the tokens input to the network are much fewer, since the average length of the Stories dataset is much shorter than 1024 used in the fixed-length setting. There are still inefficiencies in the current implementation of DP, and we leave a thorough system-level optimization to future work.

---

> ### Author Response · Authors · 2025-11-23
>
> ## Common Concern 2 [Reviewers Bfo3, ciUs, orpS, cWDw]: Inference Efficiency, Inference Strategies, and More Metrics
>
> We have added more experiments to analyze the performance of DID inference.
>
> **2.1 First, we perform more detailed efficiency analyses of DID inference.** Besides the wall time, we also report the time cost of specific parts, including neural function evaluation (NFE) and sampling.
>
> We re-evaluate the inference time of RADD and DID trained on the fixed-length data of OWT (Tab.2):
>
> |      | Steps        | 16                | 32                | 64                | 128               | 256               | 512               | 1024              |
> | ---- | ------------ | ----------------- | ----------------- | ----------------- | ----------------- | ----------------- | ----------------- | ----------------- |
> | RADD | Wall Time    | 0.240             | 0.334             | 0.507             | 0.872             | 1.627             | 2.876             | 4.506             |
> |      | **NFE**      | 0.080             | 0.161             | 0.323             | 0.642             | 1.262             | 2.234             | 3.308             |
> |      | **Sampling** | 0.019             | 0.036             | 0.071             | 0.141             | 0.280             | 0.558             | 1.115             |
> | DID  | Wall Time    | **0.171 (1.40x)** | **0.225 (1.48x)** | **0.359 (1.41x)** | **0.604 (1.44x)** | **1.029 (1.58x)** | **1.818 (1.58x)** | **2.987 (1.51x)** |
> |      | **NFE**      | **0.029 (2.76x)** | **0.076 (2.12x)** | **0.159 (2.03x)** | **0.323 (1.98x)** | **0.641 (1.97x)** | **1.131 (1.98x)** | **1.712 (1.93x)** |
> |      | **Sampling** | 0.011             | 0.031             | 0.069             | 0.144             | 0.293             | 0.589             | 1.182             |
>
> We observe that the NFE part is close to the theoretical 2x speedup, since DID saves about half the FLOPs on $\texttt{\<MASK\>}$  in the inference process compared to RADD. Notably, when the  total denoising steps are fewer, the speedup of NFE is more significant, this is because the maximum number of tokens that can be inserted in each step of DID inference cannot exceed the current length, when the total number of steps is small, the number of tokens decoded in the early steps will be bounded by this constraint, i.e. at step $i$, there are at most $2^{i-1}$ tokens input to the model, which brings greater NFE speedup of DID. Meanwhile, DID's performance is not affected by this reason. On the contrary, when the total number of steps is small, DID's decoding is much better than RADD’s, as shown in Tab.2, we have summarized our explanations on DID's high-quality fast sampling with few steps in **Common Concern 4**.
>
> Similarly, we re-evaluate the inference time of RADD and DID trained on the variable-length data of Stories (Tab. 4):
>
> |      | Steps        | 64                | 128               | 256               | 512               |
> | ---- | ------------ | ----------------- | ----------------- | ----------------- | ----------------- |
> | RADD | Wall Time    | 0.249             | 0.452             | 0.848             | 1.505             |
> |      | **NFE**      | 0.161             | 0.321             | 0.631             | 1.118             |
> |      | **Sampling** | 0.043             | 0.085             | 0.170             | 0.340             |
> | DID  | Wall Time    | **0.089 (2.78x)** | **0.135 (3.35x)** | **0.217 (3.91x)** | **0.384 (3.92x)** |
> |      | **NFE**      | **0.027 (5.96x)** | **0.058 (5.53x)** | **0.110 (5.74x)** | **0.224 (4.99x)** |
> |      | **Sampling** | **0.013 (3.30x)** | **0.027 (3.15x)** | **0.058 (2.93x)** | **0.110 (3.09x)** |
>
> where both NFE and sampling are accelerated since DID does not need to compute on the $\texttt{\<MASK\>}$ and $\texttt{\<PAD\>}$ tokens.

---

> > ### Author Response · Authors · 2025-11-23
> >
> > **2.2 Second, in addition to the uniform timestep schedule $T(i) = \frac{i}{N}$ evaluated in the paper, where $N$ is the total number of denoising steps in generation, we evaluate a different inference strategy of cosine timestep schedule $T(i) = \cos(\frac{\pi}{2}(1-\frac{i}{N}))$ , whose timesteps are denser at the beginning of the reverse process. [1]**
> >
> > We add the evaluation of the cosine timestep schedule using the fixed-length models trained on OWT:
> >
> > |      | Steps    | 16                | 32                | 64                | 128               | 256               | 512               | 1024              |
> > | ---- | -------- | ----------------- | ----------------- | ----------------- | ----------------- | ----------------- | ----------------- | ----------------- |
> > | RADD | PPL      | 220.74            | 140.23            | 107.42            | 93.72             | **85.16**         | 87.79             | **83.41**         |
> > |      | Entropy  | 8.24              | 8.17              | 8.14              | 8.12              | 8.09              | 8.09              | 8.08              |
> > |      | Time     | 0.240             | 0.364             | 0.516             | 0.904             | 1.606             | 2.817             | 4.567             |
> > |      | NFE      | 0.081             | 0.160             | 0.315             | 0.612             | 1.156             | 2.022             | 3.065             |
> > |      | Sampling | 0.023             | 0.046             | 0.090             | 0.178             | 0.356             | 0.708             | 1.414             |
> > | DID  | PPL      | **161.21**        | **115.54**        | **99.59**         | **92.52**         | 88.14             | **87.18**         | 87.00             |
> > |      | Entropy  | 8.14              | 8.12              | 8.11              | 8.08              | 8.09              | 8.09              | 8.08              |
> > |      | Time     | **0.212 (1.13x)** | **0.243 (1.50x)** | **0.314 (1.64x)** | **0.461 (1.96x)** | **0.830 (1.93x)** | **1.487 (1.89x)** | **2.536 (1.80x)** |
> > |      | NFE      | **0.028 (2.89x)** | **0.064 (2.56x)** | **0.134 (2.35x)** | **0.257 (2.38x)** | **0.514 (2.25x)** | **0.955 (2.12x)** | **1.579 (1.94x)** |
> > |      | Sampling | **0.01 (2.30x)**  | **0.026 (1.77x)** | **0.053 (1.70x)** | **0.106 (1.68x)** | **0.214 (1.66x)** | **0.429 (1.65x)** | **0.859 (1.65x)** |
> >
> > Similar to the experiment with the uniform timestep schedule (Tab.2), DID performs better when steps are fewer, and comparable or slightly inferior to RADD when using more denoising steps. Notably, we observe that the speedup of the cosine schedule is more significant (up to 1.96x) because under the cosine schedule, the average sequence length processed by DID is shorter than that of the uniform schedule, i.e. more NFEs are used for shorter sequences in the early stage of generation, resulting in greater acceleration of DID inference.
> >
> > **2.3 Third, we evaluate the sample quality of conditional generation using OWT prefixes under different prompt lengths of 256, 512, and 768:**
> >
> > | Prompt Length | Method | Steps   | 16        | 64        | 256       | 1024      |
> > | ------------- | ------ | ------- | --------- | --------- | --------- | --------- |
> > | 256           | RADD   | PPL     | 114.46    | 67.13     | 58.33     | 55.16     |
> > |               |        | Entropy | 8.12      | 8.03      | 8.03      | 8.01      |
> > |               | DID    | PPL     | **70.11** | **50.98** | **47.47** | **48.85** |
> > |               |        | Entropy | 7.92      | 7.94      | 7.92      | 7.92      |
> > |               |        | Length  | 1022.48   | 1023.81   | 1023.88   | 1024.00   |
> > | 512           | RADD   | PPL     | 52.31     | 38.17     | 35.51     | 35.11     |
> > |               |        | Entropy | 8.00      | 7.95      | 7.95      | 7.96      |
> > |               | DID    | PPL     | **35.29** | **29.95** | **29.19** | **29.00** |
> > |               |        | Entropy | 7.86      | 7.90      | 7.91      | 7.90      |
> > |               |        | Length  | 1018.66   | 1023.08   | 1023.99   | 1024.07   |
> > | 768           | RADD   | PPL     | 25.85     | 22.57     | 21.92     | 21.80     |
> > |               |        | Entropy | 7.85      | 7.84      | 7.84      | 7.84      |
> > |               | DID    | PPL     | **21.64** | **20.25** | **20.10** | **19.99** |
> > |               |        | Entropy | 7.83      | 7.86      | 7.87      | 7.86      |
> > |               |        | Length  | 1011.45   | 1023.38   | 1023.93   | 1024.10   |
> >
> > We observe that DID stably achieves lower PPL with comparable unigram entropy (diversity) and generation length, demonstrating the stable superiority of DID in conditional generation performance over RADD.

---

> > > ### Author Response · Authors · 2025-11-23
> > >
> > > **2.4 Fourth, we add more metrics to evaluate the sample quality.** We add the metrics of joint entropy measured by a stronger LLM scorer, Llama3.2-3B, and throughput (tokens per second), for the generation experiments of fixed-length models (Tab. 2):
> > >
> > > |      | Steps                       | 16           | 32           | 64           | 128          | 256         | 512         | 1024        |
> > > | ---- | --------------------------- | ------------ | ------------ | ------------ | ------------ | ----------- | ----------- | ----------- |
> > > | RADD | Joint Entropy (Llama3.2-3B) | 5.72         | 5.12         | 4.78         | 4.61         | **4.54**    | **4.46**    | **4.47**    |
> > > |      | Throughput (toks/s)         | 4655         | 3230         | 2052         | 1165         | 623         | 355         | 227         |
> > > | DID  | Joint Entropy (Llama3.2-3B) | **5.15**     | **4.77**     | **4.62**     | **4.58**     | 4.55        | 4.52        | 4.50        |
> > > |      | Throughput (toks/s)         | 6055 (1.30x) | 4163 (1.29x) | 2901 (1.41x) | 1787 (1.53x) | 978 (1.57x) | 561 (1.58x) | 341 (1.50x) |
> > >
> > > Similar to the experiments in the paper (Tab. 2), DID performs better when steps are fewer, and comparable or slightly inferior to RADD when using more denoising steps.
> > >
> > > For variable-length models (Tab. 4):
> > >
> > > |      | Steps                       | 64           | 128          | 256         | 512         |
> > > | ---- | --------------------------- | ------------ | ------------ | ----------- | ----------- |
> > > | ILM  | Joint Entropy (Llama3.2-3B) | 5.77         | 4.61         | 3.24        | 3.02        |
> > > |      | Length                      | 63.34        | 120.77       | 206.44      | 234.44      |
> > > |      | Throughput (toks/s)         | 3959         | 3552         | 2373        | 865         |
> > > | RADD | Joint Entropy (Llama3.2-3B) | 4.37         | 3.81         | 3.18        | 3.09        |
> > > |      | Length                      | 110.66       | 200.73       | 349.54      | 353.47      |
> > > |      | Throughput (toks/s)         | 450          | 455          | 423         | 242         |
> > > | DID  | Joint Entropy (Llama3.2-3B) | **2.80**     | **2.77**     | **2.81**    | **2.81**    |
> > > |      | Length                      | 182.31       | 193.77       | 202.97      | 204.96      |
> > > |      | Throughput (toks/s)         | 2026 (4.50x) | 1468 (3.23x) | 931 (2.20x) | 528 (2.18x) |
> > >
> > > Similar to the experiments in the paper (Tab. 4), DID stably outperforms both the baselines of ILM and RADD in terms of sample quality and speed.

---

> > > > ### Author Response · Authors · 2025-11-23
> > > >
> > > > ## Common Concern 3 [Reviewers YTEE, Bfo3]: Scalability of DID
> > > >
> > > > Regarding the scalability to larger models and longer sequences, following SMDM [2], we scale up the DID variable-length model to 1.1B parameters and train it on the SlimPajama dataset [3] for 1.6e21 FLOPs with a truncation length of 2048 (trained with the log-domain DP algorithm described in **Common Concern 1**), and evaluate the accuracy (%) of DID and SMDM on downstream commonsense reasoning tasks with the LM-Evaluation-Harness framework [4]:
> > > >
> > > > |                               | Arc-c     | Arc-e     | Hellaswag | Obqa      | Piqa      | Race      | Siqa      | Winogrande | Averaged accuracy |
> > > > | ----------------------------- | --------- | --------- | --------- | --------- | --------- | --------- | --------- | ---------- | ----------------- |
> > > > | SMDM-1.1B (1.6e21 FLOPs)      | 24.57     | 48.91     | 44.37     | 31.20     | 65.23     | **33.78** | **39.00** | 52.88      | 42.49             |
> > > > | DID-1.1B (8e20 FLOPs, 50%)    | **25.17** | 47.73     | 40.79     | **31.20** | 64.58     | 31.48     | 38.33     | 51.62      | 41.36             |
> > > > | DID-1.1B (1.2e21 FLOPs, 75%)  | **26.11** | **49.20** | 43.98     | **32.80** | **65.72** | 32.06     | 37.97     | **54.62**  | **42.81**         |
> > > > | DID-1.1B (1.6e21 FLOPs, 100%) | **26.54** | **49.37** | **45.72** | **32.00** | **66.16** | 32.44     | 38.89     | **54.85**  | **43.25**         |
> > > >
> > > > We observe that in the downstream task evaluations of the 1.1B models, DID can surpass MDM on the averaged accuracy with 50%-75% of the training FLOPs.

---

> ### Author Response · Authors · 2025-11-23
>
> ## Common Concern 4 (Reviewers ciUs, orpS): Theoretical Insights for the Advantages of DID over MDM
>
> We additionally summarize four structural advantages come from using insertion instead of unmasking during generation:
>
> - **Context-Aware Decoding Order:** In standard RADD sampling (e.g., $\tau$-leaping), the decision of which positions to update is governed solely by the noise schedule (random unmasking),  the probability that any masked position $i$ is unmasked in a step is governed by a scalar function $\psi(t,s)$, which depends only on the noise schedule (e.g., $\frac{t-s}{t}$ for log-linear): $\mathbb{P}[x _ s^{(i)} \neq [M]\mid \boldsymbol{x} _ t] = \psi(t,s).$ The probability is uniform across all masked position and independent of the context $\boldsymbol{x} _ t^{\mathrm{UM}}$. The learned model only determine what token to fill in if a position is unmasked, it does not influence where updates occur. When steps are few, this forces the model to predict many tokens at random absolute positions simultaneously, often leading to incoherence. In contrast, DID learns where to insert based on the existing content (via insertion score), the probability of an insertion after position $i$ is directly determined by the learned insertion score $\bar{s}_\theta(\boldsymbol{x} _ t, t)$: $\mathbb{P}[\text{Insertion at position } i \mid \boldsymbol{x} _ t] \approx \Delta t \cdot \frac{\sigma(t)e^{-\overline{\sigma}(t)}}{1-e^{-\overline{\sigma}(t)}}\sum _ {v\ne \varnothing} \bar{s} _ \theta(\boldsymbol{x} _ t, t)[i,v].$ Crucially, this mechanism is content-dependent. The model naturally chooses to generate tokens in positions where it is most probable first, improving the quality of each sampling step.
> - **Self-Correction Mechanism:** RADD relies on absolute positions. Once a token is unmasked, its content and position are fixed, so any early suboptimal predictions (common in fast sampling) lead to error accumulation. DID predicts relative order. Even if the model generates imperfect tokens early on, it can still refine the sentence structure in later steps by inserting new tokens between existing ones (see our demonstration in Appendix H.3). This dynamic refinement capability makes DID more robust to the errors inherent in few-step generation.
> - **Cautious Early Generation:** RADD works on the whole sequence length from the very first step, so it can unmask exceedingly many tokens at the earlier stage using inaccurate model predictions. In DID, we allow at most n+1 insertions if the current context length is n. (For example, the very first sampling step will insert at most 1 token.) This means we perform fewer insertions at the earlier stage, acting more cautiously than RADD.
> - **Better Handling of Adjacent Token Dependence:** Both DID and MDM employ the $\tau$-leaping method for fast sampling, which assumes multiple token transitions occur independently at each step, this independence assumption is a major source of sampling error. Note that DID will not insert multiple adjacent tokens at once, so the transitions of adjacent tokens are never independent. (Adjacent tokens are always inserted sequentially.) In contrast, $\tau$-leaping in MDM poses a stronger assumption that any pair of tokens are mutually independent. It is fair to hypothesize that adjacent tokens bear more dependence than separated tokens — thus, we believe DID handles token interdependence better.

---

> ### Author Response · Authors · 2025-12-01
>
> We further evaluate the performance of DID on the GSM8K \(Grade School Math 8K\) \[5\] benchmark for math word reasoning problems, using our 1.1B variable\-length model.
>
> Following the setup in SMDM \[2\], we fine\-tune the pretrained models—both SMDM\-1.1B and DID\-1.1B, which were pretrained on SlimPajama for 1.6e21 FLOPs—on the augmented training data \[6\] for 40 epochs, keeping all optimization configurations strictly consistent. We then assess the zero-shot accuracy \(%\) of DID and SMDM under various inference settings, including different decoding steps and top‑p sampling strategies, using the standard LM‑Evaluation‑Harness \[4\] framework.
>
> |top\-p strategy|steps|8|16|32|64|128|256|
> |---|---|---|---|---|---|---|---|
> |p=1.0 \(direct sampling\)|SMDM|27.82|33.97|35.78|36.85|36.54|36.69|
> ||DID|**31.92**|**35.41**|**37.45**|**39.35**|**39.27**|**39.88**|
> |p=0.9|SMDM|32.22|36.92|37.38|39.58|39.58|39.65|
> ||DID|**32.37**|**38.51**|**42.38**|**41.55**|**42.23**|**43.75**|
> |p=0.6|SMDM|36.01|40.03|40.56|42.00|43.37|42.61|
> ||DID|**38.82**|**41.39**|**43.90**|**45.26**|**46.47**|**46.70**|
> |p=0.3|SMDM|37.83|39.88|42.23|43.82|44.73|43.37|
> ||DID|**38.89**|**42.08**|**43.21**|**44.81**|**45.49**|**45.79**|
>
> Results demonstrate the consistent superiority of our variable\-length DID model over the fixed\-length, padding\-based SMDM approach on the conditional generation task of the GSM8K benchmark. We hope these results will help address concerns regarding scalability \(Reviewers YTEE, Bfo3\), evaluations \(Reviewers YTEE, Bfo3, cWDw\), conditional generation \(Reviewer cWDw\), and supervised fine\-tuning.
>
>
>
>
> [1] Jiaxin Shi, Kehang Han, Zhe Wang, Arnaud Doucet, and Michalis Titsias. Simplified and generalized masked diffusion for discrete data. Advances in neural information processing systems, 37:103131–103167, 2024.
>
> [2] Shen Nie, Fengqi Zhu, Chao Du, Tianyu Pang, Qian Liu, Guangtao Zeng, Min Lin, and Chongxuan Li. Scaling up masked diffusion models on text. arXiv preprint arXiv:2410.18514, 2024.
>
> [3] Daria Soboleva, Faisal Al-Khateeb, Robert Myers, Jacob R Steeves, Joel Hestness, and Nolan Dey. SlimPajama: A 627B token cleaned and deduplicated version of RedPajama. https://www.cerebras.net/blog/slimpajama-a-627b-token-cleaned-and-deduplicated-version-of-redpajama, 06 2023. URL https://huggingface.co/datasets/cerebras/SlimPajama-627B.
>
> [4] Leo Gao, Jonathan Tow, Stella Biderman, Sid Black, Anthony DiPofi, Charles Foster, Laurence Golding, Jeffrey Hsu, Kyle McDonell, Niklas Muennighoff, Jason Phang, Laria Reynolds, Eric Tang, Anish Thite, Ben Wang, Kevin Wang, and Andy Zou. A framework for few-shot language model evaluation, September 2021. URL https: [//doi.org/10.5281/zenodo.5371628](https://doi.org/10.5281/zenodo.5371628).
>
>
> [5] Karl Cobbe, Vineet Kosaraju, Mohammad Bavarian, Mark Chen, Heewoo Jun, Lukasz Kaiser, Matthias Plappert, Jerry Tworek, Jacob Hilton, Reiichiro Nakano, Christopher Hesse, and John Schulman. Training verifiers to solve math word problems. arXiv preprint arXiv:2110.14168, 2021a.
>
> [6] Yuntian Deng, Kiran Prasad, Roland Fernandez, Paul Smolensky, Vishrav Chaudhary, and Stuart Shieber. Implicit chain of thought reasoning via knowledge distillation. arXiv preprint arXiv:2311.01460, 2023.

---

### Author Response · Authors · 2025-12-03
**Summary**

Dear Area Chair,

We appreciate your time and effort in handling our submission. As the rebuttal phase is coming to an end, we would like to briefly summarize the following key aspects of our paper, as noted by the reviewers:

### DID is a Novel Approach.

- DID is a novel and theoretically grounded insertion-based diffusion language model, which enables variable-length modeling and eliminates the computational waste associated with the MASK and PAD tokens used in MDLMs. (Reviewers YTEE, Bfo3, ciUs, orpS, cWDw)
- The parallel dynamic programming to solve the subsequence counting problems for DID training is original and creative. (Reviewers Bfo3, cWDw)

### DID Introduces Boosted Performance.

- The speedup in training and inference brought by eliminating the computation for the MASK and PAD tokens is favorable and important. (Reviewers YTEE, Bfo3, orpS)
- The empirical performance of DID is promising, outperforming MDLMs in both fixed-length and variable-length settings. (Reviewers Bfo3, ciUs, orpS)

### Nice Presentation of the Paper.

- The paper is well-written, the motivation is clear, the derivations are self-contained, and the connection between math and implementation is well-explained. (Reviewers YTEE, Bfo3, ciUs, cWDw)

### Main Improvement for Rebuttal.

During the rebuttal phase, we conducted supplementary experiments and clarified more technical details of DID.

**Additional Experiments**.

- We conduct more detailed efficiency analyses of DID training and inference, and we have designed a log-domain DP algorithm to extend DID to longer sequences. (Common Concerns 1 and 2)
- We conduct additional experiments to evaluate DID inference, including the cosine timestep schedule, conditional generation, and additional metrics. (Common Concern 2)
- We scale up the DID variable-length model to 1.1B parameters and evaluate it on more downstream tasks (Arc-c, Arc-e, Hellaswag, Obqa, Piqa, Race, Siqa, Winogrande, and GSM8K) with the LM-Evaluation-Harness framework. Results have demonstrated the superiority of DID over MDM on language modeling and generation under the same compute budget. (Common Concern 3)

**Clarification about DID Technical Details.**

- We add more theoretical insights for the advantage of DID’s insertion over MDM’s unmasking in language generation. (Common Concern 4)
- We expand the discussion on the connections and advantages between DID and related works, e.g., MDM, additional insertion-based LMs, and EditFlow. (Reviewers ciUs, orpS, cWDw)
- We add more algorithm details for DID training and inference in Appendix G to enhance the clarity regarding the implementation. (Reviewer cWDw)

We thank all reviewers for their constructive feedback and positive evaluation, and hope our responses will help address their concerns. We will incorporate the new discussions in the rebuttal phase into the revised manuscript.

Best wishes,

Authors of Submission 16872

---

### Meta-Review · Area_Chair_gDHe · 2026-01-07

**Summary:**

This paper proposes Deletion-Insertion Diffusion (DID) language models, which replace the masking/unmasking paradigm of masked diffusion LMs with a deletion-based forward process and an insertion-based reverse process. The approach natively supports variable-length generation, eliminates MASK and PAD tokens, and introduces a score-based training objective solved via a dynamic programming formulation. Reviewers generally found the idea novel, theoretically grounded, and well motivated, with strong empirical evidence of improved training and inference efficiency and competitive or better modeling performance compared to MDLM baselines.

**Reviewer Concerns:**

Reviewers raised concerns about whether the proposed dynamic programming approach scales to longer sequences and larger models, whether its added complexity is justified relative to standard MDLM objectives, and how much overhead the DP introduces in practice. Additional concerns included the limited initial scale of experiments, the need for more detailed inference-time analyses across different step budgets and schedules, clearer comparisons to other insertion-based or diffusion variants, and stronger justification of why DID achieves both lower perplexity and faster sampling. In the rebuttal, the authors addressed these points by introducing a log-domain DP to improve numerical stability and scalability, providing detailed profiling of DP cost and end-to-end training/inference time, expanding inference analyses (including timestep schedules, conditional generation, throughput, and joint entropy metrics), scaling the model to 1.1B parameters with downstream evaluations, and adding clearer theoretical explanations and pseudocode. While the DP formulation remains more complex than standard objectives and extremely long-sequence scalability is left to future work, the major technical, empirical, and clarity concerns were largely resolved.

**Reviewer Scores:**

Reviewer scores were generally positive, with multiple reviewers rating the paper clearly above the acceptance threshold and others providing marginal-accept scores while explicitly noting they would not object to rejection. After the rebuttal, the added experiments, scalability analyses, and clarifications directly addressed the main sources of uncertainty raised by borderline reviewers.

---

### Decision · Program_Chairs · 2026-01-26

Accept (Poster)